# Scaling Reasoning Hop Exposes Weaknesses: Demystifying and Improving Hop Generalization in Large Language Models

**Zhaoyi Li**[1,2], **Jiatong Li**[1], **Gangwei Jiang**[1,2], **Linqi Song**[2,4*], **Defu Lian**[1*], **Ying Wei**[3*]
[1]University of Science and Technology of China, [2]City University of Hong Kong,
[3]Zhejiang University, [4]City University of Hong Kong, Shenzhen Research Institute
`{lizhaoyi777,cslijt,gwjiang}@mail.ustc.edu.cn`,
`linqi.song@cityu.edu.hk, liandefu@ustc.edu.cn,`
`ying.wei@zju.edu.cn`

## Abstract

Chain-of-thought (CoT) reasoning has become the standard paradigm for enabling Large Language Models (LLMs) to solve complex problems. However, recent studies reveal a sharp performance drop in *reasoning hop generalization* scenarios, where the required number of reasoning steps exceeds training distributions while the underlying algorithm remains unchanged. The internal mechanisms driving this failure remain poorly understood. In this work, we conduct a systematic study on tasks from multiple domains, and find that errors concentrate at token positions of a few critical error types, rather than being uniformly distributed. Closer inspection reveals that these token-level erroneous predictions stem from internal *competition mechanisms*: certain attention heads, termed *erroneous processing heads* (ep heads), tip the balance by amplifying incorrect reasoning trajectories while suppressing correct ones. Notably, removing individual ep heads during inference can often restore the correct predictions. Motivated by these insights, we propose test-time correction of reasoning, a lightweight intervention method that dynamically identifies and deactivates ep heads in the reasoning process. Extensive experiments across different tasks and LLMs show that it consistently improves reasoning hop generalization, highlighting both its effectiveness and potential. The data and code are made public[1].

## 1 Introduction

While Chain-of-thought (CoT) (Wei et al., 2022) has become the de facto approach for eliciting multi-hop reasoning in large language models (LLMs), recent works (Anil et al., 2022; Shojaee et al., 2025; Zhao et al., 2025) have reported a striking failure mode in ***reasoning hop generalization***: when the number of required reasoning hops[2] increases at test time, even the most advanced LLMs' performance drops drastically, despite the required *reasoning skill* remaining unchanged (e.g., 3-digit $\times$ 8-digit multiplication requires the same multi-digit multiplication skill as multiplying two 2-digit numbers). This generalization problem is a special case of length generalization (Kazemnejad et al., 2023) tailored to reasoning problems; unlike sequence-length generalization that stresses model context-window limits, hop generalization concerns the growth of reasoning hops where the total context length likely remains modest. This pronounced gap from human-level robust reasoning (Dekker et al., 2022; Lake & Baroni, 2023) motivates the urgency to examine why LLMs struggle with long-hop reasoning and how to bridge this gap on the path toward artificial general intelligence.

To this end, Dziri et al. (2023) systematically studied and attributed this problem to single-hop error accumulation. Hu et al. (2025) introduced a rule-following fine-tuning strategy that relies

---

*Corresponding authors

[1]`https://github.com/Zhaoyi-Li21/reasoning_hop_generalization`
[2]"reasoning hop" in our paper refers to a general reasoning step on a reasoning chain, which includes but not limits to the "hop" used in the multi-hop QA literature (Sakarvadia et al., 2024).

on fine-tuning on a specific reasoning task with its rules recited before each execution, while Fan et al. (2025) proposed a new architecture, looped transformer, that reuses computation across hops for generalizing to longer reasoning chains. While these approaches mark promising directions, a central challenge persists: how to compatibly enhance the intrinsic reasoning capabilities of off-the-shelf LLMs without post-training on downstream data. This enduring difficulty stems largely from *a lack of understanding of the internal mechanisms underlying reasoning hop generalization failures*, which constitutes the core motivation of our work.

To unveil the mechanistic causes of hop generalization errors, we investigate two research questions. ***RQ1: Where do errors occur?*** Existing mechanistic analysis tools (Bereska & Gavves, 2024) were mainly developed for local prediction tasks, such as factual recall (Meng et al., 2022), simple arithmetic (Stolfo et al., 2023), or elementary in-context learning tasks (Todd et al., 2024), where relevant signals are confined to one or a few tokens. In contrast, hop generalization errors unfold over many reasoning hops, making it unclear which token-level decisions are responsible, and rendering direct application of these tools intractable. To address this, we adopt an error-centric perspective and systematically identify *key error types* – task-specific token positions where failures consistently concentrate. This reduces the complexity of long-hop reasoning into a compact set of token-level diagnostic entry points and makes them amenable to mechanistic investigation.

Next, ***RQ2: Why do these errors arise?*** Understanding the error causes from these entry points poses great challenges, as they may arise from multiple cooperating or competing mechanisms with direct and indirect effects on the prediction, for which a single analysis tool is insufficient. To disentangle the underlying mechanisms, we combine Logit Lens decoding (nostalgebraist, 2020), head knockout interventions (Michel et al., 2019b), and activation-based circuit analysis (Wang et al., 2023). Together, these tools reveal a simplified reasoning circuit for token-level predictions (Figure 1): *correct/erroneous processing heads* (cp/ep heads) that cooperate with *basic processing heads* in shallow layers construct intermediate reasoning signals related to correct/erroneous

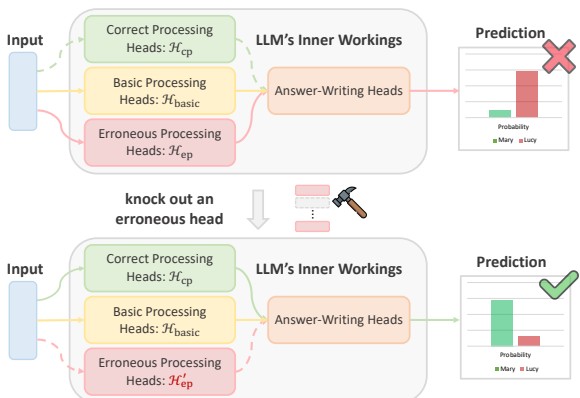

Figure 1: Illustration of the reasoning circuit and the competition mechanism inside LLMs.

predictions, while *answer-writing heads* (aw heads) in deeper layers aggregate these signals into the residual stream. Within this circuit, we uncover a ***competition mechanism***: correct and erroneous ***trajectories***[3] coexist; when ep heads tip the balance by amplifying spurious signals and suppressing correct ones, they drive erroneous predictions. Notably, beyond tracing errors back to their heads, we observe shared ep heads across various reasoning tasks and error types, laying the foundation for predicting such ep heads on the fly to intervene.

Building on the above mechanistic insights, we develop **Test-time Correction of Reasoning (TCR)**, a lightweight intervention method that is compatible with most LLMs, to dynamically rectify hop generalization errors. TCR maintains a candidate set of common ep heads per model, trains a head selector network to automatically choose which head to knock out given the input context, and employs an entropy-threshold detector to decide when to intervene. Extensive experiments show that TCR consistently improves reasoning hop generalization (e.g., averagely +6.8% accuracy on seven tasks with Qwen2.5-7B-Instruct). We also implement TCR-gold, which uses an oracle detector to precisely localize all errors, boosting Qwen2.5-7B-Instruct's average accuracy by about 20% (41.7% → 61.3%), highlighting the strong test-time rectification potential of knocking out ep heads.

**Main findings and contributions** Through extensive experiments on seven reasoning hop generalization tasks and across four open-source LLMs, we arrive at three key findings: (1) Errors in hop-generalization scenarios are dominated by token-level erroneous predictions of ***a few specific error types***, enabling focused diagnosis. (2) These token-level errors stem from the internal ***compe-***

---

[3]referring to information flows inside the reasoning circuit that support correct or erroneous predictions.

***tition mechanism***: correct trajectories co-exist with erroneous ones but are actively suppressed by erroneous processing heads; knocking out such heads can substantially restore correct predictions. (3) We propose ***TCR*** based on these insights, a test-time intervention that automatically and dynamically deactivates erroneous processing heads, consistently improving reasoning hop generalization.

## 2 PRELIMINARIES

In this section, we introduce the tasks and models that used in our study and basic ideas and tools of analyzing LLMs' internal mechanisms from the residual stream viewpoint. Due to the page limit, a detailed discussion of related work is provided in Appendix C.

**Tasks** Following the task forms widely used in the relevant literature, we collect seven reasoning hop generalization tasks from three different domains:

- *Symbolic Reasoning*: **Parity-NL** (Parity problem, natural language variant) (Anil et al., 2022; Zhou et al., 2024) and **LLC** (Last Letter Concatenation) (Wei et al., 2022; Hu et al., 2025),
- *Mathematical Calculation*: **MDM** (Multi-Digit Multiplication) (Dziri et al., 2023) and **MOAS** (Multi-Operand Addition and Substract) (Baeumel et al., 2025),
- *Coding*: **CLF** (LeetCode 1598) and **NumS** (LeetCode 1450),
- *Others*: **ObjC** (Object Counting) that requires both symbolic reasoning and mathematical calculation abilities from the BBH benchmark (Suzgun et al., 2022).

Inspired by Mirzadeh et al. (2025), we resynthesize data for each task by randomly substituting entity names and numerical values to minimize potential data leakage from LLM pretraining. A key advantage of these tasks is the ability to control reasoning hops while preserving the underlying reasoning skill (e.g., in Parity-NL, we instantiate problems with $n \in \{10, 20, \ldots, 50\}$ actions). Detailed task descriptions and examples are provided in Appendix D.

**Models** We experiment with four open-source LLMs: Qwen2.5-7B-Instruct (Alibaba, 2025a), Phi-3-Instruct (Microsoft, 2024), LLaMA3-8B-Instruct (Meta, 2024), and Qwen3-8B-Instruct (Alibaba, 2025c). Model details are in Appendix D, with extended results of the experiments presented in the main text provided there due to space constraints.

**Understanding LLM Predictions via the Residual Stream.** A key feature of Transformer-based LLMs is the *residual stream* (Elhage et al., 2021; Todd et al., 2024), which cumulatively propagates information across layers. Each layer reads the residual state, processes it via attention and MLP submodules, and writes updates back through addition. Formally, letting $\mathbf{h}^l \in \mathbb{R}^d$ be the residual state at layer $l \in [0..L-1]$, we have (omitting LayerNorm (Ba et al., 2016) for simplicity):

$$\mathbf{h}_{\text{mid}}^{l+1} = \mathbf{h}^l + \sum_{i=1}^{H} \mathbf{a}_i^l, \quad \mathbf{h}^{l+1} = \mathbf{h}_{\text{mid}}^{l+1} + \mathbf{m}^l = \mathbf{h}^l + \sum_{i=1}^{H} \mathbf{a}_i^l + \mathbf{m}^l, \tag{1}$$

where $\mathbf{a}_i^l$ is the output of the $i$-th attention head and $\mathbf{m}^l$ is the MLP output. This decomposition enables interpretable circuit analysis (Wang et al., 2023), attributing prediction dynamics to individual components by tracking how each head and MLP contributes through the residual stream.

**Mechanism Analysis Tools**
*Logit Lens* is a widely used tool that projects intermediate values in the LLM residual stream into the human-interpretable vocabulary space (Geva et al., 2023; Chughtai et al., 2024; Yu & Ananiadou, 2024) by applying the unembedding matrix $W_U \in \mathbb{R}^{d \times |V|}$ (in the output layer of LLMs):

$$f_{\text{logit-lens}}(\mathbf{v}) = \text{Softmax}(\mathbf{v} \cdot W_U) \in \mathbb{R}^{|V|}, \tag{2}$$

where $\mathbf{v} \in \mathbb{R}^d$ can be any intermediate representation in the LLM residual stream (Dar et al., 2023), such as $\mathbf{h}^l$ or $\mathbf{a}_i^l$, and $V$ is the vocabulary. Given a target token (index) $t \in [0..|V|-1]$, we can use $f_{\text{logit-lens}}(\mathbf{v})[t]$ to directly assess how much information about $t$ is contained in $\mathbf{v}$.

*Knocking Out*, initially inspired by biological genetic analysis (Griffiths, 2005), has been widely used in recent LLM mechanistic analysis works (Wang et al., 2023; Geva et al., 2023; Dutta et al., 2024) to indirectly assess the causal importance of specific components to the model prediction. Specifically, taking the attention head $\mathbf{a}_i^l$ for an example, we knock it out by setting $\mathbf{a}_i^l = \mathbf{0}$ (Michel et al., 2019a) when updating the residual stream (Eq 1). We then compute the change in the final output probability distribution as the Causal Indirect Effect (CIE) (Meng et al., 2022; Stolfo et al.,

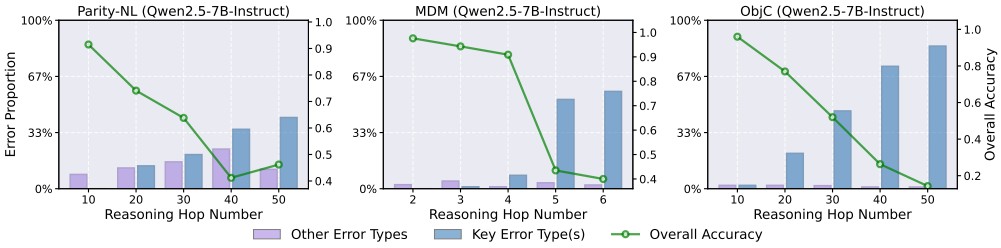

Figure 2: Overall accuracy (green curve) and error proportion for key error types (blue bar) and other error types (purple bar) of Qwen2.5-7B-Instruct on Parity-NL, MDM and ObjC.

2023) of knocking out $\mathbf{a}_i^l$. Formally,

$$g_{\text{knock-out}}(\mathbf{a}_i^l) = \mathbf{p}_\theta(x) - \mathbf{p}_\theta(x|\mathbf{a}_i^l = \mathbf{0}) \in \mathbb{R}^{|V|}, \tag{3}$$

where $\mathbf{p}_\theta$ is the predicted probability distribution of the model $\theta$ and $x$ is the model input. Large value of $g_{\text{knock-out}}(\mathbf{a}_i^l)[t]$ indicates the head knocked out plays a key role in driving the prediction of the target token $t$ (Heimersheim & Nanda, 2024). This procedure allows us to pinpoint the attention heads that are most critical for specific token-level reasoning patterns.

# 3 BREAKDOWN OF REASONING HOP GENERALIZATION ERRORS

In this section, we mainly want to answer RQ1: ***Where do errors occur?*** Specifically, how do LLMs' reasoning errors evolve as problem hop increases, and are there any specific error types and token-level mistakes account for their performance degradation?

**Decomposing LLMs' Overall Reasoning Errors into Reasoning Hops** Interpreting CoT reasoning is substantially harder than simpler objectives (e.g., factual recall (Meng et al., 2022; Li et al., 2024), basic arithmetic (Stolfo et al., 2023), or elementary in-context tasks like `get_antonym` (Todd et al., 2024; Jiang et al., 2025)) because the latter involve only a few output tokens, whereas CoT responses often span hundreds of tokens, especially under reasoning hop generalization. Existing mechanistic analysis tools excel at token-level analysis but become difficult to apply when it is unclear which token positions should serve as the entry points for analysis. To address this, we decompose a CoT response into fine-grained reasoning hops. Formally, for an $n$-hop problem $x \to r_1 \to \cdots \to r_n \to y$, the joint probability of a fully correct response factorizes as:

$$\underbrace{p(r_1, r_2, ..., r_n, y|x)}_{\text{The whole CoT response}} = \underbrace{p(r_1|x)}_{\text{1st Hop Reasoning}} \cdot \underbrace{p(r_2|x, r_1)}_{\text{2nd Hop Reasoning}} \cdots \underbrace{p(r_n|x, r_1, ..., r_{n-1})}_{n\text{th Hop Reasoning}} \cdot \underbrace{p(y|x, r_1, ..., r_n)}_{\text{Output the final answer}}.$$

Failures in CoT reasoning can thus be traced to failures in specific hops, and ***the token positions of first errors serve as a natural proxy for diagnosing the overall reasoning failure***. In the following, we focus on these positions for mechanistic analysis.

**Key Error Types: Performance Degradation is Rooted in a Few Patterns** In reasoning hop generalization, problems often contain dozens of intermediate hops Hu et al. (2025), each potentially producing multiple errors. Considering all possibilities quickly becomes intractable. We observe that many errors share the underlying reasoning skill, even if they occur in different hops. Accordingly, we categorize a small set (typically $5 \sim 10$ per task) of task-specific "**error types**". Figure 9 illustrates all error types for Parity-NL (detailed examples for other tasks are in Appendix E.1), which include, e.g., recalling wrong information and incorrectly updating state. For brevity, we denote the N-th error type of task X as "*X(N)*".

While decomposing reasoning errors catalogs the diverse mistakes that can occur, a key question remains: ***are errors evenly distributed or dominated by a few patterns?*** This distinction is critical, as concentration in a few patterns likely signals a coherent underlying mechanism to our study. We do observe that a few (one or two) error types for each task account for most errors, and define those accounting for $\geq 30\%$ as key error types (detailed in Appendix E.2). For instance, in the 50-hop Parity-NL task, $78.6\%$ of errors stem from recalling wrong names (Type 2 in Figure 9). We show the **error proportion** (number of error samples divided by the number of all samples) for

these key types and all other types and the **overall accuracy** with different reasoning hop numbers for Qwen2.5 on Parity-NL, MDM and ObjC tasks in Figure 2. We observe two key trends. First, the model experiences a pronounced drop in overall accuracy as the number of reasoning hops increases, confirming the fragility of current LLMs a problem involving more hops of reasoning. More importantly, this decline is not explained by a uniform increase in all error types, but is instead largely driven by the growth of a small set of target error types we identify. In contrast, the aggregate proportion of all other error types fluctuates only mildly with reasoning hop number. This highlights that *performance degradation in reasoning hop generalization stems primarily from a limited set of key error types that intensify with hop length*, underscoring the importance of focusing on the corresponding token positions where such errors occur.

## 4    PROBING INTERNAL MECHANISMS OF KEY ERROR TYPES

We now turn to the central question RQ2: ***Why do these errors arise?*** What internal mechanisms underlie these critical reasoning errors, and how do they shape the model's corresponding token-level predictions? Uncovering the underlying mechanisms may reveal potential intervention points to mitigate them. **We conduct comparative analysis (as illustrated in Figure 3) on both correct and erroneous model predictions at the token positions corresponding to key error types.** We start by noting that *the erroneous predictions at these token positions are often associated with higher predictive uncertainty*. A detailed analysis of token-level entropy for correct and erroneous predictions across seven reasoning hop generalization datasets is provided in Appendix F.1. This observation motivates our study of the underlying *competition mechanisms* behind such high uncertainty, where there must exist correct and erroneous reasoning trajectories to shape the error patterns in reasoning hop generalization.

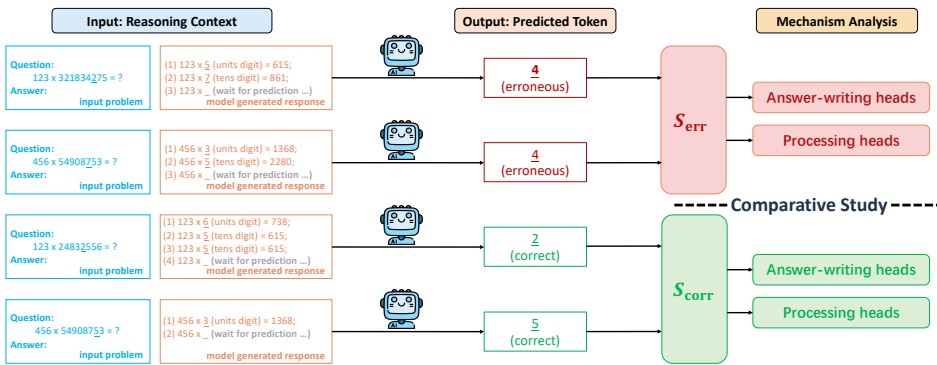

Figure 3: Illustration of the idea of comparative study and constructing the correct prediction set $S_{corr}$ and erroneous prediction set $S_{err}$ with the multi-digit multiplication (MDM) task (error type 2, digit decomposition error). Each sample contains the input reasoning context and the output predicted token. $S_{corr}$ and $S_{err}$ contains the samples that are correctly and erroneously predicted by model, respectively.

**Competition Mechanism Inside LLMs in Reasoning Hop Generalization** Motivated by prior work highlighting the crucial role of attention heads in robust CoT reasoning (Olsson et al., 2022; Dutta et al., 2024; Cabannes et al., 2024), we focus our analysis on attention heads. We first identify and categorize two functionally distinct groups central to CoT reasoning: 1) **answer-writing heads (aw heads)**, mainly in middle-to-deep layers, inject answer information into the residual stream to directly shape model predictions, and 2) **processing heads**, mostly in shallow-to-middle layers, extract and refine semantic information from the input and intermediate steps to support reasoning indirectly. Here we specifically analyze the competition mechanisms of both categories using Qwen2.5-7B-Instruct on Parity-NL (additional results for other models and tasks are in Appendix F).

### 4.1    ANSWER-WRITING HEADS

**Locating answer-writing heads** Considering a specific error position where the predicted token is $t$, previous works (Wang et al., 2023; Dutta et al., 2024; Chughtai et al., 2024) typically measure

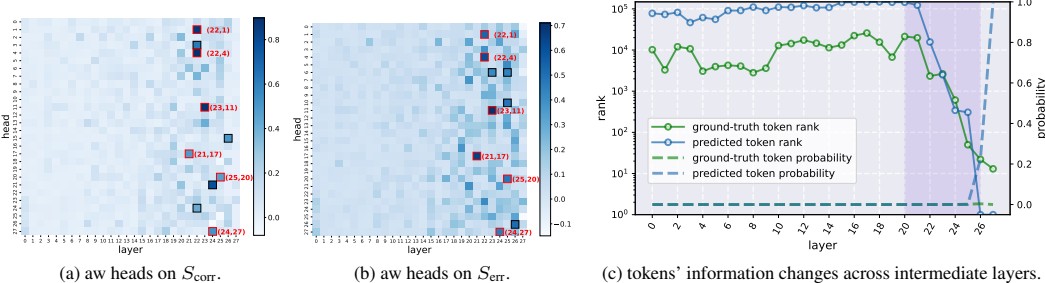

(a) aw heads on $S_{\text{corr}}$.     (b) aw heads on $S_{\text{err}}$.     (c) tokens' information changes across intermediate layers.

Figure 4: Locating answer-writing heads with our proposed measure for $S_{\text{corr}}$ and $S_{\text{err}}$ and tracing ground-truth and predicted tokens' information across intermediate layers.

how much information about $t$ is written by head $\mathbf{a}_i^l$ (at the **last input token position**) with Logit Lens $f_{\text{logit-lens}}(\mathbf{a}_i^l)[t]$ (Eq.2). However, this approach does not fully capture a head's contribution, as it ignores the joint influence of other heads in the same layer and the information already present in the residual stream (details in Appendix F.3). To address this, we measure the effect of knocking out $\mathbf{a}_i^l$ on the Logit Lens probability $f_{\text{logit-lens}}(\mathbf{h}_{\text{mid}}^{l+1})[t]$ to locate aw heads. Besides, considering the large scale discrepancy of target-token probabilities across layers, which fluctuate only within a narrow band in earlier layers but surge dramatically in the final layers (Figure 4(c)), we additionally normalize the effect with its original value. In conclusion, we propose a metric to measure how strongly $\mathbf{a}_i^l$ acts as an aw head, formally expressed as:

$$(f_{\text{logit-lens}}(\mathbf{h}_{\text{mid}}^{l+1})[t] - f_{\text{logit-lens}}(\mathbf{h}_{\text{mid}}^{l+1} - \mathbf{a}_i^l)[t]) / f_{\text{logit-lens}}(\mathbf{h}_{\text{mid}}^{l+1})[t] \triangleq s_{\text{aw-head}}(\mathbf{a}_i^l). \tag{4}$$

For a specific error type, we randomly sample two sets for correction and erroneous predictions at the corresponding key token position, denoted as $S_{\text{corr}}$ and $S_{\text{err}}$ (the illustration can be found in Figure 3), respectively. We locate aw heads for them by calculating the average $s_{\text{aw-head}}(\mathbf{a}_i^l)$ over $S_{\text{corr}}$ and $S_{\text{err}}$ for each attention head $\mathbf{a}_i^l$. The locating results are shown in Figure 4(a) and (b). There exist *a few heads, sparsely distributed, that receive markedly high average* $s_{\text{aw-head}}(\mathbf{a}_i^l)$ *values* (around $0.8$). We use bold boxes to highlight the locations (e.g., "(22,1)" refers to $\mathbf{a}_1^{22}$) of top10 (i.e.,$10/784 \approx$ top1.3%) attention heads. Surprisingly, 6 out of $10 = 60\%$ heads overlap in the top10 heads for both correct and erroneous predictions (this overlap rate remains if we check the top20 heads). In addition, for all samples in $S_{\text{err}}$, we collect the hidden states of all intermediate layers, $\{\mathbf{h}^l\}_{0 \leq l \leq L}$, and use Logit Lens to decode them, $\{f_{\text{logit-lens}}(\mathbf{h}^l)\}_{0 \leq l \leq L}$, with which we draw the average (descending) ranks and the average probabilities of the erroneously predicted tokens and the corresponding ground-truth tokens in Figure 4(c). The purple shaded region, where the average ranks of *both predicted tokens and ground-truth tokens shift from over* $10^4$ *to within* $10$, highly aligned with the primary location of aw heads (layer $20 \sim 26$) shown in Figure 4(a,b). This consistency provides further evidence for our aw heads locating results and we demonstrate the superiority of our locating measure by comparing its consistency with previous methods' in Appendix F.3.

**Inspecting the aw heads** Then we look into these aw heads: *What information do they primarily encode?* To answer this question, we step in to apply the Logit Lens to directly inspect these attention heads. We decode the $\mathbf{a}_1^{22}$ and $\mathbf{a}_{11}^{23}$ with all samples in $S_{\text{err}}$ and observe that both of the erroneously predicted tokens and corresponding ground-truth tokens rank among the top candidates:

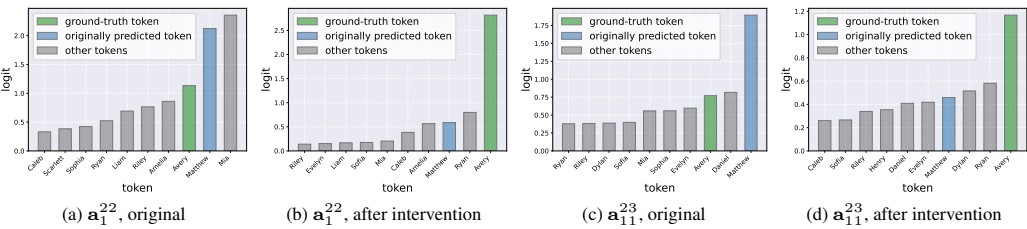

(a) $\mathbf{a}_1^{22}$, original    (b) $\mathbf{a}_1^{22}$, after intervention    (c) $\mathbf{a}_{11}^{23}$, original    (d) $\mathbf{a}_{11}^{23}$, after intervention

Figure 5: Decoding the information in aw heads: $\mathbf{a}_1^{22}$ and $\mathbf{a}_{11}^{23}$. (a,c) and (b,d) show their top10 decoded tokens before (i.e., the original model) and after knocking out the processing head $\mathbf{a}_7^0$.

they averagely rank $2.47$ and $4.33$. In Figure 5(a,c), we show the result with the logits of top10 candidates for a sample (see the result for all samples in Appendix F.3). We conclude that: ***correct and erroneous predictions largely share answer-writing heads, which encode signals for both ground-truth and erroneous predictions. In error cases, the erroneous signals dominate, but the ground-truth signals still compete.*** We next investigate the information processing mechanisms in the shallow-to-middle layers that give rise to this competitive dynamic.

## 4.2 PROCESSING HEADS

**Locating processing heads**  We identify attention heads that process input information and perform reasoning latently ("processing heads") to contribute to token predictions at critical error positions in reasoning hop generalization. Unlike aw heads, these processing heads: (i) influence the model's predictions indirectly, through intermediate reasoning that supports aw heads, and thus can act in **all input token positions** and (ii) exert a more fundamental impact on the model's final prediction, rather than directly writing prediction-related information into the residual stream. Based on these considerations, we adopt the knocking out method to intervene individual attention heads $\mathbf{a}_i^l$ at all token positions and measure the CIE (i.e., the $g_{\text{knock-out}}(\mathbf{a}_i^l)$ in Section 2) by the resulting change in the probability of the model's original prediction. We separately locate the processing heads for correct and erroneous predictions, correct and erroneous processing heads (in short, **cp heads** and **ep heads**, denoted as $\mathcal{H}_{\text{cp}}$ and $\mathcal{H}_{\text{ep}}$, as illustrated in Figure 1), by calculating the average CIE of $S_{\text{corr}}$ and $S_{\text{err}}$. The locating results are shown in Figure 6(a) and (b), where approximately $2\%$ heads with top average CIE values are highlighted with bold boxes. Analyzing the results, we first observe that knocking out some **"basic heads"** (i.e., $\{\mathbf{a}_i^0\}_{i \in \{3,5\} \cup [22,27]} \cup \{\mathbf{a}_i^3\}_{i \in \{11,18\}} \triangleq \mathcal{H}_{\text{basic}}$) would make the probabilities of original predictions (on both $S_{\text{corr}}$ and $S_{\text{err}}$) drop to near zero, indicating that they are commonly activated and play a basic role in extracting and processing basic and necessary information from the input to build up valid predictions, including both correct and erroneous ones.

Taking these heads away, the remaining processing heads (i.e., $\mathcal{H}_{\text{cp}}$ and $\mathcal{H}_{\text{ep}}$) are *almost entirely disjoint*. Such decoupled mechanisms of controlling correct and erroneous predictions are ideal for us to steer the model to correct its reasoning errors.

**Knocking out an ep head can rectify the erroneous predictions**  We find that knocking out heads in $\mathcal{H}_{\text{ep}}$ indeed can improve the probabilities of predicting ground-truth tokens. Specifically, we measure the average CIE of knocking out every single head on the *prediction of ground-truth tokens* on $S_{\text{err}}$, as shown in Figure 6(c). We observe: (i) knocking out heads in $\mathcal{H}_{\text{basic}}$ is bad for the prediction of the ground-truth tokens and (ii) the highlighted heads with the 10 smallest average CIE values, knocking out which improves the probability of ground-truth token the most, *perfectly align with $\mathcal{H}_{\text{ep}}$*. Specifically, knocking out a head $\in \{\mathbf{a}_0^0, \mathbf{a}_1^0, \mathbf{a}_7^0, \mathbf{a}_{14}^{13}\}$ improves the average probability ($\in [0, 1]$) of predicting ground-truth tokens by approximately $0.6$ on $S_{\text{err}}$. Moreover, to investigate if such rectification effects can generalize to other samples, we randomly resample 100 erroneously predicted and 300 correctly predicted samples for Parity-NL and MDM tasks, respectively. Figure 7 shows the distribution of the probabilities of predicting ground-truth answers in these samples with the original and intervened (i.e., knocking out a head $\in \mathcal{H}_{\text{ep}}$) models (details in Appendix F.2): for both two tasks, *knocking out individual ep heads can effectively rectify the model's originally erroneous prediction at critical token-level error positions in reasoning hop generalization while maintaining the originally correct predictions.* To investigate **whether the internal mechanisms of rectified and originally correct predictions are identical**, we re-locate cp heads and re-inspect the aw heads for these rectified predictions ($S_{\text{err}}$). To examine **whether rectified predictions rely on the same internal mechanisms as originally correct ones**, we re-locate cp heads and re-inspect aw heads for rectified samples ($S_{\text{err}}$). As shown in Figure 6(d), the top15 cp heads overlap by $93.3\%$ with those in Figure 6(a), indicating that rectified predictions largely share the reasoning mechanisms as correct ones. For aw heads (Figure 5(b,d)), we observe that ground-truth information is amplified while the erroneous prediction is suppressed in the decoding distributions of $\mathbf{a}_1^{22}$ and $\mathbf{a}_{11}^{23}$. Together, these findings support our hypothesis in Figure 1: ***correct reasoning mechanisms (cp heads) exist within LLMs but are suppressed by interactions with ep heads, leading to erroneous outputs; removing ep heads allows these mechanisms to rectify correct predictions.***

**Discussion**  Our results have established that a greater number of reasoning hops is associated with more errors of some certain key types, while the competition between correct and erroneous

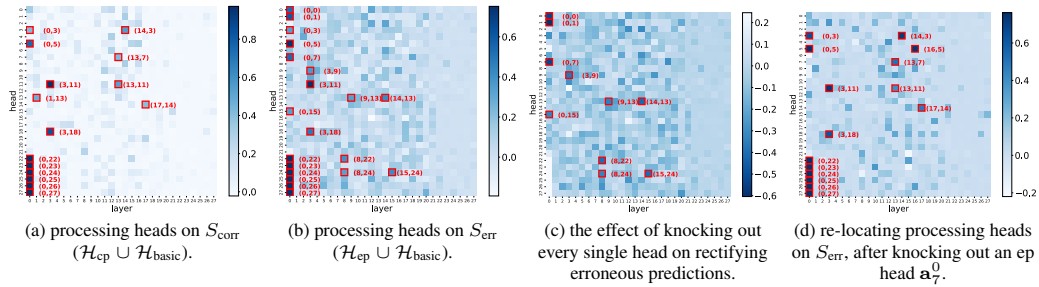

(a) processing heads on $S_{\text{corr}}$ ($\mathcal{H}_{\text{cp}} \cup \mathcal{H}_{\text{basic}}$).

(b) processing heads on $S_{\text{err}}$ ($\mathcal{H}_{\text{ep}} \cup \mathcal{H}_{\text{basic}}$).

(c) the effect of knocking out every single head on rectifying erroneous predictions.

(d) re-locating processing heads on $S_{\text{err}}$, after knocking out an ep head $\mathbf{a}_7^0$.

Figure 6: Locating processing heads for correct and erroneous predictions and measuring the effects of knocking out every single head on predicting ground-truth tokens.

trajectories contributes to these errors. Together, these findings suggest the hypothesis that **additional reasoning hops may intensify head competition**, and we next discuss the reasons why this hypothesis holds. First, longer reasoning chains are typically accompanied by larger input scales (e.g., more operations mentioned in text, longer input lists in coding problems, or numbers with more digits) as well as a greater number of intermediate reasoning states that must be tracked during generation, which *substantially enlarges the search space of retrieving the correct reasoning trajectories (i.e.,$\mathcal{H}_{cp}$)*. Second, when the required hop length significantly exceeds what the model has encountered during training (Dziri et al., 2023; Zhao et al., 2025) (i.e., reasoning hop generalization), these correct reasoning trajectories that the model has implicitly developed may fail to generalize more frequently: At certain key error positions, input contexts can still steer the model to follow $\mathcal{H}_{\text{cp}}$ and produce correct predictions, but more often $\mathcal{H}_{\text{cp}}$ is overridden by $\mathcal{H}_{\text{ep}}$, which may capture only local patterns that drive shortcut reasoning (Liu et al., 2023; Li et al., 2024) and lead to errors (e.g., Repetition (Wang et al., 2024a)).

## 5 TEST-TIME INTERVENTION TO CORRECT REASONING HOP GENERALIZATION ERRORS

Although our findings introduced in Section 4 can rectify many token-level erroneous predictions at critical error positions (Section 3) in the whole reasoning process in a **post-hoc** way, we ask the research question RQ3: Can such method (i.e., knocking out individual ep heads) really dynamically correct erroneous predictions in the generation process and finally lead to the improvement of overall reasoning hop generalization performance? To answer this question, we separately investigate its two sub-questions: (i) *how to automatically choose the individual attention heads to intervene (i.e., knock out)?* and (ii) *when to intervene the model?* in the generation process.

**Which attention heads to intervene?** In the inference time, we want to establish the correlation between target ep heads and the input context. To this end, we train a small model to automatically select a target ep head among a candidate head set $\mathbf{H}$ conditioned on the input context. We further model this as a *multi-label classification task with a variable number of positive labels*: **input** is the input context $x_i$, **output** is a label $y_i \in \{0,1\}^{|\mathbf{H}|}$ (for $k \in [0..|\mathbf{H}| - 1]$, if knocking out $\mathbf{H}[k]$ can correct the prediction $x_i$, then $y_i[k] = 1$ otherwise $y_i[k] = 0$) and $\mathcal{D} = \{(x_i, y_i)\}_{i=1}^{|\mathcal{D}|}$ is the training set. As for the training objective, among BCE loss, BPR loss (Rendle et al., 2012), ZLPR loss (Su

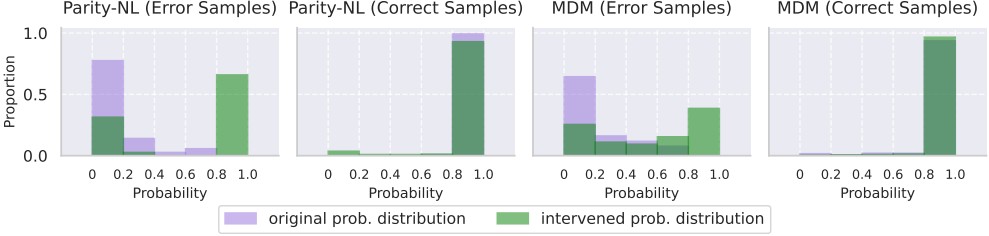

Figure 7: Probability distribution of ground-truth answers before and after model intervention.

Table 1: Hit@1 in the in-distribution ("**i.d.**") and out-of-distribution ("**o.o.d.**") evaluation of the trained classifiers. "***Random Guessing***": the baseline of randomly selecting a candidate head.

| Model | Qwen2.5-7B-Instruct | | Phi-3-Instruct | | LLaMA3-8B-Instruct | | Qwen3-8B-Instruct | |
| Testing Type | i.d. | o.o.d. | i.d. | o.o.d. | i.d. | o.o.d. | i.d. | o.o.d. |
| --- | --- | --- | --- | --- | --- | --- | --- | --- |
| *Random Guessing* | 42.3% | 32.2% | 20.8% | 25.8% | 23.6% | 27.0% | 23.0% | 19.1% |
| *Trained Classifier* | **79.6**% | **53.4**% | **75.2**% | **58.2**% | **80.8**% | **35.5**% | **87.2**% | **82.2**% |

et al., 2022) and multi-label Softmax loss (Pu et al., 2025), we find that the last one consistently achieve the best generalization performance. We fine-tune the pre-trained Qwen2.5-0.5B model with a randomly initialized classification head (denoted as $f_\theta(\cdot)$) with LoRA (Hu et al., 2022) for this task. Our training objective can be formalized as $\min_\theta \mathbb{E}_{(x,y)\in\mathcal{D}}\big[-\log\big(\frac{\exp(f_\theta(x)\cdot y)}{\exp(f_\theta(x))\cdot \mathbf{1}_{|\mathbf{H}|}}\big)\big]$.

Motivated by our earlier observation that *diverse tasks and error types often map onto a shared subset of ep heads*, we maintain a compact candidate set $\mathbf{H}$ per model. Following Section 4, we locate $\mathcal{H}_{\text{ep}}$ across five representative hop-generalization tasks (Parity-NL, MDM, LLC, CLF, MOAS), and select a subset such that (i) each head is implicated in multiple error types, and (ii) the set collectively covers all key error types (see Appendix G.1). This yields $|\mathbf{H}| = 8$ for Qwen2.5-7B, 9 for Phi-3 and Qwen3-8B, and 10 for LLaMA3-8B. We use erroneous predictions from Parity-NL, MDM, LLC, CLF, and MOAS for training and in-distribution testing, and from ObjC and NumS for out-of-distribution testing. More details can be found in Appendix G.2. To evaluate the head selector, we report Hit@1 accuracy (sampling one head from the softmax-normalized logits and checking whether it belongs to the positive class) which directly reflects its intended use. As shown in Table 1, *the trained classifiers substantially outperform random baselines on in-distribution tasks and also generalize well to out-of-distribution tasks*. This suggests that erroneous predictions from diverse tasks and error types can often be corrected by intervening the same head, indicating a shared internal error mechanism.

**When to intervene the model?** During decoding, we adopt a simple uncertainty-based detector to decide when to intervene. Specifically, we monitor the predictive entropy of each generated token, leveraging our earlier finding (Section 4 and Appendix F.1) that errors at key positions exhibit systematically higher entropy than correct predictions. A fixed threshold $\tau$ is applied: if a token's entropy exceeds $\tau$, the trained classifier selects one head from $\mathbf{H}$ to knock out; otherwise, decoding proceeds normally. Our detection design is intentionally simple, as detection is orthogonal to correction and has been extensively studied in the hallucination-detection literature (Kadavath et al., 2022; Kirchhof et al., 2025; Zhang et al., 2025c; Fu et al., 2025; Obeso et al., 2025). To isolate the effect of our correction method, we additionally evaluate a **gold-detector** setting where an oracle provides exact error locations (Section 3), revealing the full rectification potential of TCR.

**Dynamically Correct Reasoning Hop Generalization Errors** Based on above explorations, we propose **T**est-time **C**orrection of **R**easoning (**TCR**), a lightweight test-time intervention method to automatically rectify reasoning hop generalization errors, which incorporate (i) a fixed entropy threshold to detect token-level erroneous predictions and (ii) our trained classifier to identify the heads in $\mathbf{H}$ to be knocked out. Each time we trigger TCR, we select the top3 heads predicted by the classifier, separately knock out these heads and use majority-vote to determine the final correction result. We also include the variant of TCR with the gold detector, **TCR-gold**, to fully unleash the rectifying potential of TCR. Table 2 presents the average accuracy (over three random seeds) of TCR and TCR-gold on seven reasoning hop generalization tasks and four LLMs, compared against original model outputs and DoLa (Chuang et al., 2024), a decoding-based hallucination mitigation method. Each task uses 100 test instances, with a small number of invalid generations (e.g., direct answers without reasoning) excluded. (i) *TCR consistently improves over baseline generation.* For instance, it achieves large gains on Parity-NL with Qwen2.5 (+12.1%) and LLaMA3 (+12.0%), on MOAS with LLaMA3 (+16.5%), and on ObjC with Phi-3 (+14.1%). On average, TCR improves accuracy by $5\% \sim 7\%$ across tasks and models. TCR-gold, with oracle error localization, demonstrates the method's potential: for Qwen2.5, accuracy improves nearly $20\%$ ($41.7\% \rightarrow 61.3\%$). In contrast, DoLa yields only marginal or negative effects in these reasoning settings. (ii) Qwen3 exhibits strong performance on certain tasks (e.g., Parity-NL and NumS above $98\%$), leaving little room for further improvement (likely due to reasoning-oriented reinforcement learning (Chu et al., 2025)). However, in more challenging tasks, such as MDM, TCR-gold improves accuracy by $22.4\%$,

Table 2: The performance of TCR and TCR-gold on seven tasks with four LLMs.

| Method | Parity-NL | MDM | LLC | CLF | MOAS | ObjC | NumS | Average |
|---|---|---|---|---|---|---|---|---|
| *Qwen2.5-7B-Instruct* | 48.3% | 43.0% | 11.7% | 56.8% | 39.2% | 52.0% | 41.1% | 41.7% |
| +DoLa(Chuang et al., 2024) | 58.1% | 38.5% | 8.0% | 52.3% | 40.0% | 52.3% | 48.7% | 42.6% |
| +TCR (ours) | 60.4% | 48.2% | 16.2% | 66.6% | 46.0% | 56.0% | 46.0% | 48.5%(+6.8%) |
| +TCR-gold (ours) | 81.2% | 58.3% | 23.0% | 71.3% | 62.0% | 76.0% | 54.5% | 61.3%(+19.6%) |
| *Phi-3-Instruct* | 51.2% | 8.1% | 53.1% | 9.2% | 1.0% | 30.9% | 72.0% | 32.2% |
| +DoLa(Chuang et al., 2024) | 25.8% | 6.7% | 29.5% | 7.6% | 1.0% | 29.0% | 72.0% | 24.5% |
| +TCR (ours) | 55.8% | 11.5% | 56.1% | 11.2% | 4.0% | 45.0% | 76.0% | 37.1%(+4.9%) |
| +TCR-gold (ours) | 65.2% | 15.8% | 59.6% | 20.3% | 4.0% | 45.0% | 82.0% | 41.7%(+9.5%) |
| *LLaMA3-8B-Instruct* | 70.0% | 0.0% | 81.0% | 15.2% | 22.9% | 68.8% | 4.5% | 37.5% |
| +DoLa(Chuang et al., 2024) | 71.6% | 0.0% | 71.2% | 8.8% | 29.8% | 71.2% | 6.9% | 37.1% |
| +TCR (ours) | 82.0% | 0.0% | 82.3% | 28.2% | 39.4% | 67.8% | 7.8% | 43.9%(+6.4%) |
| +TCR-gold (ours) | 88.0% | 0.0% | 90.7% | 32.7% | 47.0% | 76.4% | 10.1% | 49.3%(+11.8%) |
| *Qwen3-8B-Instruct* | 98.7% | 24.7% | 67.5% | 96.9% | 92.7% | 85.0% | 98.0% | 80.5% |
| +DoLa(Chuang et al., 2024) | 100.0% | 9.0% | 57.0% | 94.4% | 89.7% | 82.7% | 98.0% | 75.8% |
| +TCR (ours) | 100.0% | 29.3% | 68.0% | 97.0% | 93.6% | 88.1% | 99.0% | 82.1%(+1.6%) |
| +TCR-gold (ours) | 100.0% | 47.1% | 71.0% | 99.0% | 95.8% | 90.0% | 100.0% | 86.1%(+5.6%) |

and both TCR and TCR-gold deliver consistent gains across other tasks. These results underscore the robustness and generality of TCR, even for stronger models.

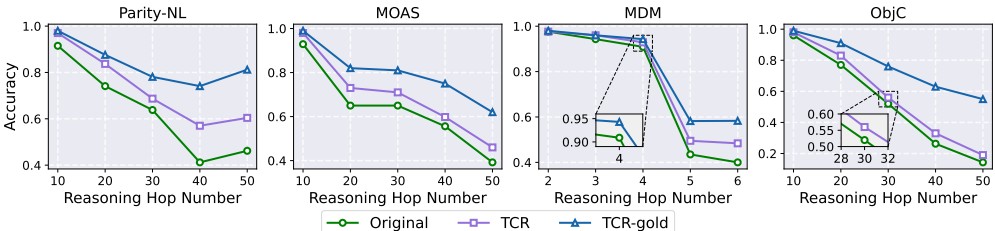

Figure 8: Performance of TCR and TCR-gold changing with reasoning hop number.

**Analyzing the Peformance of TCR** We further conduct analytical experiments on Qwen2.5-7B-Instruct to better understand the behavior of TCR. Figure 8 shows that both TCR and TCR-gold *consistently improve reasoning hop generalization across Parity-NL, MOAS, MDM, and ObjC tasks of varying hops*. Besides, a natural baseline is **Removing-and-Resampling (RR)**, which removes the logit of the model's original prediction from the final hidden state and re-samples the output. As shown in Figure 16(a), RR yields only marginal gains on Parity-NL but substantially degrades performance on MDM and CLF, highlighting the superiority of TCR. Finally, to assess the role of the trained head selector, we compare it with a random head selector. Figure 16(b) shows that both TCR and TCR-gold with the trained selector consistently outperform their random-selector counterparts on Parity-NL, MDM, and CLF, confirming the effectiveness of our design.

## 6 CONCLUSION

In this work, we systematically studied the mechanisms of reasoning hop generalization failures in LLMs. We found that these failures concentrate on token positions of a few key error types rather than being uniformly distributed across reasoning chains. Probing model internals at these positions further revealed a competition mechanism: correct and erroneous reasoning trajectories co-exist, while the erroneous ones can suppress the correct ones, eventually leading to errors. Based on these insights, we proposed TCR, a lightweight intervention method that dynamically identifies and deactivates these heads at test-time and demonstrate its effectiveness through extensive experiments.

## ACKNOWLEDGEMENT

This work was supported in part by grants from the National Natural Science Foundation of China (No. U24A20253), Scientific Research Innovation Capability Support Project for Young Faculty, the Research Grants Council of the Hong Kong SAR under Grant GRF 11217823, 11216225, and Collaborative Research Fund C1042-23GF, the National Natural Science Foundation of China under Grant 62371411, InnoHK initiative, the Government of the HKSAR, Laboratory for AI-Powered Financial Technologies.

## ETHICS STATEMENT

This work conducts mechanistic analyses of large language models using only publicly available open-source models and widely adopted reasoning benchmarks. No human subjects, private data, or personally identifiable information are involved. Our interventions (e.g., test-time correction) are designed purely for academic understanding and evaluation, without deployment in sensitive applications. We acknowledge that improving reasoning capabilities of LLMs may carry dual-use concerns, but our study focuses on fundamental error mechanisms and lightweight interventions, aiming to advance transparency and reliability in LLM reasoning research.

## REPRODUCIBILITY STATEMENT

We have taken several steps to ensure reproducibility of our results. All models used in this work are publicly available open-source LLMs, and the datasets and benchmarks are well-documented in the appendix. Details of our experimental setup, training configurations, and evaluation protocols are provided in the main text and Appendix.

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

# APPENDIX CONTENTS (WITH MAIN TEXT)

## A    THE USE OF LARGE LANGUAGE MODELS

Large Language Models (LLMs) have become valuable tools in the process of academic writing. In our work, we primarily utilize LLMs for polishing the manuscript: they help identify and correct grammatical errors, improve overall readability, and simplify verbose or cumbersome expressions. This use allows us to refine the clarity and fluency of the text efficiently, ensuring that the ideas are communicated more effectively. In addition, we leverage LLMs for secondary tasks related to manuscript preparation, such as providing guidance on LaTeX syntax and functionalities required for paper writing, and assisting with the usage of Python tools like Matplotlib for making figures. These applications facilitate the technical aspects of manuscript preparation, reducing the time and effort needed to implement formatting and visualization requirements. The use of LLMs did not infuence the research process, methodology, or the originality of the results of this work.

## B    LIMITATIONS AND DISCUSSION

**Model size is limited.**    Due to computational resource constraints, experiments were conducted on a limited set of models whose parameter scales are within 10B.

**This work mainly focus on the most classical CoT setting.**    Besides, our analysis primarily focused on the most fundamental form of reasoning, the classical CoT. While more advanced long-CoT paradigms (Yeo et al., 2025) with reflection or backtracking remain unexplored, we view our work as laying the groundwork for mechanistic studies of such settings (Wang et al., 2025). We also observed that Qwen3-8B substantially outperforms previous models across multiple tasks, suggesting the presence of architectural or training differences (e.g., the reasoning-orientated reinforcement learning (DeepSeek, 2025; Chu et al., 2025)) that merit deeper mechanistic investigation (e.g., the research question: whether reinforcement learning really helps with out-of-distribution generalization of LLMs? (Sun et al., 2025; Shojaee et al., 2025)) in the future work.

**The use of Logit Lens.**    Although being widely used (Geva et al., 2023; Wang et al., 2023; Li et al., 2024; Chughtai et al., 2024), Logit Lens (nostalgebraist, 2020) introduces potential limitations due to semantic-space misalignment (Belrose et al., 2025; Ghandeharioun et al., 2024) between intermediate hidden states and the final unembedding matrix. In future work, we plan to explore Tuned Lens (Belrose et al., 2025) and Patchscopes Ghandeharioun et al. (2024), which are designed to reduce space-misalignment issues when interpreting intermediate representations.

**Current in-distribution tasks are limited.**    Although the TCR show non-trivial performance on out-of-distribution tasks, currently the in-distribution tasks are still quite limited, which may constraint the TCR's cross-task generalization performance. Expanding the in-distribution task set to include more diverse task types is a straightforward way to further enhance cross-task generalization of TCR.

## C    RELATED WORK

**Chain-of-Thought Reasoning's Deficiency in High-Complexity Problems** Chain-of-Thought (CoT) (Wei et al., 2022), which enables LLMs to automatically decompose complex questions into elementary ones and solve them step by step, now becomes a standard paradigm of LLMs' reasoning (OpenAI, 2024; DeepSeek, 2025; Alibaba, 2025b). However, recent studies Dziri et al. (2023); Matarazzo & Torlone (2025); Baeumel et al. (2025); Xiao & Liu (2025) have shown the limitations of CoT reasoning in solving problems that require a large number of reasoning hops. Such observations reveal the inherent brittleness and superficiality of current CoT reasoning capabilities (Zhao et al., 2025). Shojaee et al. (2025), Hu et al. (2025) and Sun et al. (2025) experimentally demonstrate that even the latest large reasoning models (such as DeepSeek-R1 (DeepSeek, 2025), Claude 3.7 Sonnet (Anthropic, 2025) and OpenAI o4-mini (OpenAI, 2025)) also encounter such problems, proving that such issues stem from the inherent characteristics of LLMs. Specifically, researchers observe that the performance of CoT reasoning of modern LLMs drops drastically as the complexity of the input problem increases (e.g., the digit number of the operands increases in the multiplication arithmetic problem (Dziri et al., 2023) and the list size increases in the 0-1 parity problem (Anil

et al., 2022; Zhou et al., 2024).), which starkly contrasts to human-level robust reasoning (Collins et al., 2022; Bao et al., 2024; Zheng et al., 2025; de Wynter, 2025): once we acquire the skill of solving a simple problem, we can easily generalize it to the more complex ones (Li et al., 2023; Kim & Linzen, 2020; Lake & Baroni, 2023; Li et al., 2025). To better understand such difference, it is urging to study the underlying mechanism behind the CoT reasoning in black-box LLMs.

**Chain-of-Thought Reasoning Mechanism Analysis** Prior studies on CoT reasoning largely fall into two directions. The first line of work investigates *how* LLMs internally perform step-by-step reasoning. For example, Dutta et al. (2024) identify specific attention heads in LLaMA2-7B-Chat that correlate with each reasoning step, while a series of works (Wang et al., 2024b; Yao et al., 2025; Ye et al., 2024; Zhang et al., 2025b) train small GPT-2 like transformer models on synthetic CoT data and reveal mechanisms such as task-state encoding, buffer mechanisms, and finite-state-automaton-like circuits. The second line of work investigates *where* CoT succeeds or fails. For instance, Dziri et al. (2023) attribute reasoning errors to the accumulation of local prediction errors, while Zhao et al. (2025) show that when the reasoning hop length exceeds the training distribution, model performance deteriorates sharply. However, few works establish a bridge between these two perspectives: connecting the internal mechanisms of CoT reasoning with the conditions under which reasoning fails. Yan et al. (2025) represent a step in this direction, yet their analysis focuses on a *static* source of failure, showing that models may over-attend to demonstration tokens and thus generate spurious reasoning. In this work We study a more *dynamic* phenomenon: how increasing the reasoning hops triggers systematic competition among model's internal mechanisms, leading to reasoning collapse.

## D   TASKS, DATA AND MODELS

In this section, we mainly provide the details (e.g., the detailed description and specific examples of these tasks, the model generation configurations and so on) of seven reasoning hop generalization tasks (Parity-NL, LLC, MDM, MOAS, CLF, NumS and ObjC) and four open-sourced LLMs (Qwen2.5-7B-Instruct, Phi-3-Instruct, LLaMA3-8B-Instruct, Qwen3-8B-Instruct).

### D.1   TASKS AND DATA

Following the task forms widely used in the relevant literature, we collect seven reasoning hop generalization tasks from three different domains:

- *Symbolic Reasoning*: **Parity-NL** (Parity problem, natural language variant) (Anil et al., 2022; Zhou et al., 2024) and **LLC** (Last Letter Concatenation) (Wei et al., 2022; Hu et al., 2025),
- *Mathematical Calculation*: **MDM** (Multi-Digit Multiplication) (Dziri et al., 2023) and **MOAS** (Multi-Operand Addition and Substract) (Baeumel et al., 2025),
- *Coding*: **CLF** (LeetCode 1598) and **NumS** (LeetCode 1450),
- *Others*: **ObjC** (Object Counting) that requires both symbolic reasoning and mathematical calculation abilities from the BBH benchmark (Suzgun et al., 2022).

Inspired by the practice in Mirzadeh et al. (2025), We resynthesize data for each task by randomly substituting (human and object) names and numerical values to ensure that there is as less data leakage in the LLMs' training process as possible. With these tasks, the good thing is that one can arbitrarily control the problem complexity (e.g., for the Pairty-NL task, we instantiate problems with the reasoning hop number (action number) $n \in [10, 20, 30, 40, 50]$) while maintaining the underlying reasoning skill, which provides us with an ideal testbed for us to study LLMs' performance, error patterns and their internal mechanisms in the reasoning hop generalization scenarios. Below we introduce these tasks one-by-one in detail.

**Parity-NL**   : The parity task requires the model to predict whether the sum of the input 0-1 list is even or odd. For example, the parity of the input list $[0, 1, 1, 0, 1, 0, 0, 1, 1]$ is "odd" (represented by 1, the binary sum) as opposed to "even" (represented by 0), because there is an odd number of 1s in the input list. In our practice, we use the natural language variant of the parity problem (Wei et al., 2022; Anil et al., 2022; Zhou et al., 2024; Hu et al., 2025), the "coin-flip" problem, to make the task more challenging. We use the state of a coin to represent the partial binary sum of the input list (i.e., heads up represents 0 and tails up represents 1). Each element in the input list is represented by a filp action (e.g., "Then Jackson flips the coin" represents a input 1; "Then Ava does

not flip the coin" represents a input 0). We show an example of Parity-NL in Figure 26. One of the good nature of Parity-NL is that it is easily extensible. We can insert arbitrary number of flip actions in the problem to control its complexity, because the model needs as many reasoning hops as the flip actions to solve the problem. Specifically we make five splits in the Parity-NL dataset (hop number$\in [10, 20, 30, 40, 50]$).

**LLC** : The last letter concatenation task (Wei et al., 2022; Hu et al., 2025) requires models to extract the last character of each word in a given list and concatenate them in order, which evaluates the ability to manipulate symbolic strings based on positional rules. We illustrate an example of LLC and its corresponding solution in Figure 28. LLC is highly extensible: we can flexibly control both the length of the words and the size of the list. In our datasets, we set the list size$\in \{2, 4, 6, 8, 10\}$, resulting in five splits.

**MDM** : The multi-digit multiplication task is to ask models to calculate the product of two multi-digit numbers, which requires executing operations with numerical symbols based on a step-by-step reasoning (Dziri et al., 2023). We show an example of MDM and the corresponding solution in Figure 27. MDM is also easily extensible. We can control the digit number of the operands. In our datasets, we fix the digit number of the first operand at 3 and vary the digit number of the second operand from 2 to 6, resulting in five splits.

**MOAS** : The multi-operand addition and subtraction task (Baeumel et al., 2025) requires models to compute the sum or difference of multiple integers arranged in sequence, which tests the ability to correctly execute chained arithmetic operations with numerical symbols. We illustrate an example of MOAS and its corresponding solution in Figure 29. MOAS is highly extensible: we can flexibly control both the number of operands and their digit lengths. In our datasets, we set each operand$\in [1..99]$ and vary the number of operands$\in [10, 20, 30, 40, 50]$, resulting in five splits.

**CLF** : The crawler-log-folder task[4] (LeetCode Problem No. 1598) requires models to use the given Python code to simulate the process of navigating through a file system based on a sequence of folder operations. Each log entry corresponds to one of the following actions: moving into a subfolder (e.g., "x/"), moving up to the parent folder ("../"), or staying in the current folder ("./"). The goal is to determine the depth of the current folder relative to the root after executing all operations. We illustrate an example of CLF and its corresponding solution in Figure 30. CLF is highly extensible: we can flexibly control the length of the operation sequence. In our datasets, we set the sequence length$\in \{10, 20, 30\}$, resulting in three splits.

**NumS** : The number-of-student-doing-homework at a given time task[5] (LeetCode Problem No. 1450) requires models to follow a given Python code to determine how many students are doing homework at a specific query time, given the start and end times of multiple students' homework sessions. We illustrate an example of NumS and its corresponding solution in Figure 32. NumS is also easily extensible: we can flexibly control the number of students (i.e., intervals) in the input. In our datasets, we set the number of students$\in \{5, 10, 15, 20\}$, resulting in four splits.

**ObjC** : The object counting task is from the BIG-Bench Hard (BBH) benchmark (Suzgun et al., 2022), where models are asked to count the number of objects satisfying certain conditions in a given natural language description. The task evaluates the model's ability to perform symbolic reasoning over textual inputs, combining both symbolic reasoning and mathematical calculation. We illustrate an example of object counting and its corresponding solution in Figure 31. Object counting naturally scales with the complexity of the description: we can flexibly control the number of objects as well as the conditions imposed. In our datasets, we set the number of objects$\in \{10, 20, 30, 40, 50\}$, resulting in five splits.

**Solution Template** All tasks used in this work have deterministic algorithms that guide the concrete solutions of hop-by-hop reasoning. Note that here we adopt some "standard" solution templates (as shown in the figures of these task examples) to guide the models' reasoning on each task

---

[4] https://leetcode.com/problems/crawler-log-folder/
[5] https://leetcode.com/problems/number-of-students-doing-homework-at-a-given-time/description/

with in-context demonstrations Dutta et al. (2024); Hu et al. (2025), for better analyzing models' error patterns. These "standard" solution templates are derived from model's most frequent responses (i.e., models themselves tend to solve the problems with these internalized templates).

## D.2 Models

**Qwen2.5-7B-Instruct** : Qwen2.5[6] is a series of large language models developed by Alibaba (Alibaba, 2025a). The 7B-Instruct variant is instruction-tuned to better follow user prompts, especially in English and Chinese. It is optimized for reasoning, dialogue, and task-oriented scenarios, making it a strong baseline for mid-sized LLMs.

**Phi-3-mini-4k-Instruct** : Phi-3 is a family of lightweight but high-quality models developed by Microsoft. The mini-4k-Instruct variant, with 3.8B parameters and a 4k context window[7], is instruction-tuned to follow natural language instructions. Despite its small size, it achieves competitive performance on reasoning and coding benchmarks (Microsoft, 2024).

**LLaMA3-8B-Instruct** : LLaMA3 (Meta, 2024) is a family of foundation models released by Meta[8]. The 8B-Instruct model is tuned to improve instruction following and reasoning capabilities while maintaining efficiency. It represents a widely adopted open-source baseline, balancing model capacity and accessibility for research and applications.

**Qwen3-8B-Instruct** : Qwen3[9] (Alibaba, 2025c) is the latest generation of the Qwen series, continuing to improve performance across reasoning, code generation, and multilingual understanding. The 8B-Instruct variant is instruction-tuned for better alignment with human instructions and achieves strong results in both academic and practical benchmarks, making it a competitive model among open-source LLMs. Note that in this work we study the classic CoT (Wei et al., 2022) mode of Qwen3, rather than its long CoT (Yeo et al., 2025; DeepSeek, 2025) mode.

As suggested by the official model card on the huggingface, we adopt the following generation configurations (Listing 1) in our experiments.

Listing 1: Generation Configs for Large Language Models

```
{
  "bos_token_id": 151643, # for Qwen2.5-7B-Instruct and Qwen3-8B-Instruct
  #"bos_token_id": 1, # for Phi-3-Instruct; "bos_token_id": 128000, # for
      LLaMA3-8B-Instruct

  "pad_token_id": 151643, # for Qwen2.5-7B-Instruct and Qwen3-8B-Instruct
  # "pad_token_id": 32000, # for Phi-3-Instruct

  "do_sample": true,

  "eos_token_id": [
    151645, 151643, # for Qwen2.5-7B-Instruct and Qwen3-8B-Instruct
    # 32000,32001,32007, # for Phi-3-Instruct
    # 128001, 128009, # for LLaMA3-8B-Instruct
  ],

  "repetition_penalty": 1.05, # for Qwen2.5-7B-Instruct

  "temperature": 0.7, # for Qwen2.5-7B-Instruct
  # "temperature": 0.6, # for Phi-3-Instruct, LLaMA3-8B-Instruct and
      Qwen3-8B-Instruct

  "top_p": 0.8, # for Qwen2.5-7B-Instruct
  # "top_p": 0.9, # for Phi-3-Instruct, LLaMA3-8B-Instruct
```

---

[6] https://huggingface.co/Qwen/Qwen2.5-7B-Instruct
[7] https://huggingface.co/microsoft/Phi-3-mini-4k-instruct
[8] https://huggingface.co/meta-llama/Meta-Llama-3-8B-Instruct
[9] https://huggingface.co/Qwen/Qwen3-8B

```
 # "top_p": 0.95, # for Qwen3-8B-Instruct

 "top_k": 20, # for Qwen2.5-7B-Instruct and Qwen3-8B-Instruct

 "transformers_version": "4.37.0" # for Qwen2.5-7B-Instruct
 # "transformers_version": "4.39.3" # for Phi-3-Instruct # "4.40.0.dev0"
     # for LLaMA-3-8B-Instruct
 # "transformers_version": "4.51.0" # for Qwen3-8B-Instruct
}

messages = [
         {"role": "system", "content": "You are Qwen, created by Alibaba
             Cloud. You are a helpful assistant."}, # for
             Qwen2.5-7B-Instruct
         # {"role": "system", "content": "You are a helpful AI
             assistant."}, # for Phi-3-Instruct
         # {"role": "system", "content": "Be a helpful assistant."}, #
             for LLaMA-3-8B-Instruct
         {"role": "user", "content": prompt}, # for all models
     ]

text = tokenizer.apply_chat_template(
         messages,
         tokenize=False,
         add_generation_prompt=True
     )

# only for LLaMA3-8B-Instruct
# terminators = [tokenizer.eos_token_id,
    tokenizer.convert_tokens_to_ids("<|eot_id|>")]
```

# E  ADDITIONAL INFORMATION ABOUT LINKING OVERALL REASONING HOP GENERALIZATION PERFORMANCE DECLINE TO SPECIFIC ERROR TYPES

## E.1  ERROR TYPES OF EACH TASK

We provide a detailed description for all possible error types of each task here: Parity-NL (Figure 9), MDM (Figure 10), LLC (Figure 33), CLF (Figure 35), MOAS (Figure 34), ObjC (Figure 36), and NumS (Figure 37). In summary, We conclude 8 types of possible errors for the Parity-NL task, 9 types of possible errors for the MDM task, 5 types of possible errors for the LLC task, 7 types of possible errors for the MOAS task, 5 types of possible errors for the CLF task, 6 types of possible errors for the ObjC task, and 8 types of possible errors for the NumS task. We summarize all possible error types of each task in Table 3.

## E.2  IDENTIFYING KEY ERROR TYPES FOR EACH TASK

In this section, we present the identified key error types for each task. Firstly, we show the **error proportion** (number of error samples divided by the number of all samples) for these key types and all other types and the **overall accuracy** with different reasoning hop numbers for **both Qwen2.5-7B-Instruct and Phi-3-Instruct** on Parity-NL, MDM and ObjC tasks in Figure 11. Then we provide the detailed statistical information (**overall accuracy**, **error ratio** and **error type**) about the key error types of four LLMs on seven reasoning hop generalization tasks in Table 4.

The results further confirm our observation in the main text Section 3 with different models. First, both models experience a pronounced drop in overall accuracy as the number of reasoning hops increases, confirming the fragility of current LLMs a problem involving more hops of reasoning. More importantly, this decline is not explained by a uniform increase in all error types, but is instead largely driven by the growth of a small set of target error types we identify. In contrast, the aggregate proportion of all other error types fluctuates only mildly with reasoning hop number. This highlights that *performance degradation in reasoning hop generalization stems primarily from a limited set*

---

**Possible Error Types in a 50-hop Parity-NL Example**

**Input Problem:**
Initially The coin is heads up. (1) Then Matthew doesn't flip. (2) Then Ethan flips. **......** (*Omitting 46 events*) (49) Then Jack doesn't flips. (50) Then Samuel flips. Finally, is the coin heads up or tails up?
=============================================================
**Model Response:**
Let's analyze the sequence of events step-by-step to determine the final state of the coin.
**Initial Step.** The coin starts heads up **[Type 1: Initial State Recall Error, heads up → tails up]**.
**Step 1.** Matthew **[Type 2: Name Recall Error, Matthew → Alexander]** doesn't flip the coin. (Coin remains heads up.)
**Step 2.** Ethan flips **[Type 3: Action Recall Error, flips → doesn't flip]** the coin. (Coin becomes tails up.)
- - - - - - - - - -
**. . . . . .** (*Omitting the intermediate 46 reasoning steps*)
- - - - - - - - - -
**Step 49.** Jack doesn't flips. (Coin remains heads up **[Type 4: State Recall Error, heads up → tails up]**.)
**Step 50.** Samuel flips the coin. (Coin becomes tails up **[Type 5: State Update Error, tails up → heads up]**.)
**[Type 6 / 7: Reasoning Less / More Hops, Reasoning Hop Number ≠ 50.]**
After going through each step, we see that the coin ends up tails up **[Type 8: Final State Copy Error, tails up → heads up]**.
Therefore, the coin is tails up.

Figure 9: Possible error types in a Parity-NL problem. The underline green parts highlight the positions where errors of different types may occur. The **bold blue parts** indicate the error types. For each error type, we only mark one instance.

*of key error types that intensify with hop length*, underscoring the importance of focusing on the corresponding token positions where such errors occur.

Table 3: Summary of tasks and corresponding possible error types.

| Task | Possible Error Types |
|---|---|
| **Parity-NL** | (1) Initial State Recall Error, (2) Name Recall Error, (3) Action Recall Error, (4) State Recall Error, (5) State Update Error, (6) Reasoning Less Hops, (7) Reasoning More Hops, (8) Final State Copy Error. |
| **MDM** | (1) Operand Recall Error, (2) Digit Error, (3) Digit Copy Error, (4) Local Calculation Error, (5) Shift Error, (6) Reasoning Less Hops, (7) Reasoning More Hops, (8) Local Result Copy Error, (9) Calculation Errors. |
| **LLC** | (1) Word Recall Error, (2) Letter Error, (3) Concatenation Error, (4) Reasoning Less Hops, (5) Reasoning More Hops. |
| **MOAS** | (1) Operand Recall Error, (2) Operation Recall Error, (3) Local Result Copy Error, (4) Operand Interpretation Error, (5) Local Calculation Error, (6) Reasoning Less Hops, (7) Reasoning More Hops. |
| **CLF** | (1) Operation Recall Error, (2) Depth Copy Error, (3) Local Calculation Error, (4) Reasoning Less Hops, (5) Reasoning More Hops, (6) Final Depth Copy Error. |
| **ObjC** | (1) Name Recall Error, (2) Object Information Recall Error, (3) Local Result Copy Error, (4) Local Calculation Error, (5) Reasoning Less Hops, (6) Reasoning More Hops. |
| **NumS** | (1) Time Recall Error, (2) Time Copy Error, (3) Local Result Copy Error, (4) Local Calculation Error, (5) Condition Judge Error, (6) Calculation Logic Error, (7) Reasoning Less Hops, (8) Reasoning More Hops. |

---

**Possible Error Types in a MDM Example**

**Problem:**
326 * 3589 =? please think step-by-step.
**Answer:**
Let's break down the multiplication of 326 and 3589 step-by-step. We will multiply 326 by each digit of 3589 and then add the results.
**Step 1: Multiply 326 [Type 1: Operand Recall Error, 326 → another number] by 9 (units place of 3589)**
[ 326 * 9 = 2934 ]
(Shift this result 0 places to the left: [ 2934 ])
**Step 2: Multiply 326 by 8 [Type 2: Digit Error, 8 → 5 (e.g., skip the digit 8)] (tens place of 3589)**
[ 326 * 8 [Type 3: Digit Copy Error, 8 → another number] = 2608 ]
(Shift this result 1 place to the left: [ 26080 ] )
**Step 3: Multiply 326 by 5 (hundreds place of 3589)**
[ 326 * 5 = 1630 [Type 4: Local Calculation Error, 1630 → 1580 ] ]
(Shift this result 2 places to the left: [ 163000 ] [Type 5: Shift Error, [ 163000 ] → [ 16300 ] ])
**Step 4: Multiply 326 by 3 (thousands place of 3589)**
[ 326 * 3 = 978 ]
(Shift this result 3 places to the left: [ 978000 ])
**[Type 6 / 7: Reasoning Less / More Hops]**
**Step 5: Add all the results together**
[ 2934 + 26080 + 163000 + 978000 ]
**Step 6: Perform the addition step-by-step**
[ 2934 [Type 8: Local Result Copy Errors, 2934 → 2984] + 26080 = 29014 ]
[ 29014 + 163000 = 192014 [Type 9: Calculation Errors, 192014 → 182014] ]
[ 192014 + 978000 = 1170014 ]
**Final Result**
[ 326 * 3589 = 1170014 ]

Figure 10: Possible error types in a MDM problem. The underline green parts highlight the positions where errors of different types may occur. The **bold blue parts** indicate the error types. For each error type, we only mark one instance.

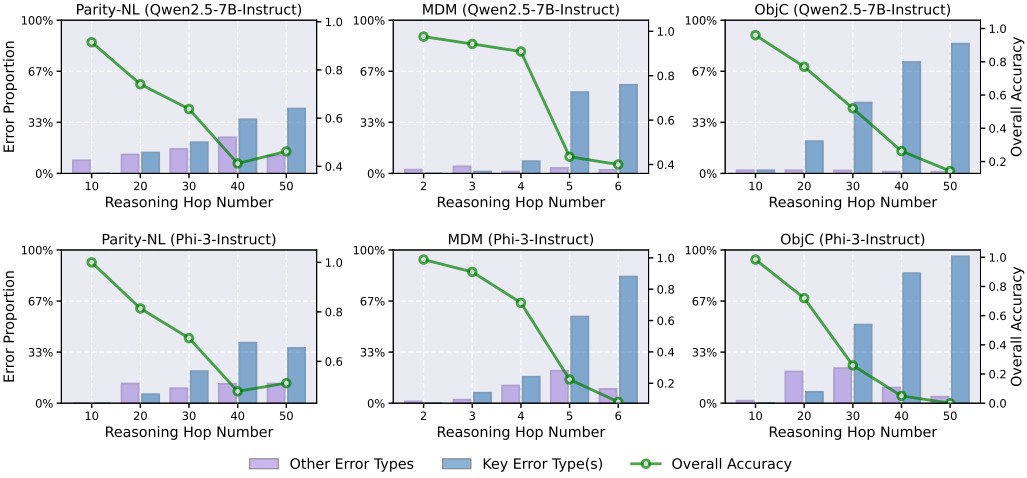

Figure 11: Overall accuracy (green curve) and error proportion for key error types (blue bar) and other error types (purple bar) of Qwen2.5-7B-Instruct and Phi-3-Instruct on Parity-NL, MDM and ObjC, as a function of reasoning hop number. Performance exhibits a sharp decline as reasoning hop number increases, which is largely driven by the surge of a few key target error types.

# F  ADDITIONAL INFORMATION ABOUT PROBING INTERNAL MECHANISMS OF KEY ERROR TYPES IN REASONING HOP GENERALIZATION

## F.1  KEY REASONING ERRORS ACCOMPANIED BY TOKEN-LEVEL UNCERTAINTY LARGER UNCERTAINTY

We comparatively analyze the entropy uncertainty (Fadeeva et al., 2024; Zhang et al., 2025a) associated with token predictions at critical positions of those key error types. The token predictive

Table 4: Statistical information about the key error types of four LLMs on seven reasoning hop generalization tasks. The number within the parentheses following the task name indicate the reasoning hop number of the task. Key Error Type(s) Ratio and Other Error Types Ratio are calculated by number of errors of target types/number of all errors

| Statistical info | Parity-NL(50) | MDM(6) | LLC(10) | CLF(30) | MOAS(50) | ObjC(30) | NumS(20) |
|---|---|---|---|---|---|---|---|
| *Qwen2.5-7B-Instruct* | | | | | | | |
| *Overall Accuracy* | 46.2% | 40.0% | 2.0% | 53.0% | 41.0% | 52.0% | 1.5% |
| *Key Error Type(s) Ratio* | **78.6**% | **96.3**% | **52.1**% | **56.4**% | **76.3**% | **96.2**% | **100**% |
| *Other Error Types Ratio* | 21.4% | 3.7% | 47.9% | 43.6% | 23.7% | 3.8% | 0.0% |
| *Key Error Type(s)* | **2** | **2, 5** | **3** | **1, 3** | **2** | **1** | **1, 5** |
| *Phi-3-Instruct* | | | | | | | |
| *Overall Accuracy* | 51.2% | 8.1% | 20.7% | 9.2% | 1.0% | 25.9% | 1.0% |
| *Key Error Type(s) Ratio* | **73.8**% | **89.9**% | **67.1**% | **100**% | **90.7**% | **69.1**% | **98.0**% |
| *Other Error Types Ratio* | 26.2% | 10.1% | 32.9% | 0.0% | 9.3% | 30.9% | 2.0% |
| *Key Error Type(s)* | **2** | **2, 5** | **3** | **1, 3** | **2** | **1** | **1, 5** |
| *LLaMA3-8B-Instruct* | | | | | | | |
| *Overall Accuracy* | 71.0% | 0.0% | 59.0% | 14.1% | 25.0% | 80.4% | 0.0% |
| *Key Error Type(s) Ratio* | **72.4**% | **92.9**% | **97.6**% | **93.4**% | **84.0**% | **78.9**% | **98.7**% |
| *Other Error Types Ratio* | 27.6% | 7.1% | 2.4% | 6.6% | 16.0% | 21.1% | 1.3% |
| *Key Error Type(s)* | **8** | **2, 5** | **3** | **3** | **2** | **1** | **5** |
| *Qwen3-8B-Instruct* | | | | | | | |
| *Overall Accuracy* | 98.8% | 25.3% | 53.0% | 98.0% | 92.6% | 89.5% | 98.0% |
| *Key Error Type(s) Ratio* | **100.0**% | **96.6**% | **100.0**% | **100.0**% | **80.0**% | **70**% | **100.0**% |
| *Other Error Types Ratio* | 0.0% | 3.4% | 0.0% | 0.0% | 20.0% | 30.0% | 0.0% |
| *Key Error Type(s)* | **2** | **2, 5** | **3** | **1** | **2** | **1** | **1** |

entropy is calculated as

$$H = -\sum_{t=0}^{|V|-1} \mathbf{p}_\theta(x)[t] \log \mathbf{p}_\theta(x)[t],$$

following the notations in Section 2 ($x$ here refers to the input context before the target token position, i.e., the token that is to be predicted). Table 5 reports average entropies for correct and erroneous predictions across seven reasoning hop generalization datasets (for each dataset, we randomly choose 100 samples). For both Qwen2.5-7B-Instruct, Phi-3-Instruct, LLaMA3-8B-Insturct and Qwen3-8B-Instruct, ***erroneous samples consistently exhibit significantly higher predictive uncertainty, indicating that the model is not decisively committed to erroneous trajectories and leaving room for competing alternatives (i.e., the correct trajectories) in its internal representations.***

This observation motivates the our investigation into the model's inner workings, particularly the ***competition mechanisms*** (Section 4) that underlie reasoning hop generalization errors.

## F.2 DETAILED EXPERIMENT SETTING

For a specific error type, we randomly sample two sets for correction and erroneous predictions at the corresponding key token position, denoted as $S_{\text{corr}}$ and $S_{\text{err}}$, respectively. In our experiments, We $|S_{\text{corr}}| = |S_{\text{err}}| = 10$. We find that the experiment results of locating answer-writing heads and processing heads are not sensitive to the size of $S_{\text{corr}}$ and $S_{\text{err}}$ (5,10 and more samples yield similar locating results), which also demonstrates the robustness of the results and our proposed locating metrics. For the experiment regarding to Figure 7, we knock out the erroneous processing head $\mathbf{a}_0^0$, $\mathbf{a}_{11}^3$ and $\mathbf{a}_{11}^{15}$ for rectifying erroneous predictions of Parity(2), MDM(2) and MDM(5), respectively.

Table 5: Average predictive entropy of correctly and erroneously predicted samples for Qwen2.5-7B-Instruct, Phi-3-Instruct, LLaMA3-8B-Instruct and Qwen3-8B-Instruct across seven datasets, along with the average predictive entropy of the rectified predictions which were originally erroneously predicted. Each value is averaged over 100 samples.

| Model (Case Type) | Parity-NL | MDM | LLC | CLF | MOAS | ObjC | NumS |
|---|---|---|---|---|---|---|---|
| *Qwen2.5-7B-Instruct (Correct)* | 0.000 | 0.000 | 0.035 | 0.000 | 0.000 | 0.000 | 0.006 |
| *Qwen2.5-7B-Instruct (Error)* | 0.097 | 0.200 | 0.230 | 0.241 | 0.178 | 0.192 | 0.073 |
| *Qwen2.5-7B-Instruct (Rectified)* | 0.022 | 0.082 | 0.133 | 0.098 | 0.038 | 0.050 | 0.035 |
| *Phi-3-Instruct (Correct)* | 0.017 | 0.107 | 0.152 | 0.032 | 0.005 | 0.016 | 0.046 |
| *Phi-3-Instruct (Error)* | 0.512 | 0.429 | 0.694 | 0.392 | 0.348 | 0.385 | 0.412 |
| *Phi-3-Instruct (Rectified)* | 0.085 | 0.288 | 0.505 | 0.130 | 0.052 | 0.109 | 0.191 |
| *LLaMA3-8B-Instruct (Correct)* | 0.009 | 0.015 | 0.002 | 0.0000 | 0.000 | 0.009 | 0.003 |
| *LLaMA3-8B-Instruct (Error)* | 0.258 | 0.195 | 0.139 | 0.201 | 0.252 | 0.266 | 0.214 |
| *LLaMA3-8B-Instruct (Rectified)* | 0.090 | 0.104 | 0.081 | 0.125 | 0.074 | 0.088 | 0.095 |
| *Qwen3-8B-Instruct (Correct)* | 0.021 | 0.000 | 0.000 | 0.008 | 0.009 | 0.000 | 0.002 |
| *Qwen3-8B-Instruct (Error)* | 0.382 | 0.059 | 0.000 | 0.079 | 0.166 | 0.017 | 0.340 |
| *Qwen3-8B-Instruct (Rectified)* | 0.017 | 0.008 | 0.011 | 0.027 | 0.053 | 0.002 | 0.013 |

### F.3 ADDITIONAL RESULTS ABOUT LOCATING AND ANALYZING ANSWER-WRITING HEADS

**Comparing our answer-writing locating methods with previous works** Considering a specific error position, where the predicted token is $t$, previous works (Wang et al., 2023; Dutta et al., 2024; Chughtai et al., 2024) typically measure how much information about $t$ is written by the head $\mathbf{a}_i^l$ (at the **last input token position**) with the Logit Lens probability: $f_{\text{logit-lens}}(\mathbf{a}_i^l)[t]$ (Eq.2) or the directly mapped logit value $(\mathbf{a}_i^l \cdot W_U)[t]$. However, we argue that such approaches that directly inspect the head representations themselves have two main deficiencies. For one thing, they neglects the joint influence of other heads within the same layer as well as the information already present in the residual stream. Specifically, recalling the Transformer residual stream updating formula (Eq.1), we derive that:

$$\mathbf{h}_{\text{mid}}^{l+1} = \mathbf{h}^l + \sum_{i=1}^{H} \mathbf{a}_i^l \Rightarrow \mathbf{h}_{\text{mid}}^{l+1} \cdot W_U = (\mathbf{h}^l + \sum_{i=1}^{H} \mathbf{a}_i^l) \cdot W_U$$

$$\Rightarrow f_{\text{logit-lens}}(\mathbf{h}_{\text{mid}}^{l+1})[t] = \text{Softmax}(\mathbf{h}_{\text{mid}}^{l+1} \cdot W_U)[t] = \text{Softmax}((\mathbf{h}^l + \sum_{i=1}^{H} \mathbf{a}_i^l) \cdot W_U)[t]$$

$$= \text{Softmax}((\mathbf{h}^l + \sum_{i=1,i\neq j}^{H} \mathbf{a}_i^l) \cdot W_U + \mathbf{a}_j^l \cdot W_U)[t]$$

$$\neq \text{Softmax}((\mathbf{h}^l + \sum_{i=1,i\neq j}^{H} \mathbf{a}_i^l) \cdot W_U)[t] + \text{Softmax}(\mathbf{a}_j^l \cdot W_U)[t]$$

$$= f_{\text{logit-lens}}(\mathbf{h}^l + \sum_{i=1,i\neq j}^{H} \mathbf{a}_i^l)[t] + f_{\text{logit-lens}}(\mathbf{a}_j^l)[t]$$

$$\Rightarrow f_{\text{logit-lens}}(\mathbf{h}_{\text{mid}}^{l+1})[t] \neq f_{\text{logit-lens}}(\mathbf{h}^l + \sum_{i=1,i\neq j}^{H} \mathbf{a}_i^l)[t] + f_{\text{logit-lens}}(\mathbf{a}_j^l)[t],$$

which indicates that the direct inspecting results $f_{\text{logit-lens}}(\mathbf{a}_j^l)[t]$ or $(\mathbf{a}_j^l \cdot W_U)[t]$ cannot precisely measure the contribution of the attention head $\mathbf{a}_j^l$ to the residual stream $\mathbf{h}_{\text{mid}}^{l+1}$ (i.e., considering the information on the token: $f_{\text{logit-lens}}(\mathbf{h}_{\text{mid}}^{l+1})[t]$). Instead, the difference $f_{\text{logit-lens}}(\mathbf{h}_{\text{mid}}^{l+1})[t] - f_{\text{logit-lens}}(\mathbf{h}_{\text{mid}}^{l+1} - \mathbf{a}_i^l)[t]$ is a better proxy of the contribution, which directly calculates the effect of knocking out $\mathbf{a}_j^l$ from the $l$-th layer on the residual stream value of the model.

For another, previous methods that directly inspect the head representations neglect the large scale discrepancy of target-token probabilities across layers, which fluctuate only within a narrow band in earlier layers but surge dramatically in the final layers (Figure 4(c)). To this end, we additionally normalize the effect with its original value. In conclusion, our aw heads measure for $\mathbf{a}_i^l$ is formally expressed as:

$$(f_{\text{logit-lens}}(\mathbf{h}_{\text{mid}}^{l+1})[t] - f_{\text{logit-lens}}(\mathbf{h}_{\text{mid}}^{l+1} - \mathbf{a}_i^l)[t])/f_{\text{logit-lens}}(\mathbf{h}_{\text{mid}}^{l+1})[t] \triangleq s_{\text{aw-head}}(\mathbf{a}_i^l).$$

We also use the Qwen2.5-7B-instruct model and the Parity-NL task (error type 2) to empirically compare the locating results of our proposed measure $s_{\text{aw-head}}(\mathbf{a}_i^l)$ (in short, ***Ours***) with several baseline methods: (i) directly inspecting the head representations with Logit Lens probability

$f_{\text{logit-lens}}(\mathbf{a}_i^l)[t]$ (in short, **LL prob**), (ii) the logit value with Logit Lens $(\mathbf{a}_i^l \cdot W_U)[t]$ (in short, **LL logit**) and (iii) our method without normalization $f_{\text{logit-lens}}(\mathbf{h}_{\text{mid}}^{l+1})[t] - f_{\text{logit-lens}}(\mathbf{h}_{\text{mid}}^{l+1} - \mathbf{a}_i^l)[t]$ (in short, **w.o. normalization**).

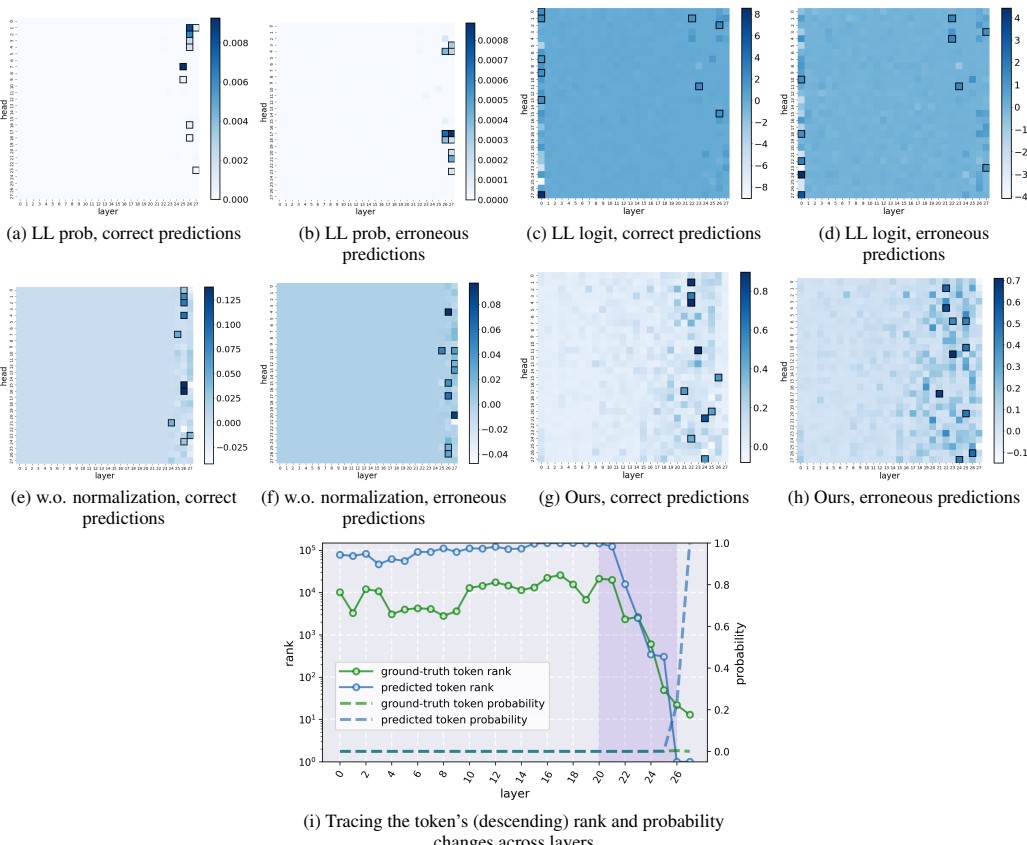

(a) LL prob, correct predictions

(b) LL prob, erroneous predictions

(c) LL logit, correct predictions

(d) LL logit, erroneous predictions

(e) w.o. normalization, correct predictions

(f) w.o. normalization, erroneous predictions

(g) Ours, correct predictions

(h) Ours, erroneous predictions

(i) Tracing the token's (descending) rank and probability changes across layers

Figure 12: Comparing our proposed answer-writing head locating measure with other baseline methods on correct and erroneous predictions.

The results are shown in Figure 12, where we can easily observe that *ours* results (Figure 12(g) and (h)) are most rational (align best with the shadowed zone highlight in Figure 12(i), where the average (descending) ranks and the average probabilities of the erroneously predicted tokens and the corresponding ground-truth tokens change sharply), while **LL prob** and **ours w.o. normalization** mainly locates heads in the last two layers and the most significant heads highlighted by **LL logit** are mainly located in the first layer. Meanwhile, our measure yields the best consistency between the locating results on the correct and erroneous predictions.

**Why using normalization is necessary when locating aw heads?** When using the unnormalized metric Figure 12(e,f), the identified answer-writing heads fall almost entirely into the final 2∼3 layers. However, this starkly contradicts the layer-wise distribution of answer-related information shown in Figure 3(c), where the information of the answer-related token emerges and outstands around layers 20–26 rather than only in the final 2∼3 layers. As we discussed, the reason behind this phenomenon is that target-token probabilities fluctuate only within a narrow band in earlier layers but surge dramatically in the final layers.

**Comparison experiment: how about the localization result of answer-writing heads with randomly select tokens?** To investigate this, we randomly select 10 random tokens from the vocabulary for each sample in $S_{corr}$ and $S_{err}$, and conduct the experiments of locating answer-writing head. The average localization result is shown in Figure 13. Unlike the original localization result, there is no attention heads with significant knock out effect (the maximum of the average intervention effect is around 0.06, in contrast to the result shown in Figure 4 of the paper), which further

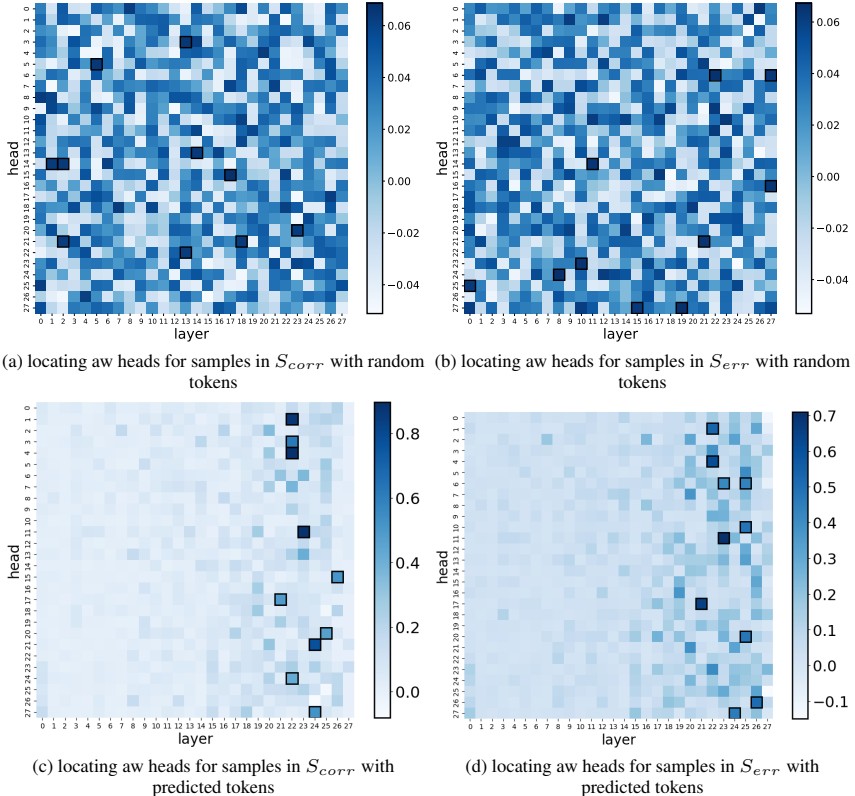

Figure 13: Comparing the answer-writing heads localization result with randomly select tokens and predicted tokens.

confirms the effectiveness of the localization result obtained with model's predicted token ($S_{err}$ and $S_{corr}$).

**Locating answer-writing heads with other tasks and models** We show the additional locating results of answer-writing heads (with our proposed measure $s_{\text{aw-head}}(\mathbf{a}_i^l)$) on all seven tasks and with Qwen2.5-7B-Instruct (Figure 17), Phi-3-Instruct (Figure 18) and LLaMA3-8B-Instruct (Figure 19).

**Case Study: inspecting three answer-writing heads with Logit Lens** We provide the results of using Logit Lens 2 to inspect three answer-writing heads ($\mathbf{a}_1^{22}$, $\mathbf{a}_4^{22}$ and $\mathbf{a}_{11}^{23}$) in the Qwen2.5-7B-Instruct model on the predictions at the token positions of Parity-NL error type 2. We show the results of 10 **originally erroneously predicted** samples accompanied with the inspecting results after intervention the model (i.e., knocking out an erroneous processing head $\mathbf{a}_7^0$) in Figure 14, Figure 20 and Figure 21.

## F.4 ADDITIONAL RESULTS ABOUT LOCATING AND ANALYZING PROCESSING HEADS

We show the locating results of erroneous processing heads (i.e., ***ep heads***) and the effects of knocking out individual heads on correcting the erroneous prediction (i.e., the probabilities of predicting ground-truth tokens) with four models and all seven tasks here:

- Qwen2.5-7B-Instruct: The results are shown in Figure 22,
- Phi-3-Instruct: The results are shown in Figure 23,
- LLaMA3-8B-Instruct: The results are shown in Figure 24,
- Qwen3-8B-Instruct: The results are shown in Figure 25.

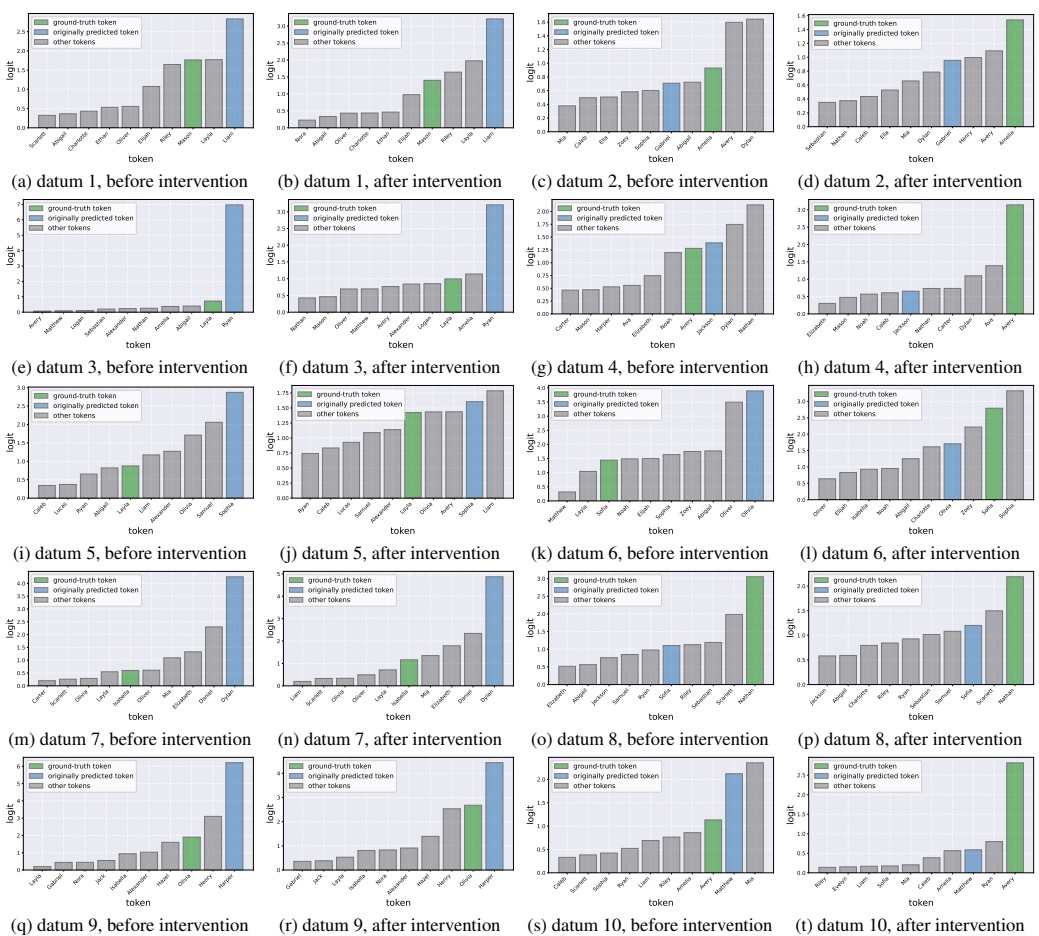

Figure 14: Detailed inspecting results of $\mathbf{a}_1^{22}$ in the Qwen2.5-7B-Instruct model on the predictions at the token positions of Parity-NL error type 2.

Note that the Parity-NL results with Qwen2.5-7B-Instruct have been shown in the main text (Figure 6) and all results are obtained by averaging over a set $S_{\text{err}}$ containing 10 independently and randomly sampled instances. We can consistently observe that knocking out some individual ep heads can effectively improve the probability of predicting the ground-truth token, which is highly aligned with our observations in Section 4 in the main text.

# G  ADDITIONAL INFORMATION ABOUT TEST-TIME INTERVENTION TO CORRECT REASONING HOP GENERALIZATION ERRORS

## G.1  IDENTIFYING CANDIDATE ERRONEOUS PROCESSING HEAD SETS

We observe that *different tasks and key error types might be mapped onto some common ep heads* in previous experiments (see the locating results for the ep heads in Appendix F.4), which motivates us to maintain a small set of key heads for each model, which can be commonly used to rectify as diverse tasks as possible and effectively reduce the size of $\mathbf{H}$. Following the procedure described in Section 4, we randomly sample two sets of correct and erroneous predictions to separately locate $\mathcal{H}_{\text{ep}}$ for each key error type in five reasoning hop generalization tasks (**Parity-NL**, **MDM**, **LLC**, **CLF** and **MOAS**), including symbolic reasoning, mathematical calculation and coding domains. We then select a list of these target heads $\mathbf{H}$, where (i) each single head should be located by as many key error types as possible, and meanwhile (ii) they together can guarantee to cover all of the key error types. We implement a simple greedy algorithm (for this Set Cover (Slavik, 1997) like

problem) to determine $\mathbf{H}$. Consequently, $|\mathbf{H}| = 8$ for Qwen2.5-7B, 9 for Phi-3 and Qwen3-8B and 10 for LLaMA3-8B. Below are the details of $\mathbf{H}$ for all of the models used in this work.

- Qwen2.5-7B-Instruct: $\mathbf{H} = \{\mathbf{a}_1^0, \mathbf{a}_3^0, \mathbf{a}_6^0, \mathbf{a}_7^0, \mathbf{a}_{15}^0, \mathbf{a}_{13}^1, \mathbf{a}_{11}^3, \mathbf{a}_{22}^8\}$.
- Phi-3-Instruct: $\mathbf{H} = \{\mathbf{a}_9^1, \mathbf{a}_{27}^2, \mathbf{a}_4^4, \mathbf{a}_{19}^7, \mathbf{a}_5^{12}, \mathbf{a}_8^{12}, \mathbf{a}_{21}^{13}, \mathbf{a}_6^{15}, \mathbf{a}_{30}^{21}\}$.
- LLaMA3-8B-Instruct: $\mathbf{H} = \{\mathbf{a}_3^0, \mathbf{a}_{23}^1, \mathbf{a}_{25}^4, \mathbf{a}_{21}^6, \mathbf{a}_{17}^{10}, \mathbf{a}_{20}^{11}, \mathbf{a}_{12}^{11}, \mathbf{a}_{19}^{13}, \mathbf{a}_{18}^{13}, \mathbf{a}_{24}^{17}\}$.
- Qwen3-8B-Instruct: $\mathbf{H} = \{\mathbf{a}_1^0, \mathbf{a}_{21}^8, \mathbf{a}_{20}^{14}, \mathbf{a}_9^{15}, \mathbf{a}_{12}^{18}, \mathbf{a}_{10}^{21}, \mathbf{a}_{28}^{25}, \mathbf{a}_0^{28}, \mathbf{a}_8^{29}\}$.

### G.2 HEAD SELECTOR TRAINING DETAILS

**Collecting Training Set, ID Testing Set and OOD Testing Set**  For each of Parity-NL, MDM, LLC, CLF and MOAS, we sample around $20,000$ instances, obtain the erroneous predictions of key error types (Section E), label the $i$-th erroneous prediction $(x_i, y_i)$ and finally collect $\mathcal{D}$. We filter out a small part of instances with $y_i = \mathbf{0}_{|\mathbf{H}|}$. We randomly held out $500$ samples from $\mathcal{D}$ for in-distribution testing. The rest data are used as the training set $\mathcal{D}_{\text{train}}$. We additionally use 200 error predictions in **ObjC** (symbolic reasoning and mathematical calculation) and **NumS** (coding) tasks for out-of-distribution testing.

**Training Details**  In the inference time, we want to establish the correlation between target ep heads and the input context. Hence, we aim to train a small model to automatically select a target ep head among a candidate head set $\mathbf{H}$ according to the input context. We further model this as a *multi-label classification task with a variable number of positive labels*: **input** is the input context $x_i$, **output** is a label $y_i \in \{0,1\}^{|\mathbf{H}|}$ (for $k \in [0..|\mathbf{H}|-1]$, if knocking out $\mathbf{H}[k]$ can correct the prediction $x_i$, then $y_i[k] = 1$ otherwise $y_i[k] = 0$) and $\mathcal{D} = \{(x_i, y_i)\}_{i=1}^{|\mathcal{D}|}$ is the training set. As for the training objective, among BCE loss, BPR loss (Rendle et al., 2012), ZLPR loss (Su et al., 2022) and multi-label Softmax loss (Pu et al., 2025), we find that the last one consistently achieve the best generalization performance. We fine-tune the pre-trained Qwen2.5-0.5B-Instruct[10] model with a randomly initialized multi-label classification head (implemented by the Huggingface transformers[11] class AutoModelForSequenceClassification, denoted as $f_\theta(\cdot)$) with LoRA (Hu et al., 2022) for this task. Our training objective can be formalized as $\min_\theta \mathbb{E}_{(x,y) \in \mathcal{D}_{train}} [-\log(\frac{\exp(f_\theta(x)) \cdot y}{\exp(f_\theta(x)) \cdot \mathbf{1}_{|\mathbf{H}|}})]$.

For the LoRA training, we use Huggingface peft package[12]. Our LoRA configurations are listed as below (Listing2):

Listing 2: LoRA configs

```
lora_config = LoraConfig(
    r = lora_rank, # lora_rank = 16 in all of the experiments
    lora_alpha = lora_alpha, # lora_alpha = 32 in all of the experiments
    inference_mode = False,
    target_modules = ['q_proj', 'k_proj', 'v_proj', 'o_proj'],
    lora_dropout = 0.05,
    bias = 'none',
    task_type = 'SEQ_CLS',
    modules_to_save = ["score"],
)
```

**Training Hyperparameters**  We set training batch size as 16, the maximum gradient norm is 1.0, the initial learning rate as $1e - 4$, and we adopt the cosine learning rate scheduler to adjust our learning rate automatically. We train each model for 3 epochs in total. We show an example of the training logs of the head selector for the Qwen2.5-7B-Instruct model in Figure 15. In this figure, we show the changes of training loss, validation loss, validation accuracy (for both 0 and 1 classes), F1-macro, Precision-macro and Recall-macro, and testing accuracy and testing F1-macro in the training process.

---

[10] https://huggingface.co/Qwen/Qwen2.5-0.5B-Instruct
[11] https://huggingface.co/docs/transformers
[12] https://github.com/huggingface/peft

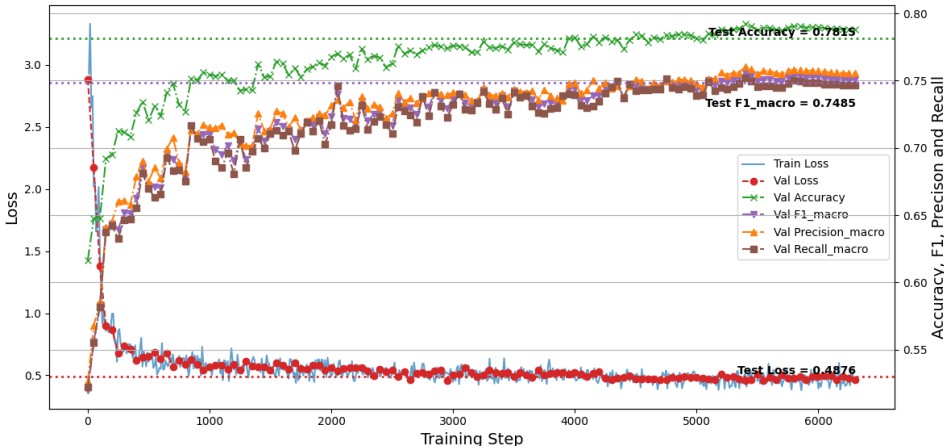

Figure 15: The training logs of the head selector for Qwen2.5-7B-Instruct.

**Random Guessing Baseline** in the Table 1 refers to the baseline of randomly selecting a candidate head. The results are averaged over five independent runs with different random seeds ($1 \sim 5$).

### G.3  THE DETAILS OF TCR AND DoLA EXPERIMENTS

This section provides more details about the experiments of dynamically correcting reasoning hop generalization errors, for both our proposed TCR method and the DoLa baseline method. The tasks used in the experiments of Qwen2.5-7B-Instruct and Phi-3-Instruct are listed as following: Parity-NL(50 hop), MDM(3 digits $\times$ 6 digits), LLC(6 hop), CLF(30 hop), MOAS(50 hop), ObjC(30 hop) and NumS (10 hop). For experiments of LLaMA3-8B-Instruct: we change the hop number of ObjC to $40$. For experiments of Qwen3-8B-Instruct: we change the digit numbers of MDM to 3 digits $\times$ 8 digits, the hop number of ObjC to $40$ and the hop number of NumS to $20$. The entropy threshold $\tau$ that is used to detect token-level erroneous predictions is set as $0.3$ in all of our experiments.

**DoLa** (Chuang et al., 2024) is a very strong test-time baseline especially for improving factuality and reducing hallucinations in language model outputs, which works by contrasting the logits from the final layer with those from earlier layers of the model, amplifying factual knowledge localized in specific layers and suppressing spurious information. DoLa improves models' performance not only on benchmarks that target on the factuality (e.g., TruthQA (Lin et al., 2022)) but also on reasoning benchmarks like GSM-8K (Cobbe et al., 2021). In our experiments, we adopt the implementation[13] that is integrated into the .generate() method in the Huggingface transfomers package. We sweep over the configurations of DoLa and choose the following ones (which is also recommended in the official document to handle with long, reasoning-intensive answers in our reasoning hop generalization scenarios):

Listing 3: DoLa configs

```
gen_kwargs["dola_layers"] = "low"
gen_kwargs["repetition_penalty"] = 1.2
```

Note that the comparison between our proposed TCR method and DoLa is not in order to demonstrate the state-of-the-art performance of TCR. Instead, we want to show that traditional test-time hallucination mitigating methods (like DoLa) may not consistently and effectively help with reasoning hop generalization.

---

[13]https://huggingface.co/transformers-community/dola

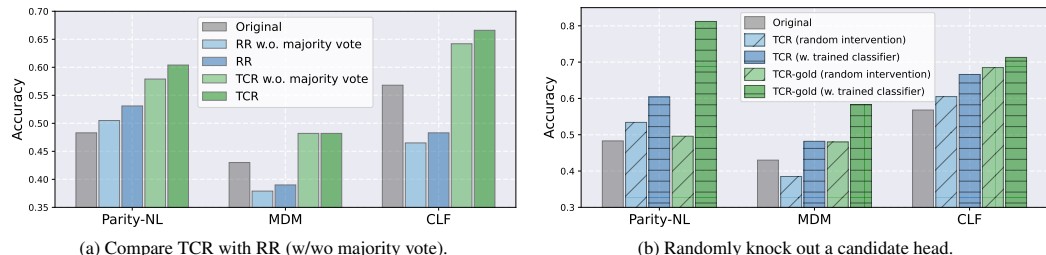

(a) Compare TCR with RR (w/wo majority vote).     (b) Randomly knock out a candidate head.

Figure 16: Ablation Study.

Table 6: Comparing TCR with LOFIT fine-tuned separately on two tasks (Parity-NL(50) and Object Counting(30)).

| Base Model | Original | LOFIT(10) | LOFIT(20) | LOFIT(30) | LOFIT(40) | LOFIT(50) | TCR | TCR-gold |
|---|---|---|---|---|---|---|---|---|
| *Parity-NL(50)* | | | | | | | | |
| Qwen2.5-7B-Instruct | 48.3% | 49.8% | 50.5% | 52.5% | 53.3% | 55.8% | **60.4%** | **81.2%** |
| Phi-3-Instruct | 51.2% | 50.2% | 49.5% | 50.8% | 51.8% | 53.1% | **55.8%** | **65.2%** |
| *ObjC(30)* | | | | | | | | |
| Qwen2.5-7B-Instruct | 52.0% | 44.5% | 50.2% | 54.0% | – | – | **56.0%** | **76.0%** |
| Phi-3-Instruct | 30.9% | 32.8% | 35.8% | 38.3% | – | – | **45.0%** | **45.0%** |

## G.4    ANALYTICAL EXPERIMENT: ABLATION STUDY OF TCR

A natural baseline is **Removing-and-Resampling (RR)**, which removes the logit of the model's original prediction from the final hidden state and re-samples the output. As shown in Figure 16(a), RR yields only marginal gains on Parity-NL but substantially degrades performance on MDM and CLF, highlighting the superiority of TCR. Finally, to assess the role of the trained head selector, we compare it against a random head selector. Figure 16(b) shows that both TCR and TCR-gold with the trained selector consistently outperform their random-selector counterparts on Parity-NL, MDM, and CLF, confirming the effectiveness of classifier-guided head selection.

## G.5    COMPARING TCR AND LOFIT ON IMPROVING REASONING HOP GENERALIZATION

LOFIT (Yin et al., 2024) shares the idea of intervening on attention-head representations, but it differs from TCR in two important ways. The first different point is that LOFIT aims to use fine-tuning to adapt the down-stream task while TCR is a test-time intervention to correct reasoning: LOFIT is primarily introduced as a lightweight alternative to PEFT methods (e.g., LoRA (Hu et al., 2022)), and is typically applied to task-specific fine-tuning (e.g., TruthfulQA, multi-hop QA). It does not particularly target reasoning-hop generalization. TCR is a test-time attention head intervention method grounded on the competition mechanism behind the reasoning hop generalization phenomenon that is studied in our paper, which suppresses the erroneous processing head to correct reasoning. The second different point is that LOFIT identifies task-dependent attention heads and therefore tends to be tied to the specific downstream task used for fine-tuning, making out-of-distribution task generalization difficult while TCR identifies erroneous processing heads with stable competitive dynamics across tasks, enabling effective transfer. We make a direct comparison between TCR and LOFIT. We evaluate two models (Qwen2.5-7B-Instruct and Phi-3-Instruct), and consider the following two settings.

**LOFIT fine-tuned separately on two tasks (Parity-NL and Object Counting).**    The result is shown in Table 6. Note that in the experiments, we try using problems of different hop numbers to train LOFIT to explore its reasoning hop generalization capacity. LOFIT(N) refers to that we use N-hop data to train LOFIT.

**LOFIT jointly fine-tuned on five tasks (Parity-NL, MDM, LLC, CLF, MOAS), while testing out-of-distribution task generalization on two unseen tasks (ObjC and NumS), consistent with our main setup.**    The results are shown in Table 7 and Table 8.

The results show that: TCR yields substantial and consistent improvements on reasoning-hop generalization over LOFIT on Parity-NL and Object Counting; LOFIT's improvements are much more

Table 7: Comparing LOFIT and TCR on Out-Of-Distribution Task Generalization with Qwen2.5-7B-Instruct.

| Qwen2.5-7B-Instruct | original | LOFIT | TCR | TCR-gold |
|---|---|---|---|---|
| ObjC | 52.0% | 53.8% | **56.0%** | **76.0%** |
| NumS | 41.1% | 40.3% | **46.0%** | **54.5%** |
| Average | 46.6% | 47.1% | **51.0%** | **65.3%** |

Table 8: Comparing LOFIT and TCR on Out-Of-Distribution Task Generalization with Phi-3-Instruct.

| Phi-3-Instruct | original | LOFIT | TCR | TCR-gold |
|---|---|---|---|---|
| ObjC | 30.9% | 33.2% | **45.0%** | **45.0%** |
| NumS | 72.0% | 68.5% | **76.0%** | **82.0%** |
| Average | 51.5% | 50.9% | **60.5%** | **63.5%** |

task-specific, and it fails to achieve effectiveness out-of-distribution task generalization observed with TCR.

### G.6 COMPARING TCR AND TRAINING THE BASE MODEL ITSELF ON IMPROVING REASONING HOP GENERALIZATION

One natural question would be ***"why we do not train the base model itself on the same trainingset used by TCR to improve reasoning hop generalization?"*** First, TCR is designed to be non-destructive and generalizable to new tasks, while direct fine-tuning inherently entangles model parameters with the training distribution. Moreover, to address this question, we train the base model (e.g., Qwen2.5-7B-Instruct) using the same datase employed for the knockout classifier, under three different schemes (i.e., Exp1, Exp2 and Exp3):

- **Exp1: trainable base model + trainable classifier head.** In this experiment, we fine-tune both the base model and the classifier head for the multi-label classification task.

- **Exp2: frozen base model + trainable classifier head.** In this experiment, we use the base model itself as a frozen backbone and only train the classifier head.

- **Exp3: fine-tuning the base model on ground-truth tokens directly.** In this experiment, we replace the head-classification labels in the training dataset with corresponding ground-truth tokens at critical token positions and directly fine-tune the base model itself.

The results of Exp1, Exp2 and Exp3 are shown in Table 9. Across all three schemes, training the base model itself leads to catastrophic forgetting (Exp1), insufficient expressivity for classification (Exp2), poor in-distribution and out-of-distribution task generalization (Exp3).

### G.7 EXPERIMENT WITH LARGER MODEL

In this section, we aim to investigate whether TCR can work with larger LLMs. To this end, we re-implement TCR with Qwen2.5-14B-Instruct[14]. First, for Qwen2.5-14B-Instruct, the selected candidate head set $\mathbf{H} = \{\mathbf{a}_3^0, \mathbf{a}_7^0, \mathbf{a}_{18}^0, \mathbf{a}_{15}^0, \mathbf{a}_{23}^0, \mathbf{a}_{14}^0, \mathbf{a}_{27}^0, \mathbf{a}_2^{14}, \mathbf{a}_{36}^{19}, \mathbf{a}_{34}^{20}\}$. Following the procedure in the main text, we also train a Qwen2.5-0.5B-Instruct based multi-label classification model (base model with a classifier head) to predict which head to be intervened in the test-time. We use LoRA to fine-tune the parameters of the backbone Qwen2.5-0.5B model and train the randomly initialized classifier head in the same time. We adopt the softmax loss (Pu et al., 2025) where the training objective can be formalized as $\min_\theta \mathbb{E}_{(x,y)\in\mathcal{D}}[-\log(\frac{\exp(f_\theta(x))\cdot y}{\exp(f_\theta(x))\cdot \mathbf{1}_{|\mathbf{H}|}})]$.

We evaluate the performance of the base model (Qwen2.5-14B-Instruct), base model with DoLa Chuang et al. (2024), base model with TCR and TCR-gold on seven tasks (five in-distribution

---

[14]Qwen2.5-14B-Instruct: `https://huggingface.co/Qwen/Qwen2.5-14B-Instruct`

Table 9: Comparing TCR and three implementations of training the base model itself on improving reasoning hop generalization.

| Exp Group | Parity-NL | MDM | LLC | CLF | MOAS | ObjC | NumS | Average |
|---|---|---|---|---|---|---|---|---|
| Base Model | 48.3% | 43.0% | 11.7% | 56.8% | 39.2% | 52.0% | 41.1% | 41.7% |
| Exp1 | 49.5% | 29.5% | 10.2% | 48.3% | 30.2% | 45.8% | 28.0% | 34.5% (-7.2%) |
| Exp2 | 50.3% | 45.0% | 13.5% | 63.6% | 44.8% | 52.6% | 41.9% | 44.5% (+2.8%) |
| Exp3 | 56.2% | 43.7% | 12.2% | 59.5% | 44.0% | 52.0% | 39.8% | 43.9% (+2.2%) |
| TCR | **60.4%** | **48.2%** | **16.2%** | **66.6%** | **46.0%** | **56.0%** | **46.0%** | **48.5%** (+6.8%) |
| TCR-gold | **81.2%** | **58.3%** | **23.0%** | **71.3%** | **62.0%** | **76.0%** | **54.5%** | **61.3%** (+19.6%) |

Table 10: Performance of TCR and TCR-gold with Qwen2.5-Instruct-14B.

| | Parity-NL | MDM | LLC | CLF | MOAS | ObjC | NumS | Average |
|---|---|---|---|---|---|---|---|---|
| Base Model | 74.4% | 32.3% | 43.3% | 89.9% | 60.5% | 47.7% | 72.4% | 60.1% |
| Dola | 76.1% | 28.5% | 42.5% | 91.0% | 58.1% | 47.3% | 70.8% | 59.2% |
| TCR | 79.0% | 37.4% | 45.0% | 94.3% | 66.0% | 54.1% | 75.8% | **64.5**% |
| TCR+gold | 87.7% | 41.5% | 48.6% | 96.0% | 69.8% | 63.7% | 77.2% | **69.2**% |

tasks: Parity-NL(70), MDM(8), LLC(6), CLF(30), and MOAS(50); and two out-of-distribution tasks: ObjC(30) and NumS(20)). The results are shown in Table 10, which demonstrates effectiveness of TCR and TCR-gold with larger models like Qwen2.5-14B-Instruct.

## G.8 The performance of TCR on the Natural Language Reasoning Task

To evaluate our method in a more realistic natural-language reasoning setting, we additionally include results on Big-GSM (Chen et al., 2024), a more complex variant of GSM8K specifically designed to require more reasoning hops. Importantly, we do not separately train the classifier on Big-GSM; instead, we directly apply the classifier trained on our five in-distribution tasks (Parity-NL, MDM, LLC, CLF, MOAS) to select which heads to knock out.

Table 11: Evaluating the performance of TCR on the Big-GSM task.

| Big-GSM | Base Model | TCR |
|---|---|---|
| Qwen2.5-7B-Instruct | $52.7\% \pm 0.4\%$ | $\mathbf{54.4\% \pm 0.5\%}$ |
| Phi-3-Instruct | $52.5\% \pm 1.3\%$ | $\mathbf{53.9\% \pm 0.7\%}$ |

The result is shown in Table 11: our method consistently improves performance on Big-GSM, demonstrating its ability to generalize to more natural-language, long-chain reasoning tasks.

## G.9 Examining the Reasoning Models

In this subsection, we try to explore whether our main findings (i.e., deactivate erroneous processing heads can correct reasoning) still hold with reasoning models that can self-reflect and backtrack in the step-by-step reasoning process (DeepSeek, 2025; OpenAI, 2025; Yeo et al., 2025). To this end,

|  | Parity-NL (50-hop) | | MOAS (50-hop) | | ObjC (30-hop) | | Average | |
| --- | --- | --- | --- | --- | --- | --- | --- | --- |
|  | Acc. | Resp. Len. | Acc. | Resp. Len. | Acc. | Resp. Len. | Acc. | Resp. Len. |
| **Original** | 63.3% | 1633.6 | 38.9% | 1765.9 | 28.5% | 1819.1 | 43.6% | 1739.5 |
| **TCR** | **66.9%** | **1365.3** | **44.1%** | **1454.3** | **31.7%** | **1688.5** | **47.6%** | **1502.7** |

Table 12: The average reasoning accuracy and response length before and after applying TCR with Deepseek-R1-distill-Qwen-7B.

we conduct experiments with the DeepSeek-R1-distill-Qwen-7B model[15]. We separately sample 10 erroneous predictions of the Parity-NL (50-hop) task, the MOAS (50-hop) task, and the ObjC (30-hop) task. Then following the procedure in Section 4, we locate the top-1 ep heads for these tasks ($\mathbf{a}_7^0$ for the Parity-NL task, $\mathbf{a}_6^0$ for the MOAS task, and $\mathbf{a}_8^4$ for the ObjC task), respectively. We implement TCR by dynamically deactivating the task-specific top-1 ep head in the reasoning process when the entropy of the generated token exceeds $0.4$. We show the average reasoning accuracy and response length before and after applying TCR in Table 12.

We can observe that after applying TCR, model's average reasoning accuracy will increase (by $4.0\%$ averagely) and meanwhile the average response will shorten (by $13.6\%$ averagely), demonstrating the TCR's potentials with reasoning models and the functional complementarity between the reasoning models' self-reflection behavior and the inference-time suppression of erroneous processing heads.

---

[15]https://huggingface.co/deepseek-ai/DeepSeek-R1-Distill-Qwen-7B

# H  FIGURES FOR THE APPENDIX SECTIONS

This section contains lots of figures mentioned in previous Appendix sections.

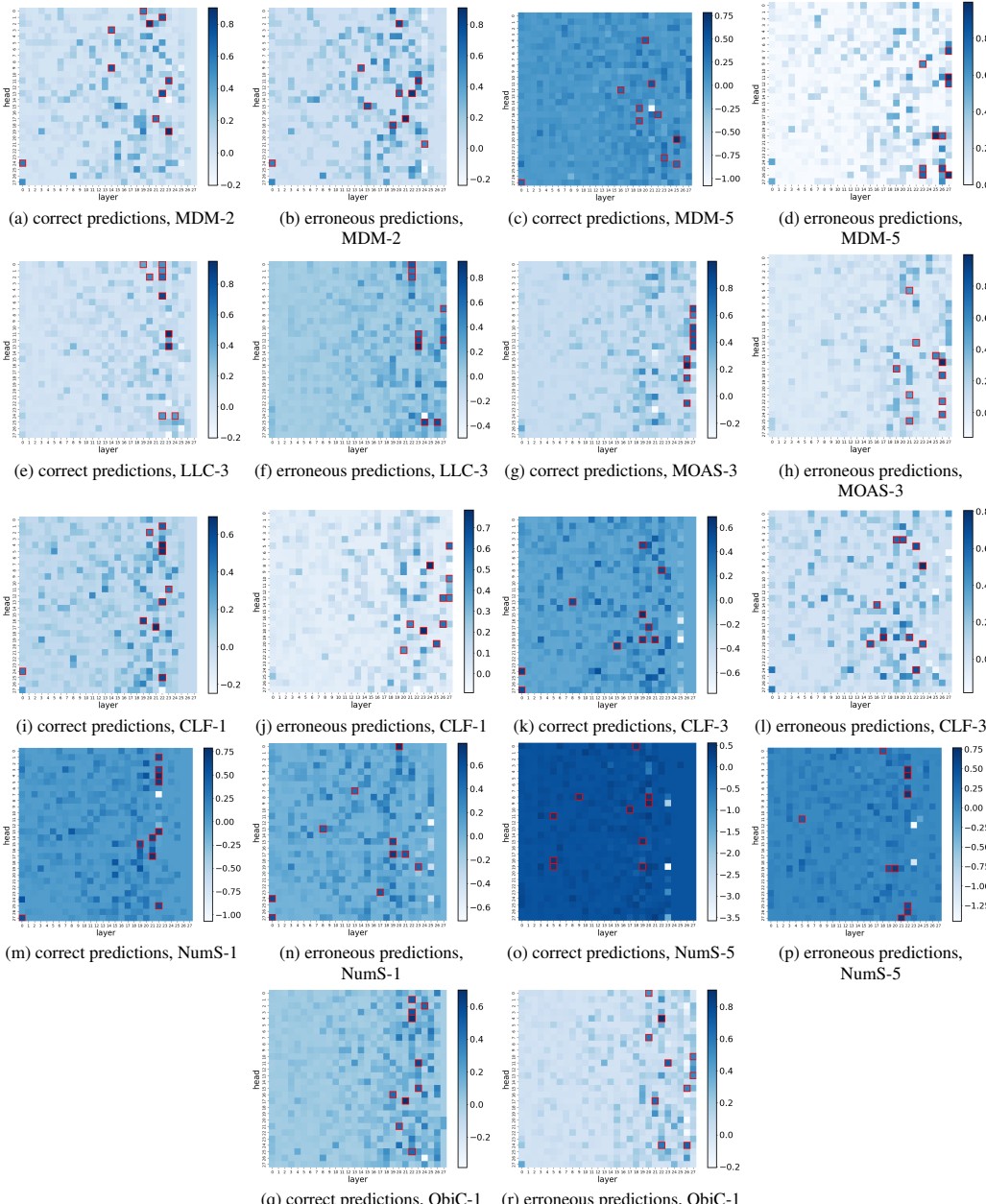

Figure 17: Additional results about locating answer-writing heads with Qwen2.5-7B-Instruct on both correct and erroneous predictions. In the caption of each sub-figure, the "X" in [Task Name]-X refers to the error type. For example, MDM-2 refers to the erroneous predictions of error type 2 of MDM.

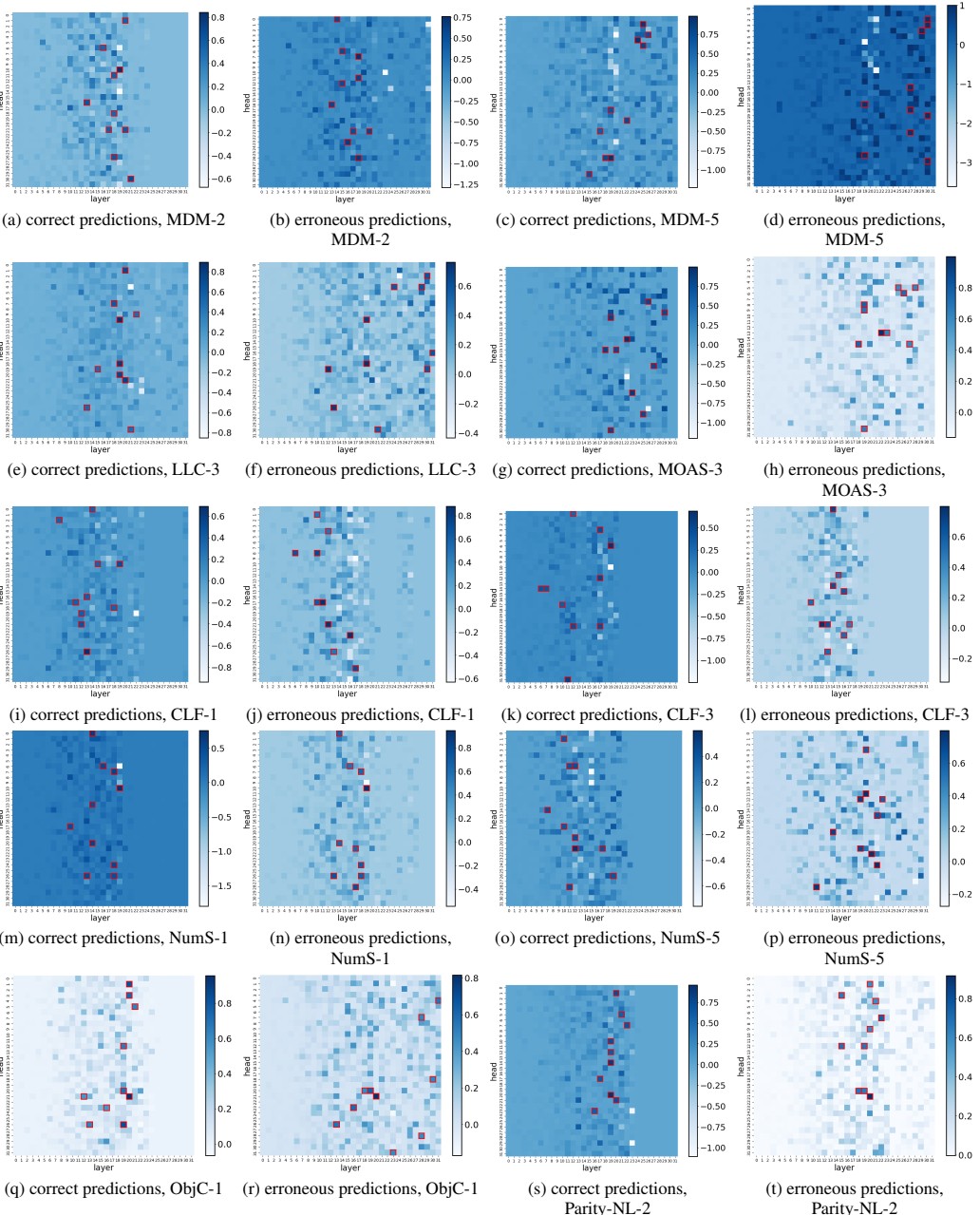

Figure 18: Additional results about locating answer-writing heads with Phi-3-Instruct on both correct and erroneous predictions. In the caption of each sub-figure, the "X" in [Task Name]-X refers to the error type. For example, MDM-2 refers to the erroneous predictions of error type 2 of MDM.

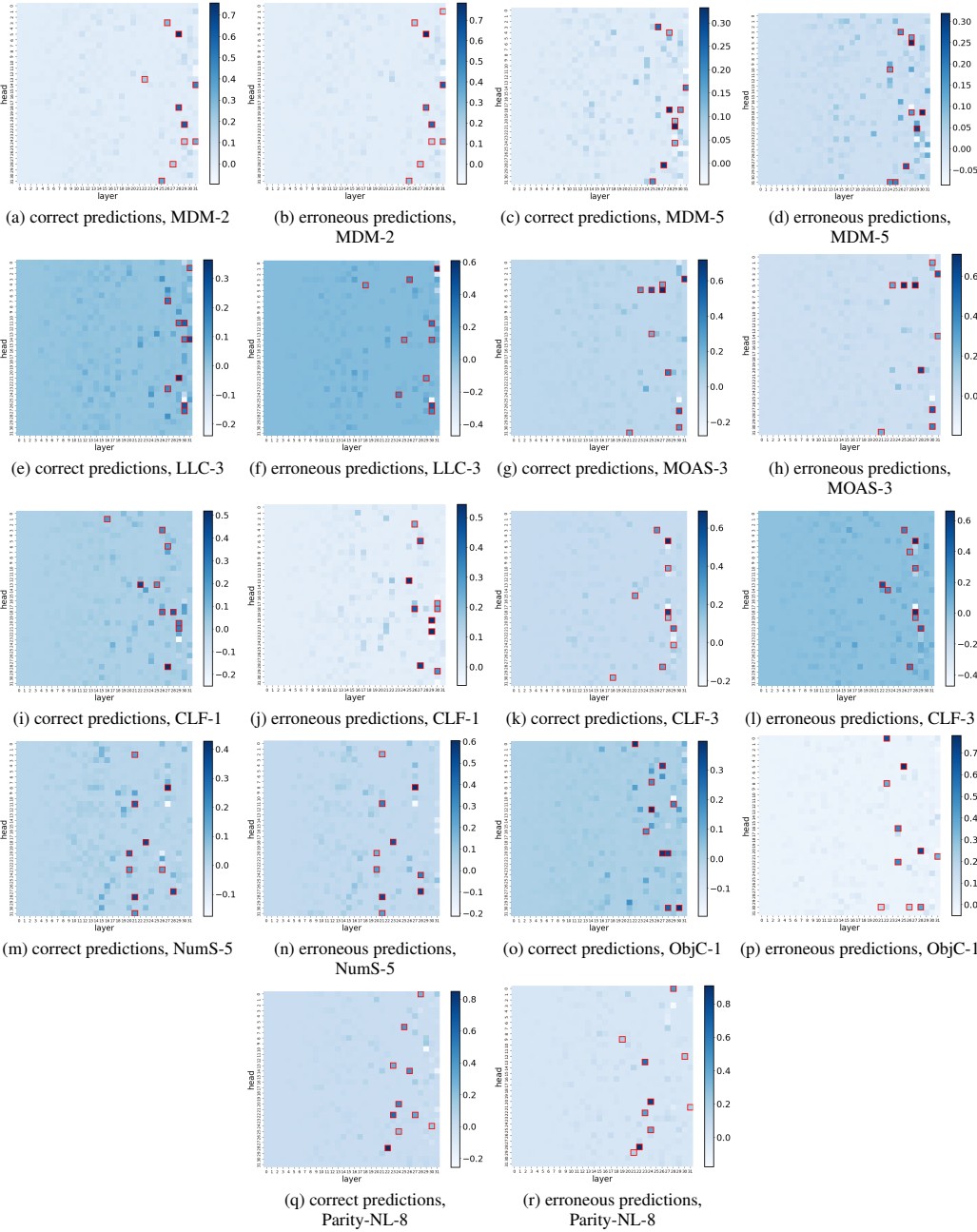

Figure 19: Additional results about locating answer-writing heads with LLaMA3-8B-Instruct on both correct and erroneous predictions. In the caption of each sub-figure, the "X" in [Task Name]-X refers to the error type. For example, MDM-2 refers to the erroneous predictions of error type 2 of MDM.

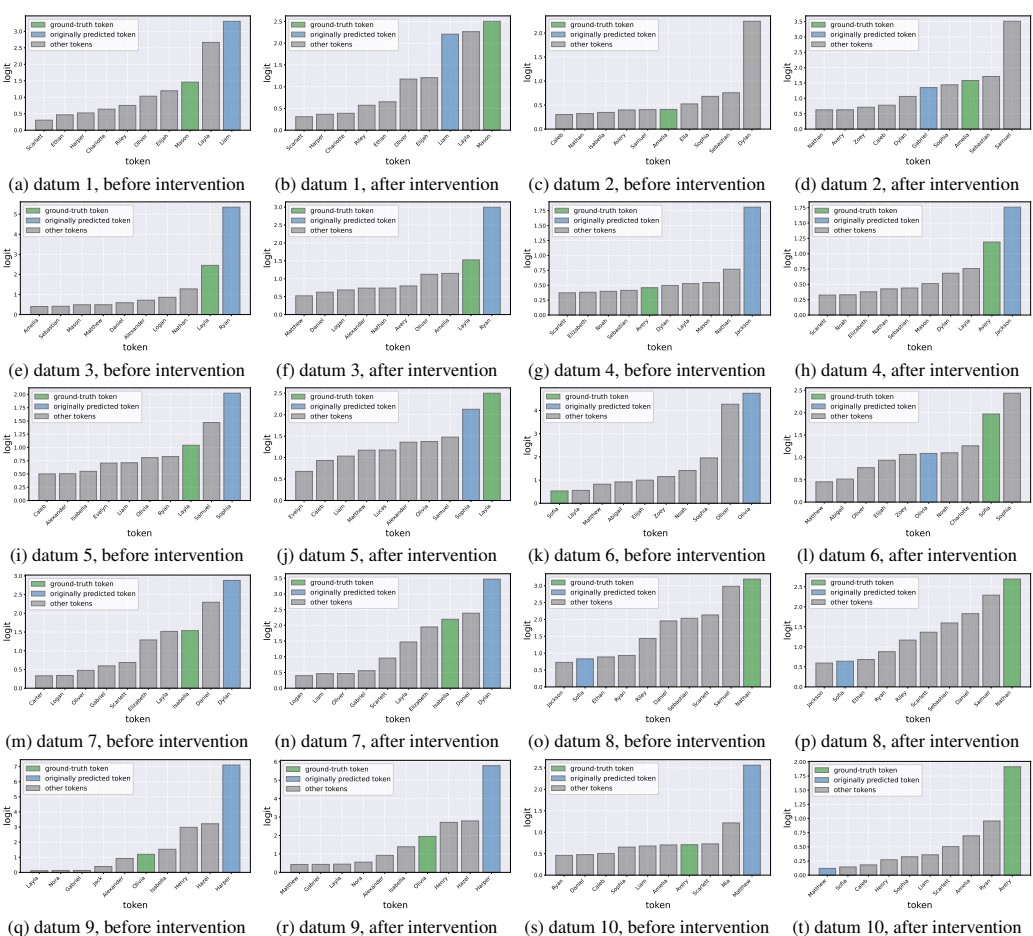

Figure 20: Detailed inspecting results of $\mathbf{a}_4^{22}$ in the Qwen2.5-7B-Instruct model on the predictions at the token positions of Parity-NL error type 2.

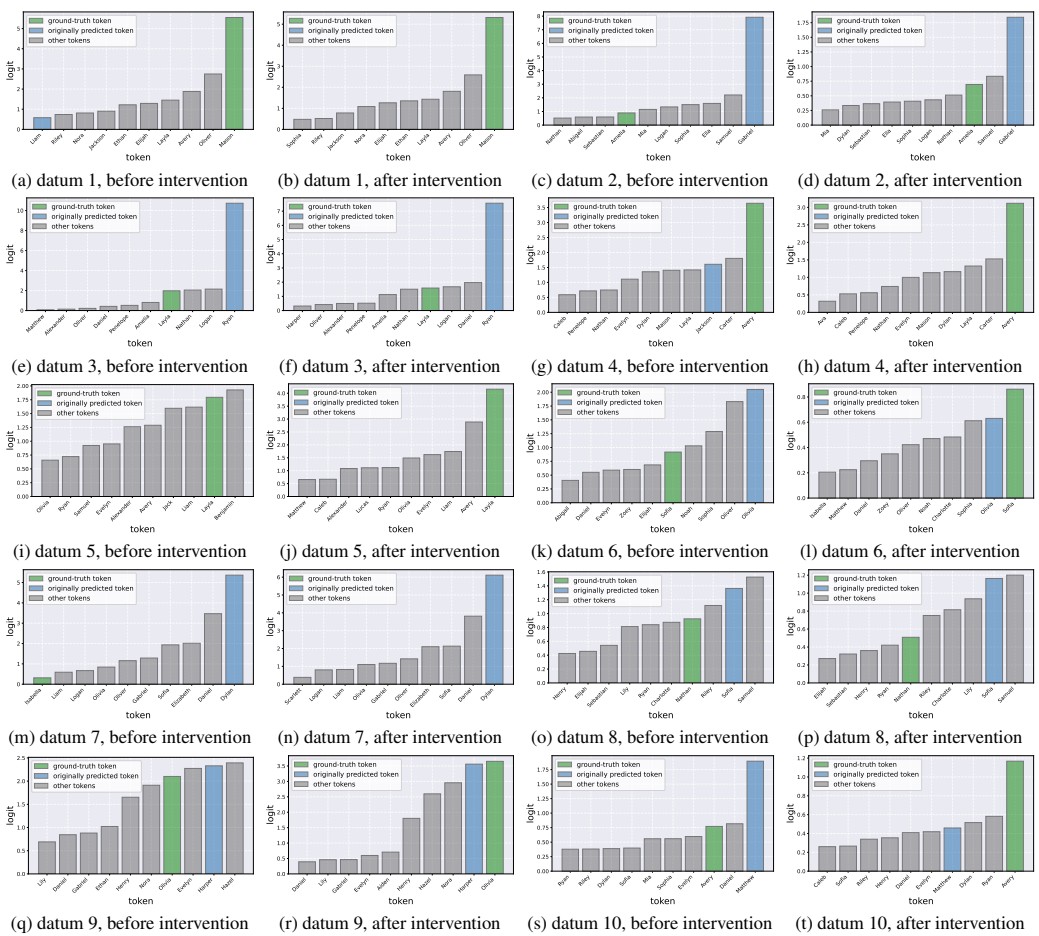

Figure 21: Detailed inspecting results of $\mathbf{a}_{11}^{23}$ in the Qwen2.5-7B-Instruct model on the predictions at the token positions of Parity-NL error type 2.

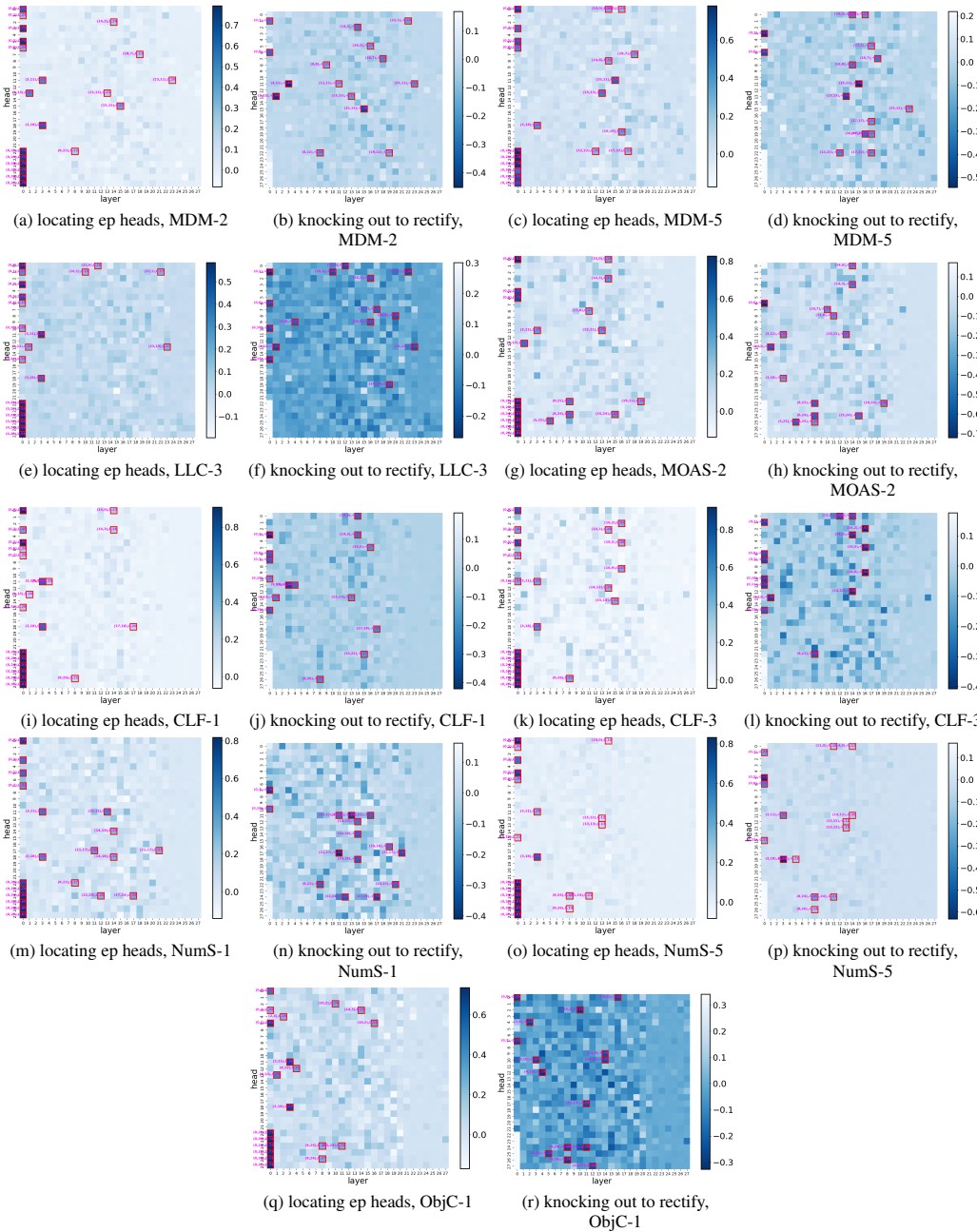

Figure 22: Additional results about locating erroneous processing heads with Qwen2.5-7B-Instruct. "knocking out to rectify" refers to measure the effect of knocking out individual heads on rectifying erroneous predictions (i.e., the probabilities of predicting ground-truth tokens). In the caption of each sub-figure, the "X" in [Task Name]-X refers to the error type. For example, MDM-2 refers to the erroneous predictions of error type 2 of MDM.

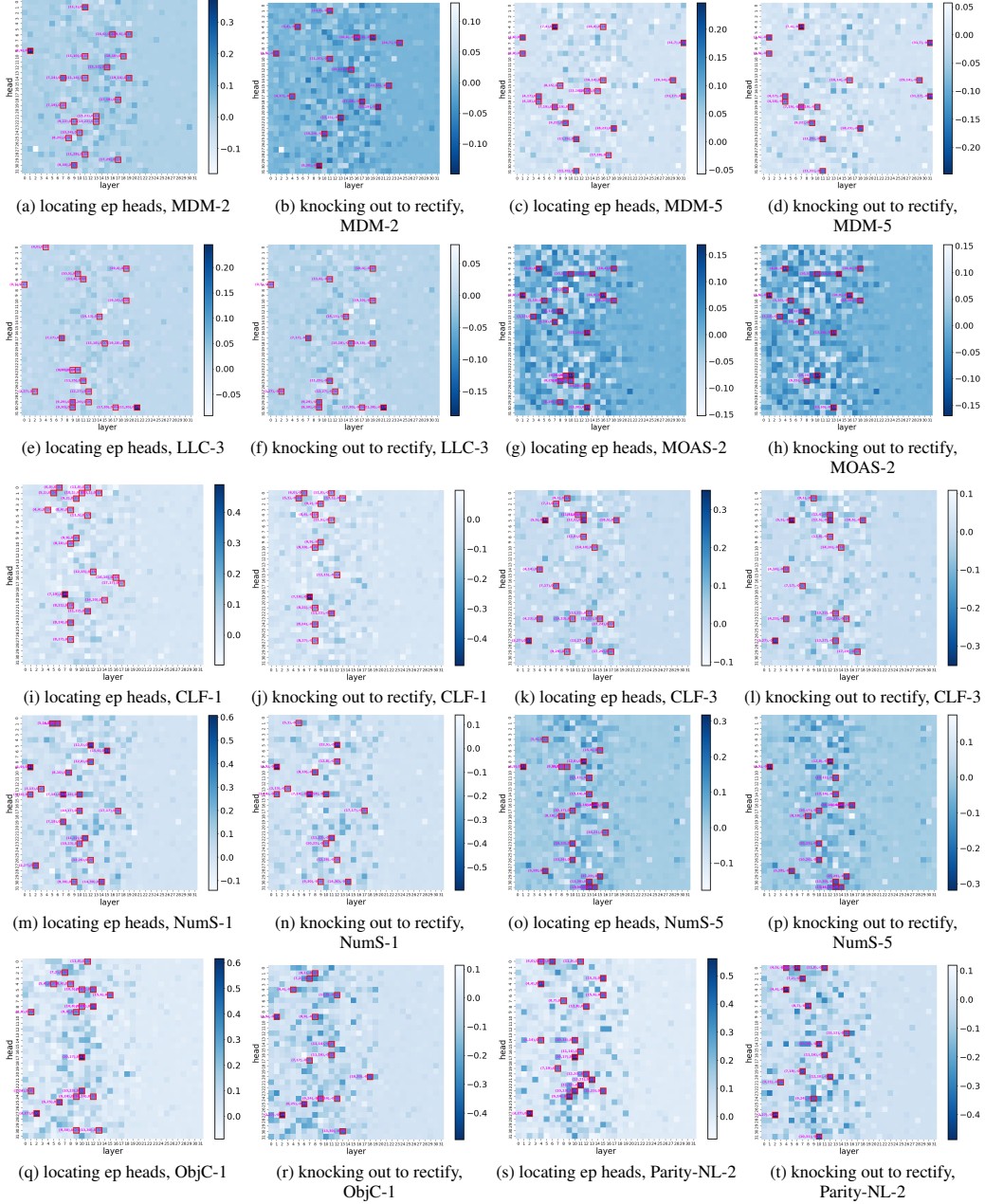

(a) locating ep heads, MDM-2    (b) knocking out to rectify, MDM-2    (c) locating ep heads, MDM-5    (d) knocking out to rectify, MDM-5

(e) locating ep heads, LLC-3    (f) knocking out to rectify, LLC-3    (g) locating ep heads, MOAS-2    (h) knocking out to rectify, MOAS-2

(i) locating ep heads, CLF-1    (j) knocking out to rectify, CLF-1    (k) locating ep heads, CLF-3    (l) knocking out to rectify, CLF-3

(m) locating ep heads, NumS-1    (n) knocking out to rectify, NumS-1    (o) locating ep heads, NumS-5    (p) knocking out to rectify, NumS-5

(q) locating ep heads, ObjC-1    (r) knocking out to rectify, ObjC-1    (s) locating ep heads, Parity-NL-2    (t) knocking out to rectify, Parity-NL-2

Figure 23: Additional results about locating erroneous processing heads with Phi-3-Instruct. "knocking out to rectify" refers to measure the effect of knocking out individual heads on rectifying erroneous predictions (i.e., the probabilities of predicting ground-truth tokens). In the caption of each sub-figure, the "X" in [Task Name]-X refers to the error type. For example, MDM-2 refers to the erroneous predictions of error type 2 of MDM.

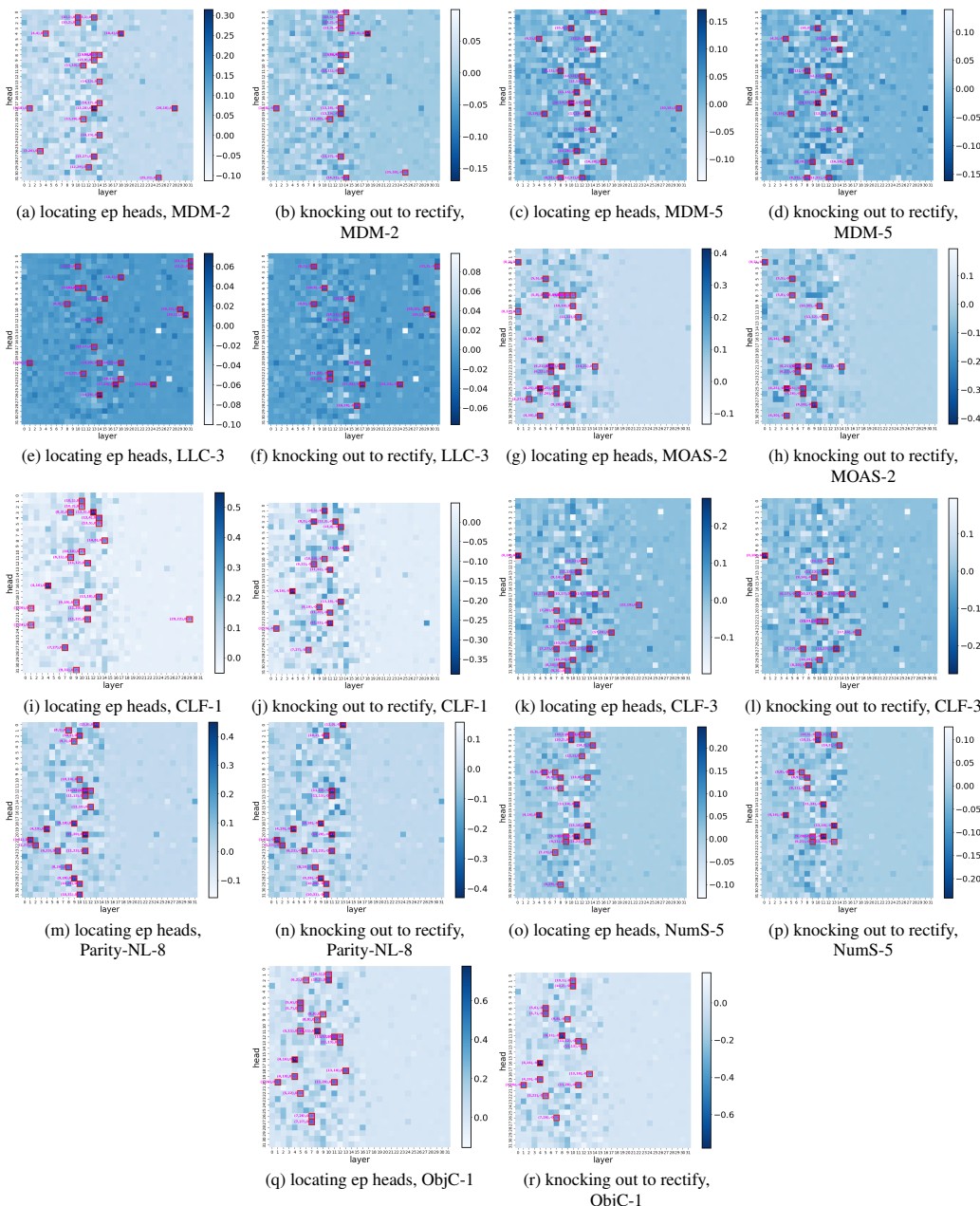

Figure 24: Additional results about locating erroneous processing heads with LLaMA3-8B-Instruct. "knocking out to rectify" refers to measure the effect of knocking out individual heads on rectifying erroneous predictions (i.e., the probabilities of predicting ground-truth tokens). In the caption of each sub-figure, the "X" in [Task Name]-X refers to the error type. For example, MDM-2 refers to the erroneous predictions of error type 2 of MDM.

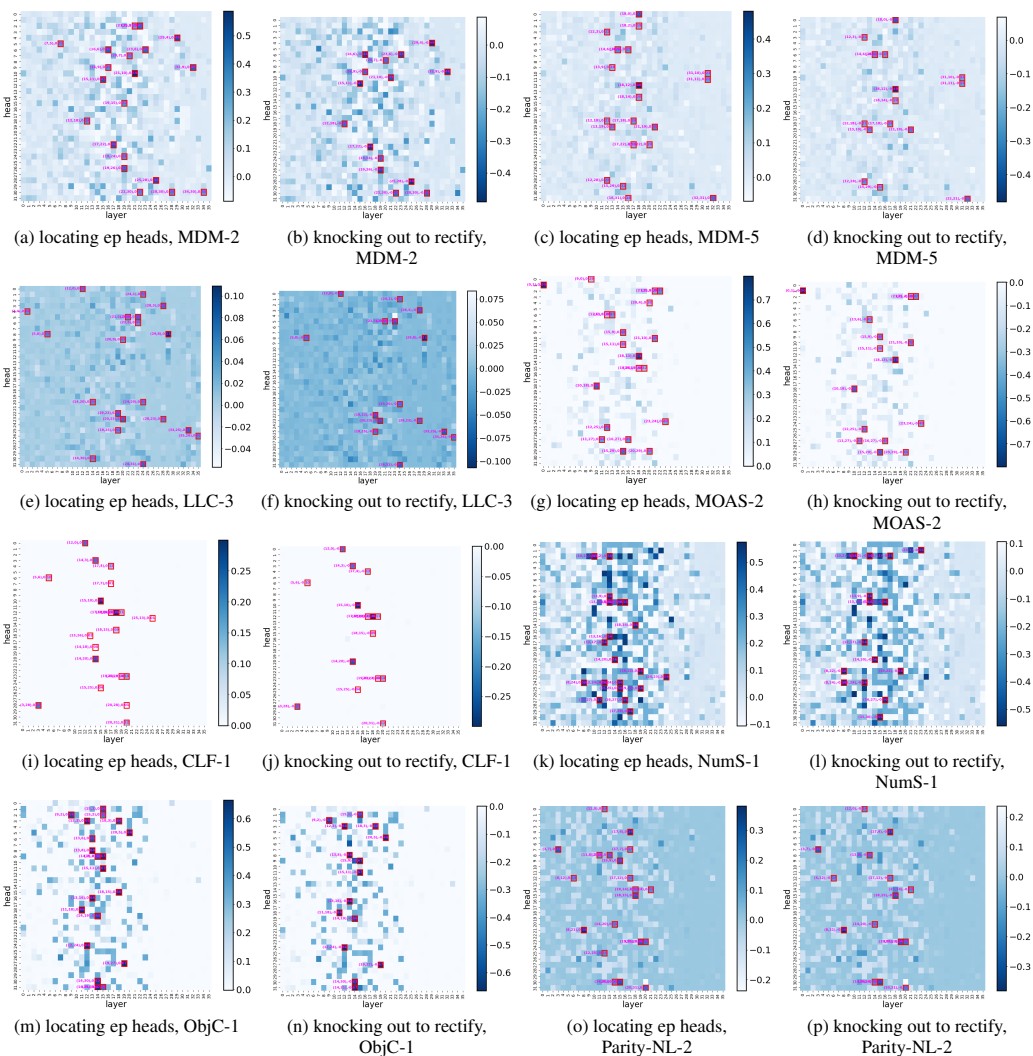

Figure 25: Additional results about locating erroneous processing heads with Qwen3-8B-Instruct. "knocking out to rectify" refers to measure the effect of knocking out individual heads on rectifying erroneous predictions (i.e., the probabilities of predicting ground-truth tokens). In the caption of each sub-figure, the "X" in [Task Name]-X refers to the error type. For example, MDM-2 refers to the erroneous predictions of error type 2 of MDM.

---

**Parity-NL Example**

**Problem:**
The coin is initially heads up. Then Matthew doesn't flip. Then Daniel flips. Then Carter flips. Then Carter doesn't flip. Then Ethan flips. Then Matthew doesn't flip. Then Ethan flips. Then Daniel doesn't flip. Then Matthew flips. Then Ethan flips.
Is the coin finally heads up or tails up?

**Answer:**
Let's analyze the sequence of events step-by-step to determine the final state of the coin.
1. The coin starts heads up.
2. Matthew doesn't flip the coin. (Coin remains heads up.)
3. Daniel flips the coin. (Coin becomes tails up.)
4. Carter flips the coin. (Coin becomes heads up.)
5. Carter doesn't flip the coin. (Coin remains heads up.)
6. Ethan flips the coin. (Coin becomes tails up.)
7. Matthew doesn't flip the coin. (Coin remains tails up.)
8. Ethan flips the coin. (Coin becomes heads up.)
9. Daniel doesn't flip the coin. (Coin remains heads up.)
10. Matthew flips the coin. (Coin becomes tails up.)
11. Ethan flips the coin. (Coin becomes heads up.)
After going through each step, we see that the coin ends up heads up after the last flip by Ethan.
Therefore, the coin is heads up.

---

Figure 26: An example of the Parity-NL problem.

---

**Multi-Digit Multiplication (MDM) Example**

**Problem:**
326 * 3589 =? please think step-by-step.
**Answer:**
Let's break down the multiplication of 326 and 3589 step-by-step. We will multiply 326 by each digit of 3589 and then add the results.
**Step 1: Multiply 326 by 9 (units place of 3589)**
[ 326 * 9 = 2934 ]
(Shift this result 0 places to the left: [ 2934 ])
**Step 2: Multiply 326 by 8 (tens place of 3589)**
[ 326 * 8 = 2608 ]
(Shift this result 1 place to the left: [ 26080 ])
**Step 3: Multiply 326 by 5 (hundreds place of 3589)**
[ 326 * 5 = 1630 ]
(Shift this result 2 places to the left: [ 163000 ])
**Step 4: Multiply 326 by 3 (thousands place of 3589)**
[ 326 * 3 = 978 ]
(Shift this result 3 places to the left: [ 978000 ])
**Step 5: Add all the results together**
[ 2934 + 26080 + 163000 + 978000 ]
**Step 6: Perform the addition step-by-step**
[ 2934 + 26080 = 29014 ]
[ 29014 + 163000 = 192014 ]
[ 192014 + 978000 = 1170014 ]
**Final Result**
[ 326 * 3589 = 1170014 ]

---

Figure 27: An example of the MDM problem.

---

**Last Letter Concatenation (LLC) Example**

**Problem:**
Take the last letters of the words in "garden sound valid potato numb write tiger truth sound hotel" and concatenate them.
**Answer:**
Let us concatenate the last letters one by one.
1. The last letter of 'garden' is 'n'. The current concatenating result is 'n'.
2. The last letter of 'sound' is 'd'. The current concatenating result is 'nd'.
3. The last letter of 'valid' is 'd'. The current concatenating result is 'ndd'.
4. The last letter of 'potato' is 'o'. The current concatenating result is 'nddo'.
5. The last letter of 'numb' is 'b'. The current concatenating result is 'nddob'.
6. The last letter of 'write' is 'e'. The current concatenating result is 'nddobe'.
7. The last letter of 'tiger' is 'r'. The current concatenating result is 'nddober'.
8. The last letter of 'truth' is 'h'. The current concatenating result is 'nddoberh'.
9. The last letter of 'sound' is 'd'. The current concatenating result is 'nddoberhd'.
10. The last letter of 'hotel' is 'l'. The current concatenating result is 'nddoberhdl'.
Therefore, the answer is 'nddoberhdl'.

Figure 28: An example of the LLC problem.

---

**Multi-Operand Addition and Subtraction (MOAS) Example**

**Problem:**
17 + 48 - 25 + 99 - 4 - 85 + 19 + 68 - 31 - 88 = ? Please think step-by-step.
**Answer:**
Certainly! Let us solve the equation step by step:
1. Start with 17.
2. Add 48: 17 + 48 = 65.
3. Subtract 25: 65 - 25 = 40.
4. Add 99: 40 + 99 = 139.
5. Subtract 4: 139 - 4 = 135.
6. Subtract 85: 135 - 85 = 50.
7. Add 19: 50 + 19 = 69.
8. Add 68: 69 + 68 = 137.
9. Subtract 31: 137 - 31 = 106.
10. Subtract 88: 106 - 88 = 18.
So, the final answer is 18.

Figure 29: An example of the MOAS problem.

---

**Crawler Log Folder (CLF) Example**

**Python code:**
```
def min_operations(logs):
    folder_depth = 0
    for operation in logs:
        if operation == '../':
            folder_depth = max(0, folder_depth - 1)
        elif operation != './':
            folder_depth += 1
    return folder_depth
```
**Problem:**
Given a list of change folder operations: "logs = ['d1/', 'd2/', '../', 'd2/', 'd3/', './', '../', 'd3/', 'd4/', './']", what is the return value (folder_depth) when running the "min_operations()" function with the input "logs"?
**Answer:**
Let us break down the operations and compute the folder depth step-by-step for the given list logs = ['d1/', 'd2/', '../', 'd2/', 'd3/', './', '../', 'd3/', 'd4/', './'].
1. **Initial state:** folder_depth = 0.
2. **Operation 'd1/'**: folder_depth = folder_depth + 1 = 0 + 1 = 1.
3. **Operation 'd2/'**: folder_depth = folder_depth + 1 = 1 + 1 = 2.
4. **Operation '../'**: folder_depth = max(0, folder_depth - 1) = max(0, 1) = 1.
5. **Operation 'd2/'**: folder_depth = folder_depth + 1 = 1 + 1 = 2.
6. **Operation 'd3/'**: folder_depth = folder_depth + 1 = 2 + 1 = 3.
7. **Operation './'**: folder_depth = 3. (no change)
8. **Operation '../'**: folder_depth = max(0, folder_depth - 1) = max(0, 2) = 2.
9. **Operation 'd3/'**: folder_depth = folder_depth + 1 = 2 + 1 = 3.
10. **Operation 'd4'**: folder_depth = folder_depth + 1 = 3 + 1 = 4.
11. **Operation './'**: folder_depth = 4. (no change)
Therefore, the final return value (folder_depth) for the "min_operation()" function with the input "logs" is 4.
So the answer is 4.

Figure 30: An example of the CLF problem.

---

**Object Counting (ObjC) Example**

**Problem:**
Alessia got 4 bananas. Gregory picked up 6 watermelons. Lila acquired 4 lemons. Lila bought 3 pineapples. Alan obtained 3 strawberries. Alessia obtained 7 lemons. Alan picked up 9 watermelons. Gregory picked up 6 apples. Alessia picked up 2 oranges. Alan acquired 8 apples.
How many fruit items are mentioned above? Please think step-by-step to answer the question.
**Answer:**
Let's go through the information step-by-step to count the total number of fruit items mentioned.
1. Initially the total number is 0.
2. **Sentence**: Alessia got 4 bananas. **Current Total Number**: 0 + 4 = 4.
3. **Sentence**: Gregory picked up 6 watermelons. **Current Total Number**: 4 + 6 = 10.
4. **Sentence**: Lila acquired 4 lemons. **Current Total Number**: 10 + 4 = 14.
5. **Sentence**: Lila bought 3 pineapples. **Current Total Number**: 14 + 3 = 17.
6. **Sentence**: Alan obtained 3 strawberries. **Current Total Number**: 17 + 3 = 20.
7. **Sentence**: Alessia obtained 7 lemons. **Current Total Number**:20 + 7 = 27.
8. **Sentence**: Alan picked up 9 watermelons. **Current Total Number**: 27 + 9 = 36.
9. **Sentence**: Gregory picked up 6 apples. **Current Total Number**: 36 + 6 = 42.
10. **Sentence**: Alessia picked up 2 oranges. **Current Total Number**: 42 + 2 = 44.
11. **Sentence**: Alan acquired 8 apples. **Current Total Number**: 44 + 8 = 52.
Therefore, the total number of fruit items mentioned is 52.

Figure 31: An example of the ObjC problem.

---

**Number Student (NumS) Example**

**Python code:**
```
def solution(startTime: List[int], endTime: List[int], queryTime: int):
    count = 0
    for i in range(len(startTime)):
        if startTime[i] ≤ queryTime ≤ endTime[i]:
            count += 1
    return count
```
**Problem:**
startTime = [1, 2, 3, 4, 6, 2, 5, 1, 2, 4], endTime = [4, 8, 5, 5, 9, 5, 7, 3, 4, 9], queryTime = 4. What is the final return value after running the function solution(startTime, endTime, queryTime)?
**Answer:**
Let's go through each event and check if queryTime (which is 4) falls within the range of each event's start and end times:
1. startTime[0] = 1, endTime[0] = 4. **range**: [1, 4], 4 is within this range. **count**: 0 + 1 = 1.
2. startTime[1] = 2, endTime[1] = 8. **range**: [2, 8], 4 is within this range. **count**: 1 + 1 = 2.
3. startTime[2] = 3, endTime[2] = 5. **range**: [3, 5], 4 is within this range. **count**: 2 + 1 = 3.
4. startTime[3] = 4, endTime[3] = 5. **range**: [4, 5], 4 is within this range. **count**: 3 + 1 = 4.
5. startTime[4] = 6, endTime[4] = 9. **range**: [6, 9,] 4 is not within this range. **count**: 4.
6. startTime[5] = 2, endTime[5] = 5. **range**: [2, 5], 4 is within this range. **count**: 4 + 1 = 5.
7. startTime[6] = 5, endTime[6] = 7. **range**: [5, 7], 4 is not within this range. **count**: 5.
8. startTime[7] = 1, endTime[7] = 3. **range**: [1, 3], 4 is not within this range. **count**: 5.
9. startTime[8] = 2, endTime[8] = 4. **range**: [2, 4], 4 is within this range. **count**: 5 + 1 = 6.
10. startTime[9] = 4, endTime[9] = 9. **range**: [4, 9], 4 is within this range. **count**: 6 + 1 = 7.
After checking all events, the function returns the final count, which is 7.

Figure 32: An example of the NumS problem.

---

**Possible Error Types in a LLC Example**

**Problem:**
Take the last letters of the words in "garden sound valid potato numb write tiger truth sound hotel" and concatenate them.
**Answer:**
Let us concatenate the last letters one by one.
1. The last letter of 'garden' **[Type 1: Word Recall Error, 'garden' → 'valid' ]** is 'n'. The current concatenating result is 'n'.
2. The last letter of 'sound' is 'd' **[Type 2: Letter Error, 'd' → 'n' ]**. The current concatenating result is 'nd'.
3. The last letter of 'valid' is 'd'. The current concatenating result is 'ndd' **[Type 3: Concatenation Error, 'ndd' → 'dnd' ]**.

_________

*Omitting intermediate six steps ...*
_________

10. The last letter of 'hotel' is 'l'. The current concatenating result is 'nddoberhdl'.
**[Type 4/5: Reasoning Less/More Steps ]**
Therefore, the answer is 'nddoberhdl'.

Figure 33: Possible error types in a LLC problem. The underline green parts highlight the positions where errors of different types may occur. The **bold blue parts** indicate the error types. For each error type, we only mark one instance.

---

**Possible Error Types in a MOAS Example**

**Problem:**
17 + 48 - 25 + 99 - 4 - 85 + 19 + 68 - 31 - 88 = ? Please think step-by-step.
**Answer:**
Certainly! Let us solve the equation step by step:
1. Start with 17 **[Type 1: Operand Recall Error, 17 → 48 ]**.
2. Add 48 **[Type 2: Operation Recall Error, Add 48 → Subtract 31 ]**: 17 + 48 = 65.
3. Subtract 25: 65 **[Type 3: Local Result Copy Error, 65 → 35 ]** - **[Type 4: Operand Interpretation Error, - → + ]** 25 = 40.
4. Add 99: 40 + 99 = 139 **[Type 5: Local Calculation Error, 139 → 129 ]**.

_________

*Omitting intermediate five steps ...*
_________

10. Subtract 88: 106 - 88 = 18.
**[Type 6/7: Reasoning Less/More Steps ]**
So, the final answer is 18.

Figure 34: Possible error types in a MOAS problem. The underline green parts highlight the positions where errors of different types may occur. The **bold blue parts** indicate the error types. For each error type, we only mark one instance.

---

**Possible Error Types in a CLF Example**

**Python code:**
```
def min_operations(logs):
        folder_depth = 0
        for operation in logs:
                if operation == '../':
                        folder_depth = max(0, folder_depth - 1)
                elif operation != './':
                        folder_depth += 1
        return folder_depth
```
**Problem:**
Given a list of change folder operations: "logs = ['d1/', 'd2/', '../', 'd2/', 'd3/', './', '../', 'd3/', 'd4/', './']", what is the return value (folder_depth) when running the "min_operations()" function with the input "logs"?
**Answer:**
Let us break down the operations and compute the folder depth step-by-step for the given list logs.
1. **Initial state:** folder_depth = 0.
2. **Operation 'd1/' [Type 1: Operation Recall Error, 'd1/' → 'd2/' ]**: folder_depth = folder_depth + 1 = 0 + 1 = 1.
3. **Operation 'd2/'**: folder_depth = folder_depth + 1 = 1 [Type 2: Depth Copy Error, 1 → 0 ] + 1 = 2.
4. **Operation '../'**: folder_depth = max(0, folder_depth - 1) = max(0, 1) = 1.
5. **Operation 'd2/'**: folder_depth = folder_depth + 1 = 1 + 1 = 2 [Type 3: Local Calculation Error, 2 → 1 ].

——————
*Omitting intermediate five steps ...*
——————

11. **Operation './'**: folder_depth = 4. (no change)
[Type 4/5: Reasoning Less/More Steps ]
Therefore, the final return value (folder_depth) for the "min_operation()" function with the input "logs" is 4 [Type 6: Final Depth Copy Error, 4 → 3 ].
So the answer is 4.

---

Figure 35: Possible error types in a CLF problem. The underline green parts highlight the positions where errors of different types may occur. The **bold blue parts** indicate the error types. For each error type, we only mark one instance.

---

**Possible Error Types in an ObjC Example**

**Problem:**
Alessia got 4 bananas. Gregory picked up 6 watermelons. Lila acquired 4 lemons. Lila bought 3 pineapples. Alan obtained 3 strawberries. Alessia obtained 7 lemons. Alan picked up 9 watermelons. Gregory picked up 6 apples. Alessia picked up 2 oranges. Alan acquired 8 apples.
How many fruit items are mentioned above? Please think step-by-step to answer the question.
**Answer:**
Let's go through the information step-by-step to count the total number of fruit items mentioned.
1. Initially the total number is 0.
2. **Sentence**: Alessia [Type 1: Name Recall Error, Alessia → Lila ] got 4 bananas. **Current Total Number**: 0 + 4 = 4.
3. **Sentence**: Gregory picked up 6 watermelons [Type 2: Object Infromation Recall Error, 6 watermelons → seven lemons ]. **Current Total Number**: 4 + 6 = 10.
4. **Sentence**: Lila acquired 4 lemons. **Current Total Number**: 10 [Type 3: Local Result Copy Error, 10 → 20 ] + 4 = 14.
5. **Sentence**: Lila bought 3 pineapples. **Current Total Number**: 14 + 3 = 17 [Type 4: Local Calculation Error, 17 → 18 ].

——————
*Omitting intermediate five steps ...*
——————

11. **Sentence**: Alan acquired 8 apples. **Current Total Number**: 44 + 8 = 52.
[Type 5/6: Reasoning Less/More Steps ]
Therefore, the total number of fruit items mentioned is 52.

---

Figure 36: Possible error types in a ObjC problem. The underline green parts highlight the positions where errors of different types may occur. The **bold blue parts** indicate the error types. For each error type, we only mark one instance.

---

**Possible Error Types in a NumS Example**

**Python code:**
```
def solution(startTime: List[int], endTime: List[int], queryTime: int):
    count = 0
    for i in range(len(startTime)):
        if startTime[i] ≤ queryTime ≤ endTime[i]:
            count += 1
    return count
```

**Problem:**
startTime = [1, 2, 3, 4, 6, 2, 5, 1, 2, 4], endTime = [4, 8, 5, 5, 9, 5, 7, 3, 4, 9], queryTime = 4. What is the final return value after running the function solution(startTime, endTime, queryTime)?

**Answer:**
Let's go through each event and check if queryTime (which is 4) falls within the range of each event's start and end times:

1. startTime[0] = 1 **[Type 1: Time Recall Error, 1 → 5 ]**, endTime[0] = 4. **range**: [1, 4], 4 is within this range. **count**: 0 + 1 = 1.

2. startTime[1] = 2, endTime[1] = 8. **range**: [2 **[Type 2: Time Copy Error, 2 → 8 ]**, 8], 4 is within this range. **count**: 1 + 1 = 2.

3. startTime[2] = 3, endTime[2] = 5. **range**: [3, 5], 4 is within this range. **count**: 2 **[Type 3: Local Result Copy Error, 2 → 1 ]** + 1 = 3.

4. startTime[3] = 4, endTime[3] = 5. **range**: [4, 5], 4 is within this range. **count**: 3 + 1 = 4 **[Type 4: Local Calculation Error, 4 → 3 ]**.

5. startTime[4] = 6, endTime[4] = 9. **range**: [6, 9,] 4 is not within **[Type 5: Condition Judge Error, not within → within ]** this range. **count**: 4.

6. startTime[5] = 2, endTime[5] = 5. **range**: [2, 5], 4 is within this range. **count**: 4 + 1 = 5 **[Type 6: Calculation Logic Error, "4 + 1 = 5" → "4" ]**.

---
*Omitting intermediate three steps ...*

---

10. startTime[9] = 4, endTime[9] = 9. **range**: [4, 9], 4 is within this range. **count**: 6 + 1 = 7.
**[Type 7/8: Reasoning Less/More Steps ]**
After checking all events, the function returns the final count, which is 7.

---

Figure 37: Possible error types in a NumS problem. The underline green parts highlight the positions where errors of different types may occur. The **bold blue parts** indicate the error types. For each error type, we only mark one instance.

