# OpenReview forum: "Scaling Reasoning Hop Exposes Weaknesses: Demystifying and Improving Hop Generalization in Large Language Models"
_ICLR.cc/2026/Conference — ICLR 2026 Poster_

### Official Review · Reviewer_C6Fu · 2025-10-22

**Soundness:** 2
**Presentation:** 2
**Contribution:** 2
**Rating:** 4
**Confidence:** 3

**Summary:**

This paper studied language model tasks that involve “reasoning hop generalization”. They localize errors in the reasoning chains of LMs by identifying predicted token positions which erroneously diverge from the expected token positions. The authors study the role of activation heads in correct vs. incorrect text generation trajectories. They identify a key set of attention heads which play a role in erroneous prediction via a knock-out mechanism. They find that there is substantial overlap between the critical attention heads for both erroneous and correct completion trajectories. They use this finding to motivate the design of test-time correction of reasoning (TRC). They show that applying TRC improves performance across many models across many benchmark tasks.

**Strengths:**

1. The proposed method, TRC, does improve model performance on a wide range of tasks across a reasonable number of models (table 2).
2. It is interesting to show that simply intervening on a trained model can substantially improve model performance. It indicates that models often have the knowledge needed to complete long-context tasks, but simply struggle to do so due to length of context. This validates well known phenomena of LMs struggling to be performant to varying context lengths.
3. I felt the tables and figures were easy to understand.
4. I appreciate the appendix, I found myself needing to reference it several times while reading this paper.
5. Lines 176-178 are a good observation. Existing interpretability tools do struggle to be applied across vast token ranges.
6. The procedure to determine when to intervene (lines 408-417) are both light-weight and reasonable.

**Weaknesses:**

1. I feel like there is a conflation between what this paper dubs “reasoning hops generalization” and a commonly studied NLP task of: “multi-hop reasoning”. Multi-hop reasoning is a task which may require the successful completion of many intermediate reasoning steps to arrive at the final answer [1]. Chain-of-though is a prompting strategy that can be employed with any LM task (including and not limited to: multi-hop reasoning). You use the term “n-hop” problem on line 179, but this is almost identical terminology to the multi-hop reasoning literature. It's fine to use the term since it's defined, but, like me, I suspect it may confuse readers who are doing a quick skim between your problem setting and the canonical multi-hop setting.
2. On lines 184-185 you use erroneous token-positions as proxies for identifying reasoning errors within the prompt completion. By this extension, in figure 2, is each reasoning hop number simply another erroneous token? If so, this is an overstatement of “reasoning hops” and rather should just be plainly labeled as erroneous token predictions. It makes more sense to predetermine all of the reasoning hops the model needs to make before identifying if there are errors at those positions; here you are using known errors to post-hoc identify reasoning hops? A related question: in figure 2, if you observe an error at a later reasoning hop number (e.g., later token position), was this after you corrected all previous errors at previous token positions, or is this a setting with cascading auto-regressive errors? If it is the autoregressive setting, this is not a surprising – models continue to make mistakes when fed erroneous prompts.
3. In response to lines 49-52, it seems like you are missing some related work which has studied this issue [1].
4. You discuss how you use the knockout mechanism to identify answer-writing heads in line 245-254, but it is not clear to me what the “answer” is here. Is it the expected-next-token immediately following the error token position you are analyzing? Is the answer the expected-next-token for the entire prompt? From formula 4, it looks like you are measuring the relative impact of an attention head on the vocabulary token associated with the erroneously predicted token t (shouldn’t you be measuring the impact on the correct/expected-next-token if you are looking to analyze the head’s relevance to the answer and not the error?
5. From line 258-259 it is  not clear to me what is Scorr  vs Serr? What are the correction and erroneous prediction sets? Are these the erroneously predicted tokens and anticipated correct tokens? What about the token selections is “random”?
6. You have a single competing baseline which you report (DoLa). DoLa is a decoding-based method, and your work is focused on activation interventions. I think your work could benefit from a direct comparison to LoFiT [2], or at least a discussion of what distinguishes both approaches.
7. Typos: Line 17-18: Do you mean “...systematic study {of} tasks…”

[1] Memory Injections: Correcting Multi-Hop Reasoning Failures during Inference in Transformer-Based Language Models. BlackboxNLP Workshop at EMNLP 2023.

[2] LOFIT: Localized Fine-tuning on LLM Representations. NeurIPS, 2024.

**Questions:**

1. On line 182 you formalize the n-hop problem, but in practice how do you decompose a CoT response into individual “hops”? Does a human have to go in and annotate all of the “hops”?
2. Along the lines of the previous question, how do you identify errors within reasoning chains as per the error-types outlined in Appendix E.1? For example on line 300 you write for the 50-hop parity-NL tasks, ~79% of errors are type 2; how did you calculate this? Did a human go and read through all of the reasoning chains for the 50-hop parity tasks and label errors?
3. Suggestion for clarity: You use an example to detail the errors you track for all tasks in Appendix E.1, but I would prefer a clear table that outlines the names of all of the tasks and the names of all of the corresponding error types in addition to the examples you provide.
4. In section 5, I am always left wondering if the premise of training a model (e.g., knockout classifier model) to intervene on a model (e.g., language model) is maybe a bit convoluted. Why not just train your language model further on the same dataset you used to train the knock-out model? How would the results compare on the same test dataset? Upon reading further it looks like your knockout classifier model is  Qwen2.5-0.5B w/ a classifier head…since this model is the same size as the models you are intervening on (E.g., table 1 models are all roughly 7-8B parameter counts), then it is not like you are saving compute cost by training a smaller classifier model; it is about the same size as the base model itself.
5. I appreciate that once the classifier is trained, the test-time intervention (TRC) is computationally efficient and light-weight. What is the cost of training the classifier as opposed to just training the base model itself?

[1] Memory Injections: Correcting Multi-Hop Reasoning Failures during Inference in Transformer-Based Language Models. BlackboxNLP Workshop at EMNLP 2023.

[2] LOFIT: Localized Fine-tuning on LLM Representations. NeurIPS, 2024.

---

> ### Author Response · Authors · 2025-11-23
> **Response to Reviewer C6Fu (1/7)**
>
> We sincerely appreciate the reviewer’s time, thorough review, insightful comments, and helpful suggestions on our work!
> **Below We detail our response to the weaknesses (i.e., W(1)$\sim$W(7)) and questions (i.e., Q(1)$\sim$Q(5)) point by point.**
>
> #### **(W1)** Conflation between what this paper dubs “reasoning hops generalization” and a commonly studied NLP task of: “multi-hop reasoning”.
>
> > Thanks very much for pointing this out!
> > To avoid confusion, we have **refined the definition** when the term “hop” first appears in the introduction by explicitly stating that “reasoning hop” in our paper refers to a **general reasoning step on a reasoning chain, which includes but not limits to the “hop” used in the multi-hop QA literature** [1] **as stated in line 53 of the revised paper.**
> > Our definition of “hop” applies broadly across reasoning tasks (e.g., arithmetic, symbolic manipulation, logical deduction), where each intermediate reasoning step constitutes a hop. This usage is aligned with several prior reasoning works (e.g., GoT [2]).
> >
> > - Specifically, we highlight the difference below:
> >
> >   - in the traditional multi-hop QA, a “n-hop” question means: the model retrieves n fact associations and then combine them to answer.
> >   - in our paper, “n-hop” question means: model perform a chain-of-thought reasoning with n intermediate reasoning steps, regardless of whether they involve factual association retrieval, symbolic manipulation, arithmetic or something.
> >
> >   Thus, while the terminology overlaps, the underlying concepts differ in scope and purpose.
> >
> > - Here we use two specific examples (in the Parity-NL task and the MDM task) to better explain the meaning of reasoning hop in our paper:
> >   - In the Parity-NL task, each reasoning hop corresponds to an event (e.g., Thomas flips the coin. [coin state heads up $\rightarrow$ tails up.]).
> >   - In the MDM task, supposing that we are calculating $384\times482349$, the first reasoning hop is to calculate $384\times9$ (ones-digit of $48234\underline{9}$), the second reasoning hop is to calculate $384\times4$ (tens-digit of $4823\underline{4}9$) and so forth.
> >
> > We sincerely thank the reviewer again for the very constructive comment!
> >
> > [1] Memory Injections: Correcting Multi-Hop Reasoning Failures during Inference in Transformer-Based Language Models. BlackboxNLP Workshop at EMNLP 2023.
> > [2] Graph of Thoughts: Solving Elaborate Problems with Large Language Models. AAAI 2024.

---

> ### Author Response · Authors · 2025-11-23
> **Response to Reviewer C6Fu (2/7)**
>
> #### **(W2)** In figure 2, is each reasoning hop number simply another erroneous token? Here you are using known errors to post-hoc identify reasoning hops? A related question: in figure 2, if you observe an error at a later reasoning hop number (e.g., later token position), was this after you corrected all previous errors at previous token positions, or is this a setting with cascading auto-regressive errors?
>
> > We appreciate the reviewer's thoughtful question. Below we clarify it point-by-point.
> >- **Reasoning hops are not post-hoc defined from observed errors.**
> > Rather, for each task we **predefine** all of the reasoning hops: **individual steps the model takes in its own step-by-step problem-solving process**. Please see specific examples in our response to (**W1**).
> > We then check the model's output **hop-by-hop** to identify **the first erroneously predicted token** and label the corresponding **error type**.
> >- **Interpretation of Figure 2.**
> >    - First, the x-axis coordinate value **“reasoning hop number”** refers to **the total number of the reasoning hops** that are used for solving the problem that we consider, **rather than refering to (the positions of) erroneous tokens**.
> >        - Taking the MDM task for an example: “reasoning hop number = $n$” simply means that we are considering the problems of calculating a $3$-digit number $\times$ a $n$-digit number (e.g., an instance of $n=6$ is $384\times482349$: hence we need to decompose $482349$ into 6 digits ($9, 4, 3, 2, 8, 4$) and calculate the product step-by-step.).
> >    - In Figure 2, we show how the model's **overall reasoning accuracy** and **the proportion of different error types** change **as the total number of reasoning hops needed to solve the question increases**. As stated in the line 190 $\sim$ 192 in the paper, for each problem that model fails to solve, we **only track the first error point** that appears in the model's reasoning trace. Hence this is **not the setting with cascading auto-regressive errors**.
> >- **How do we identify and label token-level erroneous predictions?**
> >    - For each **predefined reasoning hop**, we examine the model’s output at the corresponding output token positions and compare it against the ground truth for that hop, which is produced by a simulation program without needing additional human annotation. This allows us to determine **whether the reasoning hop is executed correctly**.
> >    - For example, in the Parity-NL task, if at the $k$-th reasoning hop the simulation program outputs “Thomas flips the coin. (The coin becomes tails up.)”, then in the LLM's $k$-th reasoning hop we check, in order:
> >       - if the person operating the coin in step $𝑘$ is “Thomas” (**if not $\rightarrow$ Error Type 2**);
> >       - whether the described action is “flips the coin” (**if not $\rightarrow$ Error Type 3**);
> >       - whether the resulting coin state is “tails up” (**if not $\rightarrow$ Error Type 5**).
> >
> >       Through this stepwise comparison, we identify **errorneous token-level predictions** and further assign the **corresponding error type**. Therefore, **“erroneous token” is observable we use to study different error types, not what we use to define a hop**.
>
> #### **(W3)** In response to lines 49-52, it seems like you are missing some related work which has studied this issue [1].
>
> > Thank the reviewer very much for brining this line of related work into our attention! **We have added the discussion on this line of work to the introduction of the revised paper (line 105 $\sim$ 107).** Below we would like to clarify the different scope that our work and the line of Memory Injections work focus on.
> > - The Memory Injections paper [1] and subsequent studies [2,3] indeed investigate multi-hop reasoning failures, but they focus on **latent multi-hop reasoning that occurs implicitly inside the model without explicit CoT generation**. Their objective is to analyze or patch internal reasoning traces during **non-CoT inference**.
> > - Our work, by contrast, specifically studies **explicit CoT reasoning**, where the model outputs a step-by-step chain of thought. The core problem we address, reasoning-hop generalization under explicit CoT supervision, is **mechanistically different from the latent-reasoning setting [4]**.
> > - we would like to investigate whether the mechanism insights gained from this work can be used for understanding and improving latent multi-hop reasoning in the future work.
> >
> > [1]: Memory Injections: Correcting Multi-Hop Reasoning Failures during Inference in Transformer-Based Language Models. BlackboxNLP Workshop at EMNLP 2023.
> > [2]: Understanding and Patching Compositional Reasoning in LLMs. ACL 2024 Findings.
> > [3]: Do Large Language Models Latently Perform Multi-hop Reasoning? ACL 2024.
> > [4]: Unveiling the Mechanisms of Explicit CoT Training: How CoT Enhances Reasoning Generalization. arXiv preprint 2025.

---

> ### Author Response · Authors · 2025-11-23
> **Response to Reviewer C6Fu (3/7)**
>
> #### **(W4)** You discuss how you use the knockout mechanism to identify answer-writing heads in line 245-254, but it is not clear to me what the “answer” is here. From formula 4, it looks like you are measuring the relative impact of an attention head on the vocabulary token associated with the erroneously predicted token t. Shouldn’t you be measuring the impact on the correct/expected-next-token if you are looking to analyze the head’s relevance to the answer and not the error?
>
> > Thanks for raising the question, and below we address it point-by-point.
> > - **What the “answer” is here?** In our paper, the **“answer”** in answer-writing heads refers to **the token that the model predicts at the critical token positions associated with the key error types** (as stated in line 221 $\sim$ 222 and 262 $\sim$ 263). Although these “answer” tokens all appear at positions corresponding to key error types, during the reasoning process the “answer” might be:
> >     - **an erroneously predicted token** (i.e., samples in the $\mathbf{S_{err}}$ set), or
> >     - **a correctly predicted token** (i.e., samples in the $\mathbf{S_{corr}}$ set)
> >
> >     (**as stated in line 262 $\sim$ 263 and 285 $\sim$ 287, and as illustrated in Figure 3 in the revised paper**).
> >      To make this clear, we take specific instances of $S_{err}$ and $S_{corr}$ from the MDM task to explain the meaning of the **“answer”**:
> >     - (**$\mathbf{S_{err}}$, erroneous prediction**) input problem: $384\times482349=?$
> >         - The **expected reasoning process** is that the model should sequentially multiply $384$ and $9$(unit-digit), $4$(tens-digit), $3$(hundreds-digit), $2$(thousands-digit), $8$(ten-thousands digit), $4$(hundred-thousands digit).
> >         - **However, the model only correctly outputs the first two steps ($384\times9$ and $384\times4$), but at the third step incorrectly infers that the hundreds digit of the multiplier is $8$ instead of $3$, i.e., it predicts $384\times\underline{\mathbf{8}}$ whereas the correct prediction should be $384\times\underline{\mathbf{3}}$.**
> >         - This constitutes a sample in $S_{err}$. For this case, we **concatenate** the input question with the model’s response up to the critical error position (i.e., before token $\underline{8}$) as the model **input**, and treat the erroneously predicted token $\underline{8}$ as the **output**.
> >
> >        **We use $\mathbf{S_{err}}$ to analyze which heads are responsible for writing down the incorrect answer in this input–output mapping.**
> >     - (**$\mathbf{S_{corr}}$, correct prediction**) input problem: $119\times54321=?$
> >         - The expected reasoning process is that the model should sequentially multiply $119$ and $1$(unit-digit), $2$(tens-digit), $3$(hundreds-digit), $4$(thousands-digit), $5$(ten-thousands digit).
> >         - **The model predicts all steps correctly up to the fourth step and outputs $119\times\underline{\mathbf{4}}$ in the fourth step.**
> >         - This gives us a possible sample in $S_{corr}$. We then **concatenate** the input problem with the model’s response up to this critical token position (i.e., before token 4) as the **input**, and treat the correctly predicted token 4 as the **output**.
> >
> >       **$\mathbf{S_{corr}}$ allows us to study which heads are responsible for writing down the correct answer in this input–output mapping**.
> > - **Interpretation of Formula 4.** Accordingly, in Formula 4, the token $t$ we measure represents **the token that the model actually predicts**. For samples in $S_{err}$, **$t$ refers to the erroneously predicted tokens**. For samples in $S_{corr}$, **$t$ refers to the correctly predicted tokens**.
> >     - We include both correct and incorrect predictions because our goal is to conduct a comparative analysis (**as illustrated in the Figure 3 in the revised paper**) to examine, at these critical token positions—positions at which models frequently make reasoning mistakes—how the mechanisms responsible for producing the model’s predictions (i.e., the answer-writing heads) behave in cases of correct prediction ($S_{corr}$) versus erroneous prediction ($S_{err}$).
> >     - As shown in Figure 4(a,b), we find that **correct and incorrect predictions largely rely on the sharing set of answer-writing heads**, and that these heads simultaneously encode both correct and incorrect reasoning trajectories.
> >
> > We appreciate the reviewer’s question and have **included a new figure (Figure 3) in the revised paper** to detailedly illustrate the meaning of “answer” in $\mathbf{S_{corr}}$ and $\mathbf{S_{err}}$.

---

> ### Author Response · Authors · 2025-11-23
> **Response to Reviewer C6Fu (4/7)**
>
> #### **(W5)** From line 258-259 it is not clear to me what is Scorr vs Serr? What are the correction and erroneous prediction sets? Are these the erroneously predicted tokens and anticipated correct tokens? What about the token selections is “random”?
>
> > We sincerely thank the reviewer for raising the questions! **We have included a new figure (Figure 3) in the revised paper to help illustrate the meaning of $\mathbf{S_{\text{corr}}}$ and $\mathbf{S_{\text{err}}}$**, and discussion on the **comparison experiment of using randomly selected token to conduct the locating experiment in Appendix F.3, Figure 13.**
> > - **The definition of $S_{\text{err}}$ and $S_{\text{corr}}$.**
> > $S_{\text{err}}$ and $S_{\text{corr}}$ are two sampled sets for correction and erroneous predictions at the corresponding key error-type token position.
> > These correction and erroneous predictions are both **predicted tokens output by the model** (**not the erroneously predicted token and the anticipated correct one**) under different input reasoning contexts.
> >     - **Please refer to our response to (**W4**) for specific instances of $S_{\text{err}}$ and $S_{\text{corr}}$ from the MDM task.**
> > - **What about the token selection is “random”?**
> >  We have conducted the **comparison experiment** of locating answer-writing heads for **10 randomly selected tokens**.
> >  **The average localization results are shown in Figure 13 in the revised paper.**
> >  Unlike the original localization result, there is **no attention heads with significant knock out effect** (the maximum of the average intervention effect is around 0.06, in constrast to the result shown in Figure 4 of the paper), which further confirms **the effectiveness of the localization result obtained with model's predicted token** ($S_{\text{err}}$ and $S_{\text{corr}}$). Please refer to the detailed results in the (Appendix F.3, Figure13) in the revised paper.

---

> ### Author Response · Authors · 2025-11-23
> **Response to Reviewer C6Fu (5/7)**
>
> #### **(W6)** You have a single competing baseline which you report (DoLa). DoLa is a decoding-based method, and your work is focused on activation interventions. I think your work could benefit from a direct comparison to LoFiT [2], or at least a discussion of what distinguishes both approaches.
> #### [2] LOFIT: Localized Fine-tuning on LLM Representations. NeurIPS, 2024.
>
>
> > Thank the reviewer for this very constructive suggestion! **We have added the experiments and discussion of comparing TCR and LOFIT on improving reasoning hop generalization to Appendix G.5 in the revised paper**, following the reviewer's suggestion.
> > - LOFIT indeed shares the idea of intervening on attention-head representations, but it **differs** from TCR in two important ways.
> >     - **Fine-tuning to adapt the down-stream task vs. Test-time intervention to correct reasoning**:
> >         - LOFIT is primarily introduced as a lightweight alternative to PEFT methods (e.g., LoRA), and is typically applied to **task-specific fine-tuning** (e.g., TruthfulQA, multi-hop QA). It does **not particularly target reasoning-hop generalization.**
> >         - TCR is a **test-time attention head intervention** method grounded on the **competition mechanism behind the reasoning hop generalization phenomenon** that is studied in our paper, which suppresses the erroneous processing head to correct reasoning.
> >     - **Task specificity vs. Task generality**:
> >         - LOFIT identifies **task-dependent attention heads** and therefore tends to be tied to the specific downstream task used for fine-tuning, making out-of-distribution task generalization difficult.
> >         - In contrast, TCR identifies erroneous processing heads with stable competitive dynamics across tasks, enabling effective transfer.
> > - Following the reviewer’s suggestion, we make a direct comparison between TCR and LOFIT. We evaluated two models (Qwen2.5-7B-Instruct and Phi-3-Instruct), and considered the following two settings:
> >     - **LOFIT fine-tuned separately on two tasks**: (LOFIT(N) means that using N-hop problem as the training data of LOFIT.)
> >     (1) Comparing LOFIT and TCR on Parity-NL (50-hop).
> >         | **Base Model**      | **original** | **LOFIT(10)** | **LOFIT(20)** | **LOFIT(30)** | **LOFIT(40)** | **LOFIT(50)** | **TCR**   | **TCR-gold** |
> >         | ------------------- | ------------ | ------------- | ------------- | ------------- | ------------- | ------------- | --------- | ------------ |
> >         | Qwen2.5-7B-Instruct | 48.3%        | 49.8%         | 50.5%         | 52.5%         | 53.3%         | 55.8%         | **60.4%** | **81.2%**    |
> >         | Phi-3-Instruct      | 51.2%        | 50.2%         | 49.5%         | 50.8%         | 51.8%         | 53.1%         | **55.8%** | **65.2%**    |
> >
> >         (2) Comparing LOFIT and TCR on ObjC (30-hop).
> >         | **Base Model**      | **original** | **LOFIT(10)** | **LOFIT(20)** | **LOFIT(30)** | **TCR**   | **TCR-gold** |
> >         | ------------------- | ------------ | ------------- | ------------- | ------------- | --------- | ------------ |
> >         | Qwen2.5-7B-Instruct | 52.0%        | 44.5%         | 50.2%         | 54.0%         | **56.0%** | **76.0%**    |
> >         | Phi-3-Instruct      | 30.9%        | 32.8%         | 35.8%         | 38.3%         | **45.0%** | **45.0%**    |
> >
> >
> >     - **LOFIT jointly fine-tuned on five tasks (Parity-NL, MDM, LLC, CLF, MOAS), while testing out-of-distribution task generalization on two unseen tasks (ObjC and NumS), consistent with our main experiment setup (Table 2).**
> >
> >         | **Qwen2.5-7B-Instruct** | original | LOFIT | TCR       | TCR-gold  |
> >         | ----------------------- | -------- | ----- | --------- | --------- |
> >         | ObjC                    | 52.0%    | 53.8% | **56.0%** | **76.0%** |
> >         | NumS                    | 41.1%    | 40.3% | **46.0%** | **54.5%** |
> >         | Average                 | 46.6%    | 47.1% | **51.0%** | **65.3%** |
> >
> >
> >         | **Phi-3-Instruct** | original | LOFIT | TCR       | TCR-gold  |
> >         | ------------------ | -------- | ----- | --------- | --------- |
> >         | ObjC               | 30.9%    | 33.2% | **45.0%** | **45.0%** |
> >         | NumS               | 72.0%    | 68.5% | **76.0%** | **82.0%** |
> >         | Average            | 51.5%    | 50.9% | **60.5%** | **63.5%** |
> >
> >
> >     The results show that: **TCR yields substantial and consistent improvements on reasoning-hop generalization over LOFIT** on Parity-NL and Object Counting;
> >     LOFIT’s improvements are much more **task-specific**, and it **fails to achieve effectiveness out-of-distribution task generalization** observed with TCR.
>
> #### **(W7)** Typos: Line 17-18: Do you mean “...systematic study {of} tasks…”
>
> > Thank the reviewer for pointing this out! We have corrected it in the revised paper.

---

> ### Author Response · Authors · 2025-11-23
> **Response to C6Fu (6/7)**
>
> #### **(Q1)** On line 182 you formalize the n-hop problem, but in practice how do you decompose a CoT response into individual “hops”? Does a human have to go in and annotate all of the “hops”?
>
> > Thank the reviewer for raising the question.
> > **No human annotation is required to decompose a CoT response into individual hops.**
> > In practice, we simply treat **each individual reasoning step in the model-generated CoT** as one hop, consistent with the formal definition in the paper. The hop boundaries are therefore directly induced from the model’s own step-by-step solution and do not rely on any manual labeling. Representative examples are provided in Appendix D.1.
> > **For example:**
> > - **Multi-digit Multiplication (MDM)**
> >     When solving $325\times284349$, the model typically produces a sequence of partial-product computations.
> >     - Hop 1: compute $325 \times 9$ (units digit), then shift.
> >     - Hop 2: compute $325 \times 4$ (tens digit), then shift.
> >     - ... ...
> >
> >     Each paritial-product computation forms one reasoning hop.
> > - **Parity-NL**
> >     The model simulates each person's action in sequence.
> >     - Hop 1: Jack flips the coin. (The coin becomes tails up.)
> >     - Hop 2: Peter doesn't flip the coin. (The coin remains tails up.)
> >     - Hop 3: Daniel flips the coin. (The coin becomes heads up.)
> >     - ... ...
> >
> >     Each state update constitutes one hop, and the final parity is determined after the last hop.
> >
> > In all tasks, a hop corresponds to a single step in the model’s reasoning chain, and we extract these steps directly from its generated CoT without any additional human intervention.
>
> #### **(Q2)** Along the lines of the previous question, how do you identify errors within reasoning chains as per the error types outlined in Appendix E.1? For example on line 300 you write for the 50-hop parity-NL tasks, $\sim$ 79% of errors are type 2; how did you calculate this? Did a human go and read through all of the reasoning chains for the 50-hop parity tasks and label errors?
>
> > We would like to thank the reviewer for raising the question.
> >
> > **Similar to (Q1), no human annotation is involved in identifying errors or labeling the error types.** All error detection and categorization in Appendix E.1 are performed automatically by **comparing each model-produced reasoning step with a task-specific Chain-of-Thought reasoning simulator** (i.e., a computer program which automatically generates the ground-truth CoT reasoning trace).
> >
> > Our pipeline is as follows:
> > 1. Obtain the model’s **full CoT response** for a given input.
> > 2. **Segment the response into hops** according to the model’s own step-by-step reasoning (as described in Q1).
> > 3. For each hop $k$, **compare the model’s generated reasoning hop with the simulator’s ground-truth reasoning hop**.
> > For example, in the Parity-NL task, the simulator specifies for hop $k$:
> > **“Jack flips the coin. (The coin becomes tails up.)”**
> > The model’s hop $k$ is then automatically checked:
> >     - if the acting person is **Jack** (if not $\rightarrow$ **Error Type 2**),
> >     - if the action is **flips the coin** (if not $\rightarrow$ **Error Type 3**),
> >     - if the resulting coin state is **tails up** (if not $\rightarrow$ **Error Types 4/5**).
> >
> >     Any mismatch determines both (i) **whether an errorneous prediction occurs** and (ii) **the corresponding error type**. For each instance, we only record the firstly occurred error for analysis, which distinguishes from the “cascading auto-regressive setting” mentioned by the reviewer in (W2).
> >
> > Using this **fully automated procedure** across all 50-hop Parity-NL instances, we can directly compute the proportion of each error type: for example, **that $\sim$ 79% of errors fall into error type 2 (i.e., error for predicting the wrong acting person, labeled by our automated checking program)**, without any human reading or labeling of the reasoning chains.
>
> #### **(Q3)** Suggestion for clarity: You use an example to detail the errors you track for all tasks in Appendix E.1, but I would prefer a clear table that outlines the names of all of the tasks and the names of all of the corresponding error types in addition to the examples you provide.
>
> > Thank the reviewer for the helpful suggestion!
> > **We have added a new summary table (Table 3) that explicitly lists all tasks together with the names of all possible error types in Appendix E.1, following the reviewer's suggestion.**
> > We agree that this consolidated table improves clarity and makes the error taxonomy easier to follow. Thanks again for this constructive comment.

---

> ### Author Response · Authors · 2025-11-23
> **Response to Reviewer C6Fu (7/7)**
>
> #### **(Q4 and Q5)** How would the **results and training cost** of (training the classifier) and (training the base model itself) compare on the same training and test dataset?
>
> > Thank the reviewer for the insightful comment and question!
> Below we discuss (1) **why we do not simply train the base model itself**, (2) **how different implementations of “fine-tuning the base model itself” compare to the orgininal TCR**, and (3) **comparison of training cost**.
> > **We have included the discussion and the experiment result in the revised paper (Appendix G.6).**
> > - **Why not directly fine-tune the base language model?**
> > TCR is conceptually different from directly **fine-tuning the base model in the training time**:
> >     - Fine-tuning changes the model’s parameters to adapt the training distribution.
> >     - TCR aims to identify and suppress erroneous head representations at test time, without modifying models' original parameters.
> >
> >     Thus, TCR is designed to be **non-destructive** and **generalizable to new tasks**, while direct fine-tuning **inherently entangles model parameters with the training distribution**.
> >
> > - **Why use a 0.5B knockout-classifier model rather than the 7B-8B base model?**
> >     - Qwen2.5-0.5B-Instruct with a classifier head ($\sim$ 0.5B parameters) is **14× smaller** than the 7–8B base models, which results in much **lower training cost** and **faster inference** (when triggering TCR).
> >
> > - **Comparing reasoning hop generalization performance of directly training the base model itself with the proposed TCR.**
> > Following the reviewer's suggestion, we **train the base model** (e.g., Qwen2.5-7B-Instruct) **using the same dataset** employed for the knockout classifier, under three different schemes (i.e., Exp1, 2, 3):
> >     | Experiment Group    | Parity-NL | MDM       | LLC       | CLF       | MOAS      | ObjC      | NumS      | Average            |
> >     | ------------------- | --------- | --------- | --------- | --------- | --------- | --------- | --------- | ------------------ |
> >     | Qwen2.5-7B-Instruct | 48.3%     | 43.0%     | 11.7%     | 56.8%     | 39.2%     | 52.0%     | 41.1%     | 41.7%              |
> >     | Exp1                | 49.5%     | 29.5%     | 10.2%     | 48.3%     | 30.2%     | 45.8%     | 28.0%     | 34.5% (-7.2%)      |
> >     | Exp2                | 50.3%     | 45.0%     | 13.5%     | 63.6%     | 44.8%     | 52.6%     | 41.9%     | 44.5% (+2.8%)      |
> >     | Exp3                | 56.2%     | 43.7%     | 12.2%     | 59.5%     | 44.0%     | 52.0%     | 39.8%     | 43.9% (+2.2%)      |
> >     | TCR                 | **60.4%** | **48.2%** | **16.2%** | **66.6%** | **46.0%** | **56.0%** | **46.0%** | **48.5%** (+6.8%)  |
> >     | TCR-gold            | **81.2%** | **58.3%** | **23.0%** | **71.3%** | **62.0%** | **76.0%** | **54.5%** | **61.3%** (+19.6%) |
> >
> >
> >     - **Exp1: trainable base model + trainable classifier head.**
> >     We fine-tune both the base model and the classifier head for the head classification task.
> >     - **Exp2: frozen base model + trainable classifier head.**
> >     We freeze the base model itself and only train the classifier head for the head classification task.
> >     - **Exp3: fine-tuning the base model on ground-truth tokens directly.**
> >     We do not use an additional classifier head and directly fine-tune the base model itself to correct the erroneous predictions.
> > - **Comparing the training cost.**
> > Below is the comparison between the cost of training the light-weight classifier and training the base model itself (Exp 1,2,3).
> >   | Experiment                                             | training FLOPs     | GPU memory | trainable parameters |
> >   | ------------------------------------------------------ | ------------------ | ---------- | -------------------- |
> >   | training the 0.5B classifier (Ours)                     | $1.9\times10^{17}$ | 9.6GB      | 1.08M                |
> >   | Exp1      | $3.6\times10^{18}$ | 64.7GB     | 5.09M                |
> >   | Exp2 | $9.5\times10^{17}$ | 37.9GB     | 0.04M                |
> >   | Exp3                           | $4.0\times10^{18}$ | 69.4GB     | 5.05M                |
> >
> > **Conclusion:**
> > Across all three schemes, comparing to the original TCR (training the 0.5B classifier), training the base model itself leads to **catastrophic forgetting**[1] (Exp1), **insufficient expressivity for classification** (Exp2), **poor in-distribution and out-of-distribution task generalization** (Exp1,2,3), and **higher training cost** (Exp1,2,3).
> >
> > [1] Retaining by Doing: The Role of On-Policy Data in Mitigating Forgetting, arXiv preprint 2025.
>
> We sincerely thank the reviewer again for the thorough review, helpful feedback and constructive suggestions! Please let us know if there are any remaining concerns. We would be happy to discuss them further.

---

> ### Comment · Reviewer_C6Fu · 2025-11-24
>
> Thank you for clarifying. It seems I was confused w.r.t. the vocabulary based on similar language in tangential work. I appreciate the additional experiments. My concerns have been addressed. I have raised my score.

---

> > ### Author Response · Authors · 2025-11-25
> > **Thanks for Reviewer C6Fu's prompt response**
> >
> > Dear Reviewer C6Fu,
> >
> > We are very happy to know that our previous response has addressed your concerns. We also sincerely appreciate that the reviewer can increase the score taking it into consideration. Thanks again for your time and very constructive comments and helpful suggestions, which have improved the clarity and quality of our paper.
> >
> > Submission#11334 Authors

---

### Official Review · Reviewer_VfVH · 2025-10-29

**Soundness:** 3
**Presentation:** 3
**Contribution:** 3
**Rating:** 6
**Confidence:** 4

**Summary:**

This paper investigates why LLMs fail to generalize in multi-hop reasoning tasks, finding that errors are concentrated in the several error types and dominated by few attention heads. Using residual stream analysis and attention head interventions, the authors uncover a competition mechanism between correct and erroneous reasoning signals within the model. Based on these insights, they propose Test-time Correction of Reasoning (TCR), a lightweight intervention that dynamically deactivates erroneous heads to significantly improve hop generalization performance.

**Strengths:**

1. I think overall this is a good paper; it not only provides a mechanistic analysis but also proposes an improved algorithm based on these insights.

2. The paper is presented in a clear and logical manner, making it easy to follow and understand.

**Weaknesses:**

1. Although the TCR method effectively improves performance, considering that it requires prior attention-head selection and training an additional small model, I am unsure whether this workflow would be computationally acceptable in practice.

2. I think reporting additional metrics on the consistency of model outputs after applying TCR would strengthen the paper. For example, showing whether the predictive entropy is effectively reduced could further support the method’s effectiveness.

3. I noticed that Equation (4) uses normalization to represent relative change trends. One concern is that the final averaged results may be overly influenced by cases where the denominator is small. For example, if the change $f(h) - f(h-a) = 0.2$, the resulting $s(a)$ will be 100% if $f(h) = 0.2$ and only 0.25% if $f(h) = 0.8$. I am unsure whether removing this normalization would affect the final head localization. Reporting the change $f(h) - f(a)$ or the distributions of $f(h)$ and $f(h) - f(a)$ would be somewhat helpful.

4. In locating the answer-writing heads, the paper uses the Logit Lens interpretability tool (mentioned in Section 2). I understand that this is a common approach in interpretability studies. However, different layers and heads in the model may operate in entirely different semantic spaces, which do not necessarily align with the semantic space of the final unembedding matrix. Therefore, this mapping may not fully carry the intended meaning. I think a discussion of the limitations and some justification for its use would be helpful.

5. While the logic of the content is clear, Section 4 is very dense in text. It might be helpful to simplify it or break it into smaller subsections, making it easier for readers to follow.

**Questions:**

see Weakness.

---

> ### Author Response · Authors · 2025-11-23
> **Response to Reviewer VfVH (1/3)**
>
> We sincerely thank the reviewer for the time, thorough review, constructive comments and suggestions on our work!
> **Below We detail our response to the weaknesses (i.e., W(1)$\sim$W(5)) point by point.**
>
> #### **(W1)** Although the TCR method effectively improves performance, I am unsure whether this workflow would be computationally acceptable in practice.
>
> > (W1) Thank the reviewer very much for raising this thoughtful concern. We would like to clarify that the overall computation introduced by TCR workflow is practically acceptable.
> > To this end, We break down the costs below.
> > - **Prior attention-head selection.**
> > TCR indeed requires identifying task-specific correct/erroneous processing heads. However, this procedure is very lightweight: for each task, only 10-shot examples are sufficient to reliably determine the relevant heads withnot requiring any model training.
> >
> > - **Training the lightweight classifier.**
> > The remaining overhead comes from preparing a classification training set and fine-tuning a small language model to perform the head selection task.
> >     - **Training data generation:**
> >     For our experiments, we generate approximately 4,000 training instances per in-distribution task (20,000 instances in total). For each instance, we sweep over each head in $\mathbf{H}$ ($|\mathbf{H}|\approx10$) and label it, which is also fast since the base model is frozen and we can process the data batch by batch.
> >     - **Training a small language model as the classifier**：
> >     We fine-tune a **Qwen2.5-0.5B-Instruct** model as the classifier. We adopt **LoRA** (on the $W_q,W_k,W_v,W_o$ parameters in the self-attention modules) and **16-bit quantization** in the fine-tuning process, which only contains about $1.08$M trainable parameters and take about $9.6$ GB GPU Memory and $1.9\times10^{17}$ training FLOPs. This procedure takes around 2–3 GPU hours on a single NVIDIA A100 (80GB) GPU in our implementation. This is negligible compared to training or fine-tuning the base model.
> >
> > **Moreover, TCR can generalize to new tasks without additional overhead.**
> > A key advantage of TCR is that the trained classifier (head selector) **generalizes to new tasks**. As shown in the paper (Table 1 and Table 2), After training on five in-distribution tasks (Parity-NL, MDM, LLC, CLF, MOAS), the classifier successfully generalizes to unseen tasks such as ObjC, NumS (the result is shown in the Table 2 in the paper), and BigGSM[1] (a variant of GSM-8K that requires more reasoning hops, the result is shown in the table below). This means **the entire workflow involves a one-time preprocessing cost, rather than per-task overhead, which further improves practical viability**.
> > |       Big-GSM       | Base Model |       TCR      |
> > |:-------------------:|:----------:|:--------------:|
> > | Qwen2.5-7B-Instruct | 52.7%±0.4% | **54.4%**±0.5% |
> > |    Phi-3-Instruct   | 52.5%±1.3% | **53.9%**±0.7% |
> >
> > In summary, although TCR introduces several additional steps beyond standard inference, the overall computational overhead is **acceptable** and **one-off**. We sincerely thank the reivewer for the very insightful comment and we have clarified these details in the revised version.
> >
> > [1] Unlocking the Capabilities of Thought: A Reasoning Boundary Framework to Quantify and Optimize Chain-of-Thought. NeurIPS 2024.
>
> #### **(W2)** Showing whether the predictive entropy is effectively reduced could further support the method’s effectiveness.
>
> > We thank the reviewer very much for the insightful suggestion. **Following the reviewer's suggestion, we have added an analysis of the model’s predictive entropy before and after applying TCR to Appendix F.1 Table 5 in the revised paper.**
> >
> > As shown in the table below (the “erroneous predictions” row and the “intervened erroneous predictions” row), **TCR consistently reduces predictive entropy of originally erroneous predictions across tasks** (by average 63.2%) with Qwen2.5-7B-Instruct (results with other base models can be found in the paper, Appendix F.1 Table 5).
> This finding provides additional evidence supporting the effectiveness of our method.
> >
> > | Qwen2.5-7B | Parity-NL | MDM | LLC | CLF | MOAS | ObjC | NumS | Average |
> > | :---: | :---: | :---: | :---: | :---: | :---: | :---: | :---: | :---: |
> > | **correct predictions** | 0.000 | 0.000 | 0.035 | 0.000 | 0.000 | 0.000 | 0.006 | 0.006 |
> > | **erroneous predictions** | 0.097 | 0.200 | 0.230 | 0.241 | 0.178 | 0.192 | 0.073 | 0.173 |
> > | **intervened erroneous predictions** | 0.022 | 0.082 | 0.133 | 0.098 | 0.038 | 0.050 | 0.035 | **0.065** |
> > | **reduce** | 77.3% | 59.0% | 42.2% | 59.3% | 78.7% | 74.0% | 52.1% | **63.2%** |
> >
> > Again, we sincerely appreciate the reviewer's valuable suggestion.

---

> ### Author Response · Authors · 2025-11-23
> **Response to Reviewer VfVH (2/3)**
>
> #### **(W3)** I noticed that Equation (4) uses normalization to represent relative change trends. One concern is that the final averaged results may be overly influenced by cases where the denominator is small. Reporting the change $f(h)-f(h-a)$ or the distributions of $f(h)-f(h-a)$ and $f(h)$ would be somewhat helpful.
>
> > (W3) Thank the reviewer for the insightful comment and suggestion!
> > We fully understand the concern that small denominators could potentially bias the normalized metric in Equation (4).
> > In fact, **we conducted an additional analysis that removes the normalization term and uses the raw change $f(h)-f(h-a)$. The corresponding results are provided in Appendix F.3 and Figures 12(e,f).**
> > Below we detailedly discuss the reason why we use the $f(h)$ to normalize the intervention effect.
> >
> > - **Effect of removing normalization.**
> > When using the unnormalized metric $f(h)-f(h-a)$ (i.e., the results in Appendix F.3, Figure 12(e,f)), **the identified heads fall almost entirely into the final 2 $\sim$ 3 layers**. However, this starkly **contradicts the layer-wise distribution of answer-related information shown in Figure 3c and Figure 12(i)**, where the information of the answer-related token emerges and outstands around layers 20–26 rather than only in the final 2 $\sim$ 3 layers. **Note that after applying normalization, the localization (Figure 12(g,h), our method) aligns well with Figure 12(i).**
> > - **Why identified heads fall into final layers without normalization?**
> > As discussed in the main text (line 268 $\sim$ 269) and Appendix F.3, **the scale of the probabilities of predicted-answer-related tokens varies drastically across layers**: deeper layers exhibit much larger probability shifts due to the well-known calibration bias in deep LLMs, where earlier layers operate in a low-confidence regime (**probabilities fluctuate only within a narrow band in earlier layers**) and the final layers sharply increase the predicted-token confidence [1] (**probabilities surge dramatically in the final layers**).
> > Because of this inherent scale difference, **the raw change $f(h)-f(h-a)$ disproportionately amplifies effects in the deepest layers.**
> > - **Why normalization is necessary.**
> > Normalizing by $f(h)$ compensates for these cross-layer scale disparities and prevents late-layer dominance. This allows the method to correctly identify answer-writing heads in earlier layers, heads that play an essential role in accumulating task, relevant information but would otherwise be overshadowed by the large-scale shifts in the final layers.
> >
> > Overall, **normalization is crucial for obtaining faithful and stable head localization**. We sincerely appreciate the reviewer's valuable suggestion and have included and highlighted this discussion in the Appendix F.3 in the revised paper.
> >
> > [1] Calibration Across Layers: Understanding Calibration Evolution in LLMs. EMNLP 2025.

---

> ### Author Response · Authors · 2025-11-23
> **Response to Reviewer VfVH (3/3)**
>
> #### **(W4)** I think a discussion of the limitations and some justification for the use of Logit Lens would be helpful.
>
> > (W4) Thank the reviewer for raising this important point and very valuable suggestion. While Logit Lens is widely used in recent interpretability works [3,4,5], its limitations [1,2] indeed warrant explicit discussion.
> > - **Discussion on the limitations of Logit Lens**
> > Following the reviewer's suggestion, **we have added a dedicated discussion in the Limitations and Discussion Section (Appendix B)** regarding the semantic-space mismatch issue of Logit Lens and the potential risks of over-interpreting its decoded distributions.
> > - **Justification of the localization of answer-writing heads**
> > Our identification of answer-writing heads is supported through **two complementary analyses**:
> >     - **(Analysis 1, Figure 4c):** we show the layer-wise evolution of answer-related information, where the information of answer-related tokens emerge in layer 20 $\sim$ 26. This result **validates the answer-writing heads localization result from the perspective of layer distribution.**
> >     - **(Analysis 2, Figure 5):** we analyze the internal distributions contained in the identified answer-writing heads and how these change when erroneous processing heads in the early-to-mid layers are intervened. This analysis **validates the answer-writing heads localization result from the internal information perspective.**
> >
> >     Although these additional analysis still involving  decoding hidden representation with the unembedding matrix to some extent, they are characterizing answer-writing heads from different perspectives and collectively provide **strong cross-validation**.
> > - **Future work and alternative tools**
> > We also agree that more robust alternatives could help mitigate semantic misalignment. In future work, we plan to explore Tuned Lens [1] and Patchscopes [2], which are designed to reduce space-misalignment issues when interpreting intermediate representations.
> >
> > We appreciate your insightful suggestion and believe the additional discussion would be very helpful.
> >
> > [1] Eliciting Latent Predictions from Transformers with the Tuned Lens, arXiv preprint 2023.
> > [2] Patchscopes: A Unifying Framework for Inspecting Hidden Representations of Language Models, ICML 2024.
> > [3] Dissecting Recall of Factual Associations in Auto-Regressive Language Models, EMNLP 2023.
> > [4] Interpretability in the Wild: a Circuit for Indirect Object Identification in GPT-2 small, ICLR 2023.
> > [5] Knowledge Circuits in Pretrained Transformers, NeurIPS 2024.
>
> #### **(W5)** While the logic of the content is clear, Section 4 is very dense in text. It might be helpful to simplify it or break it into smaller subsections, making it easier for readers to follow.
>
> > (W5) Thank the reviewer very much for recognizing the clear logic of the content and the helpful suggestion.
> > We agree that the original Section 4 was overly dense due to the page-limit constraints at submission time.
> **Following your feedback, we have revised this section by restructuring it into smaller, clearer subsections.**
> > - Now the hierarchical organization of Section is as follows:
> >     - Overview of Section 4
> >         - introducing the idea of comparative study.
> >         - briefly introducing the competition mechanism.
> >     - Section 4.1: The Mechanism of Answer-Writing Heads
> >         - paragraph: locating answer-writing heads.
> >         - paragraph: inspecting answer-writing heads.
> >     - Section 4.2: The Mechanism of Processing Heads
> >         - paragraph: locating processing heads.
> >         - paragraph: knocking out an erroneous processing head can rectify the erroneous predictions.
> >         - paragraph: further discussion on the insights behind the competition mechanism.
> >
> > We believe these changes substantially improve readability and make the flow of the argument easier to follow. We sincerely appreciate the reviewer's constructive comment.
>
> We would like to thank the reviewer once again for the thorough review, thoughtful feedback and helpful suggestions! If there are any remaining concerns, we are glad to further clarify or discuss them.

---

> ### Author Response · Authors · 2025-11-27
> **We would like to hear from Reviewer VfVH**
>
> Hi Reviewer VfVH,
>
> We sincerely appreciate your dedicated effort in reviewing our manuscript. As the rebuttal is coming to a close, we kindly remind you that we have submitted a response to your comments. We would be very grateful if you could confirm whether our responses have addressed your concerns? If you have any additional concerns, please don't hesitate to let us know. We would really appreciate the opportunity to discuss with you further.
>
> Thank you again!
>
> Submission#11334 Authors

---

### Official Review · Reviewer_iLuH · 2025-11-01

**Soundness:** 3
**Presentation:** 3
**Contribution:** 3
**Rating:** 6
**Confidence:** 3

**Summary:**

Recent studies reveal a sharp performance drop in reasoning hop generalization scenarios. To understand these failures, the authors first investigate two research questions: 1) Where do errors occur? 2) Why do these errors arise? They found that these failures concentrate on token positions of a few key error types, and the error trajectories can suppress the correct ones. To address this issue, the authors proposed Test-time Correction of Reasoning (TCR), which trains a classifier to dynamically identify and deactivate the erroneous heads during inference. Experimental results prove the effectiveness of the proposed method, TCR.

**Strengths:**

1. The paper investigates the reasoning hop generalization problem,  and provides valuable insights to understand where and why the problem happens. The findings may contribute to the recent reasoning research community.
2. Based on their findings, the authors propose an effective solution.

**Weaknesses:**

1. The proposed method needsto  train a classifier for tasks. Although the results achieve some ood performance. However, there are still marginal gaps between id and ood.
2. Lack experiments on larger models, like 13B, 32B. Only 7B is not convincing enough.

**Questions:**

1. The classifier is an interface to LLMs. Can we directly improve the LLM's ability by teaching it to stop searching the error trajectories?
2. See weakness 2. How does the method perform on large models?

---

> ### Author Response · Authors · 2025-11-23
> **Response to Reviewer iLuH (1/2)**
>
> We sincerely appreciate the reviewer’s time, thorough review, and very insightful feedback on our work!
> **Below We detail our response to the weaknesses (i.e., W(1)$\sim$W(2)) and questions (i.e., Q(1)$\sim$Q(2)) point by point.**
>
> #### **(W1)** The proposed method needsto train a classifier for tasks. Although the results achieve some ood performance. However, there are still marginal gaps between id and ood.
>
> > We thank the reviewer for the very thoughtful comment.
> > - The classifier is evaluated on five in-distribution (ID) tasks (Parity-NL, MDM, LLC, CLF, and MOAS) and two out-of-distribution (OOD) tasks (ObjC and NumS). As shown in Table 1, although the Hit@1 accuracy on OOD tasks is lower than on ID tasks, the classifier still achieves 57.3% accuracy on average across models in the OOD setting, which is **substantially higher than the random-guessing baseline** (26.0% on average). This indicates that **the classifier indeed captures transferable task-related signals and generalizes beyond the tasks it is trained on**.
> > - The observed gap between ID and OOD performance is **expected** due to the natural semantic and structural differences between training and unseen tasks, which is common in out-of-distribution task generalization of inference-time intervention methods[1,2].
> > - We agree that further reducing this gap is an important direction. **Expanding the in-distribution task set to include more diverse task types** is a straightforward way to further enhance cross-task generalization.
> >
> > Again, we appreciate the reviewer's thoughtful comment and will discuss this generalization gap and potential solutions in the revised paper (Appendix B: Limitations and Discussion section).
> >
> > [1] Inference-Time Intervention: Eliciting Truthful Answers from a Language Model, NeurIPS 2023.
> > [2] Non-Linear Inference Time Intervention: Improving LLM Truthfulness, Interspeech 2024.
>
> #### **(W2)** Lack experiments on larger models, like 13B, 32B. Only 7B is not convincing enough.
>
> > We thank the reviewer for the helpful suggestion. Following the suggestion, **we have added experiments on a larger model, Qwen2.5-14B-Instruct, to the Appendix G.7 in the revised paper**, to further validate the scalability of our findings.
> > | Qwen2.5-Instruct-14B | Parity-NL | MDM   | LLC   | CLF   | MOAS  | ObjC  | NumS  | Average |
> > |----------------------|-----------|-------|-------|-------|-------|-------|-------|---------|
> > | Base Model           | 74.4%     | 32.3% | 43.3% | 89.9% | 60.5% | 47.7% | 72.4% | 60.1%   |
> > | Dola                 | 76.1%     | 28.5% | 42.5% | 91.0% | 58.1% | 47.3% | 70.8% | 59.2%   |
> > | TCR                  | 79.0%     | 37.4% | 45.0% | 94.3% | 66.0% | 54.1% | 75.8% | **64.5%**   |
> > | TCR+gold             | 87.7%     | 41.5% | 48.6% | 96.0% | 69.8% | 63.7% | 77.2% | **69.2%**   |
> >
> > **The results are consistent with those observed on the 7B model and further strengthen the effectiveness of TCR and TCR-gold.**

---

> ### Author Response · Authors · 2025-11-23
> **Response to Reviewer iLuH (2/2)**
>
> #### **(Q1)** The classifier is an interface to LLMs. Can we directly improve the LLM's ability by teaching it to stop searching the error trajectories?
>
> > Thank the reviewer for the very insightful question. Directly “teaching” the base model to stop exploring error trajectories through end-to-end fine-tuning is generally challenging for several reasons:
> > - **Lack of explicit supervision.**
> > Error reasoning trajectories (discussed in this paper) are latent signals (e.g., head representations) that the LLM internally produces. These signals are **not observable** in the training corpus, making it difficult for the LLM to learn such behavior directly through standard next-token prediction objectives.
> > - **High training cost and task specificity.**
> > Fine-tuning the LLM to internalize such control requires large amounts of task-specific data and computation, and often does **not generalize to out-of-distribution tasks**. In contrast, our classifier is lightweight, plug-and-play and generalizable to new tasks.
> > - **Stability and controllability.**
> > Modifying the base model parameters may **change its general reasoning behavior in unintended ways** (e.g., catastrophic forgetting [1]). Our approach avoids altering the underlying model itself and intervenes in the test-time, offering **more reliable and interpretable** control.
> >
> > Besides, we design an experiment, where we **directly fine-tune the base LLM to predict the ground-truth tokens at the token positions of key error types in the CoT reasoning traces**, to explore whether we can fine-tune the LLM to improve their reasoning hop generalization. Aligned with the experiments in our paper (Table 1 and Table 2), the training data include five in-distribution tasks (Parity-NL, MDM, LLC, CLF and MOAS) and we evaluate the fine-tuned model's in-distribution task generalization and out-of-distribution task generalization (ObjC and NumS).
> > | Qwen2.5-7B-Instruct | Parity-NL | MDM   | LLC   | CLF   | MOAS  | ObjC  | NumS  | Average |
> > |---------------------|-----------|-------|-------|-------|-------|-------|-------|---------|
> > | Base Model          | 48.3%     | 43.0% | 11.7% | 56.8% | 39.2% | 52.0% | 41.1% | 41.7%   |
> > | Direct Fine-tuning  | 56.2%     | 43.7% | 12.2% | 59.5% | 44.0% | 52.0% | 39.8% | 43.9%   |
> > | TCR                 | 60.4%     | 48.2% | 16.2% | 66.6% | 46.0% | 56.0% | 46.0% | **48.5%**   |
> > | TCR-gold            | 81.2%     | 58.3% | 23.0% | 71.3% | 62.0% | 76.0% | 54.5% | **61.3%**   |
> >
> > We can observe that **the fine-tuned LLM fails to learn stable mechanisms to stop searching the error trajectories**, while our proposed TCR and TCR-gold provide a **lightweight interface that allows us to effectively mitigate erroneous trajectories in a generalizable manner** through test-time intervention, meanwhile **without changing the model's original parameter**.
> >
> > We sincerely thank the reviewer for raising this valuable question and hope our response fully resolves the concern. **We have added the discussion and the experiments of “teaching the base model itself to improve reasoning hop generalization” to the Appendix G.6 in the revised paper.**
> >
> > [1] Retaining by Doing: The Role of On-Policy Data in Mitigating Forgetting, arXiv preprint 2025.
>
> #### **(Q2)** See weakness 2. How does the method perform on large models?
>
> > Thanks the reviewer for raising the question. Please refer to our response to **(W2)**.
>
>
> We sincerely appreciate the reviewer’s thoughtful feedback. If there are any additional questions or concerns, we would be more than happy to address them.

---

> > ### Comment · Reviewer_iLuH · 2025-11-27
> >
> > Thanks for your response. I kept the score unchanged.

---

> > > ### Author Response · Authors · 2025-11-27
> > > **Thanks for the reply**
> > >
> > > Hi Reviwer iLuH,
> > >
> > > Thanks for the reply. Again, we sincerely appreciate your dedicated effort in reviewing our manuscript!
> > >
> > >  Submission#11334 Authors

---

> ### Author Response · Authors · 2025-11-27
> **We would like to hear from Reviewer iLuH**
>
> Hi Reviewer iLuH,
>
> We sincerely appreciate your thorough review, and very insightful feedback on our work! As the rebuttal is coming to a close, we kindly remind you that we have submitted a response to your comments. We would be very grateful if you could confirm whether our responses have addressed your concerns? If you have any additional concerns, please don't hesitate to let us know. We would really appreciate the opportunity to discuss with you further.
>
> Thank you again!
>
> Submission#11334 Authors

---

### Official Review · Reviewer_ZZNy · 2025-11-01

**Soundness:** 3
**Presentation:** 4
**Contribution:** 2
**Rating:** 6
**Confidence:** 3

**Summary:**

The paper studies reasoning-hop generalization in LLMs. It decomposes chain-of-thought into hop-level positions, defines task-specific error types, and then uses logit-lens / knockout to identify answer-writing heads and processing heads. The core claim is that correct and erroneous trajectories coexist internally, and hop failure largely comes from a small set of erroneous processing heads dominating late layers. A test-time head-selection / knock-out procedure improves accuracy.

**Strengths:**

- The angle is genuinely new. Instead of saying “model can’t reason”, it argues that correct reasoning traces often coexist but get overshadowed by competing erroneous heads. This produces a novel mechanistic framing of hop failure.
- The paper is extremely well-structured. The narrative from “error type → internal mechanism → test-time correction” is clean and easy to follow.

**Weaknesses:**

**W1.** “error types” are human-defined task-specific taxonomies (Appendix E.1), not emergent internal state clusters of the model, so the mapping from annotation layer → mechanism layer is not guaranteed.

**W2.** Evaluation covers only synthetic / programmatic hop-style tasks. No natural-language reasoning benchmarks (e.g., GSM8K, MATH, AIME, Leetcode).

**W3** No experiments on reasoning-style models (RL-based, verifier-based, or scratchpad-policy optimized models), whose non-linear reasoning chain dynamics are known to differ substantially from vanilla SFT LLMs. So universality claims are untested.

**Questions:**

**Q1.Error-type identification in real-world NL reasoning.**

In this work, the key step for mechanistic probing is that “critical error types” are first localized (Appendix E.1) and then those token positions are used as entry points for head attribution. For general NL benchmarks, error boundaries are not syntactically aligned to deterministic step indices, and model outputs are free-form.

How do you envision the error-type mining procedure to scale to such settings? Would this require a learned error tagger? a verifier? or a retrieval oracle to define step boundaries?

**Q2. Applicability to non-linear reasoning policies (with backtracking / exploring).**

Your entire head-tracing argument implicitly assumes a linear monotonic accumulation of residual signal along a single chain (hop 1 → hop 2 → hop 3 …).
RL-style models perform backtracking, branch scoring, and dynamic re-evaluation of earlier states.

Do you believe the cp vs ep competition in late layers is still a meaningful concept in a policy that does dynamic branch selection?
If yes, is the right object to knock out still an “attention head” or does the unit become a search operator?

---

> ### Author Response · Authors · 2025-11-23
> **Response to Reviewer ZZNy (1/3)**
>
> We sincerely appreciate the reviewer's time, thoroughly review and valuable feedback on our work.
> **Below We detail our response to the weaknesses (i.e., W(1)$\sim$W(3)) and questions (i.e., Q(1)$\sim$Q(2)) point by point.**
>
> #### **(W1)** “error types” are human-defined task-specific taxonomies (Appendix E.1), not emergent internal state clusters of the model, so the mapping from annotation layer → mechanism layer is not guaranteed.
>
> > (W1) We thank the reviewer for raising this important and insightful point.
> > Indeed, the key “error types” in our paper are first defined at the annotation layer, serving as diagnostic entry points for subsequent mechanistic analysis. However, our experimental evidence consistently indicates that **these error types capture meaningful and shared erroneous mechanisms in the model, rather than being arbitrary, task-specific taxonomies**.
> > - **Each error type corresponds to a sparse and identifiable set of erroneous processing heads (ep heads).** For every error type, we are able to locate a coherent and sparse set of heads responsible for driving the corresponding erroneous reasoning pattern (Section 4, Figure 6c; Appendix F.4). This establishes **a concrete link between the human-defined error type and internal mechanisms.** For example, in Qwen2.5-7B-Instruct, Parity-NL(2) maps to the ep head set $\{a_{7}^{0},a_{22}^{8},...\}$.
> > - **Some ep heads are shared across tasks and across distinct error types.** As discussed in Section 5 (line 462 $\sim$ 465) in the paper, we find that several ep heads recur across different tasks and different error types. For instance, in Qwen2.5-7B-Instruct, head $a_{22}^{8}$ is an ep head for both Parity-NL(2) and MOAS(2). This cross-task recurrence suggests that **different error types (even from different tasks) may be rooted in similar underlying erroneous mechanisms**.
> > - **The trained cross-task classifier constructs the map between annotated error types to internal mechanisms.** Motivated by the above observation, in Section 5 we constructs a shared candidate set $\mathbf{H}$ (with $|\mathbf{H}|\approx10$) of cross-task ep heads. A classifier is then trained to automatically select the appropriate ep head to intervene on, based on the input context in which the erroneous reasoning occurs. As discussed in line 462 $\sim$ 463, this classifier achieves strong Hit@1 performance on both in-distribution and out-of-distribution tasks—substantially higher than random selection. Thus, the classifier effectively learns a mapping from input erroneous-context (“error type”) → internal mechanism (“ep head”), further supporting that the error types reflect meaningful mechanisms.
> > - **Why task-specific error types can map onto shared erroneous mechanisms?** Our analysis indicates that these **error types share fundamental structural patterns in their erroneous reasoning behaviors**. For example, Parity-NL(2) and MOAS(2) both map to the ep head $a_{22}^{8}$. Although Parity-NL(2) and MOAS(2) differ in surface formulation, they both exhibit a shortcut-style reasoning failure: under the reasoning-hop generalization setting, the model updates the current reasoning state based only on the immediately preceding hop rather than the full CoT history. This often causes the model to retrieve locally co-occurring information (e.g., the person right following “Jack”) even when multiple conflicting candidates appear in the input.
> > Such shortcut reasoning patterns are **not task-specific, but reflect emergent internal failure modes governed by shared erroneous mechanisms: i.e., the ep heads we identify.**
> >
> > Again, we would like to thank the reviewer for insightful comments, and hope our clarification addresses the reviewer's concern.

---

> ### Author Response · Authors · 2025-11-23
> **Response to Reviewer ZZNy (2/3)**
>
> #### **(W2)** Evaluation covers only synthetic / programmatic hop-style tasks. No natural-language reasoning benchmarks (e.g., GSM8K, MATH, AIME, Leetcode).
>
> > We appreciate the reviewer’s valuable comment. We fully agree that evaluating the method on natural-language reasoning benchmarks is important.
> > - Our work primarily focuses on the **reasoning hop generalization setting**, where the evaluation and mechansitic investigation require **precise control over both hop counts and intermediate reasoning states**. This motivates our use of synthetic / programmatic tasks, which allow such control but are **difficult to achieve in GSM8K, MATH, or AIME**.
> >
> > - To address the reviewer’s concern and to evaluate our method in a more realistic natural-language reasoning setting, we additionally include results on **Big-GSM** [1], a more complex variant of GSM8K specifically designed to require more reasoning hops. Importantly, we do **not separately train the classifier on Big-GSM**; instead, we **directly apply the classifier trained on our five in-distribution tasks** (Parity-NL, MDM, LLC, CLF, MOAS) to select which heads to knock out.
> >
> >   |       Big-GSM       | Base Model |       TCR      |
> >   |:-------------------:|:----------:|:--------------:|
> >   | Qwen2.5-7B-Instruct | 52.7% ± 0.4% | **54.4%** ± 0.5% |
> >   |    Phi-3-Instruct   | 52.5% ± 1.3% | **53.9%** ± 0.7% |
> >
> >   The results (see added experiments) show that our method consistently improves performance on Big-GSM, demonstrating its ability to generalize to more natural-language, long-chain reasoning tasks.
> > Again, we appreciate reviewer's valuable comment and **have added the experiment on Big-GSM to the Appendix G.8 in the revised paper**.
> >
> > [1] Unlocking the Capabilities of Thought: A Reasoning Boundary Framework to Quantify and Optimize Chain-of-Thought. NeurIPS 2024.
>
>
> #### **(W3)** No experiments on reasoning-style models (RL-based, verifier-based, or scratchpad-policy optimized models), whose non-linear reasoning chain dynamics are known to differ substantially from vanilla SFT LLMs.
>
> > Thank the reviewer for the insightful comment. We fully agree that advanced reasoning-style models (e.g., RL-trained or verifier-augmented LLMs) may exhibit more complex and non-linear reasoning [1] dynamics compared to standard SFT models.
> >
> > - **Reasoning hop generalization has been observed in state-of-the-art reasoning models.**
> > Recent studies, including Illusion of Thinking [2] and OMEGA [3], report that even the state-of-the-art reasoning-optimized models such as DeepSeek-R1 and Claude 3.7 Sonnet-Thinking continue to suffer from reasoning-hop generalization failures. These observations suggest that **the phenomenon we study is not unique to vanilla SFT models, and that understanding its root causes is of broad relevance**.
> > - **Scope clarification: our goal is to analyze the fundamental mechanism under the most controlled setting.**
> > Our primary aim in this work is to provide a mechanistic understanding of reasoning-hop generalization in its **simplest and most classical form**: i.e., standard Chain-of-Thought (CoT) on symbolic reasoning, arithmetic reasoning, and code execution tasks.
> > To **isolate the core mechanism**, we intentionally avoid the additional complexity introduced by RL training, verifier loops, backtracking policies, or reflective long-CoT procedures. Studying the phenomenon in a simplified setting is a standard approach in mechanistic interpretability [4,5] and enables us to **obtain sharper, more interpretable conclusions**.
> > - **Limitations explicitly acknowledged.**
> > We explicitly discuss this limitation in **Appendix B (“Limitations and Discussions”)**, noting that our analysis focuses on classical CoT and does not yet cover advanced long-CoT reasoning paradigms [1] of reasoning-style models that involve backtracking, reflection, or verification.
> > - **Towards broader applicability.**
> > We view our contribution as establishing the **foundational mechanism needed to understand more sophisticated reasoning models**. Extending our framework to RL-trained large reasoning models is an exciting direction for future work, and we believe the insights uncovered here **lay the groundwork** for such investigations.
> >
> > We sincerely the reviewer again for the thoughtful comment.
> >
> > [1]: Demystifying Long Chain-of-Thought Reasoning in LLMs. arXiv preprint 2025.
> > [2]: The Illusion of Thinking: Understanding the Strengths and Limitations of Reasoning Models via the Lens of Problem Complexity. arXiv preprint 2025.
> > [3]: OMEGA: Can LLMs Reason Outside the Box in Math? Evaluating Exploratory, Compositional, and Transformative Generalization. arXiv preprint 2025.
> > [4]: How to think step-by-step: A mechanistic understanding of chain-of-thought reasoning. TMLR 2024.
> > [5]: Arithmetic Transformers Can Length-Generalize in Both Operand Length and Count. ICLR 2025.

---

> ### Author Response · Authors · 2025-11-23
> **Response to Reviewer ZZNy (3/3)**
>
> #### **(Q1)** Error-type identification in real-world NL reasoning.
>
> > We sincerely appreciate the reviewer's valuable question. We agree that identifying “critical error types” in general NL reasoning tasks is less straightforward than in synthetic hop-style settings. Below we outline how we envision scaling our framework to NL benchmarks.
> >
> > - **NL reasoning also exhibits stable, step-structured error patterns.**
> > Although NL tasks lack explicit programmatic hops, prior work [1] shows that LLMs’ reasoning traces on NL reasoning tasks (e.g., MATH and GPQA) **naturally decompose into step-like segments**, and these segments consistently **exhibit shared error patterns** (e.g., skipped-thinking steps, redundant sub-derivations, repetition loops). These patterns provide natural and semantically meaningful anchors that can play the same role as our “critical error types” for mechanistic probing.
> >
> > - **Our synthetic error types already reflect patterns common in NL reasoning, enabling transfer.**
> > Several of our synthetic error types correspond to widely reported NL failure modes such as repeated generation [2] and shortcut reasoning [3]. Empirically, our TCR classifier, which was trained only on five synthetic in-distribution tasks, already improves performance on BIG-GSM [4] without defining any task-specific error types for this benchmark. This suggests that the **underlying erroneous mechanisms captured by TCR are not tied to synthetic tasks**. In future work, we plan to incorporate more diverse reasoning tasks into the in-distribution set, with the goal of learning broader error patterns that generalize more strongly to open-ended NL reasoning.
> >
> > [1] What Makes a Good Reasoning Chain? Uncovering Structural Patterns in Long Chain-of-Thought Reasoning. EMNLP 2025.
> > [2] Understanding the Repeat Curse in Large Language Models from a Feature Perspective. ACL 2025 Findings.
> > [3] Discovering Highly Influential Shortcut Reasoning: An Automated Template-Free Approach. arXiv preprint 2023.
> > [4] Unlocking the Capabilities of Thought: A Reasoning Boundary Framework to Quantify and Optimize Chain-of-Thought. NeurIPS 2024.
>
> #### **(Q2)** Applicability to non-linear reasoning policies (with backtracking / exploring).
>
> > We thank the reviewer for raising this insightful question. We agree that RL-style reasoning models incorporate backtracking, branch evaluation, and dynamic revisiting of past reasoning states that go beyond a single linear reasoning chain. Below we clarify how our framework may relate to such settings.
> >
> > - **Even in branching policies, each branch is internally a linear reasoning chain.**
> > Although RL-style models explore multiple paths, exhaustively searching the entire solution space is infeasible for high-complexity problems, as it would require exponential time and memory. **Each candidate branch still unfolds as a step-by-step sequence (a local “linear” chain)**. Thus, competitive interactions between cp heads and ep heads can meaningfully arise within each branch’s local trajectory. Our proposed TCR method can serve as an efficient mechanism for **correcting erroneous reasoning trajectories within each branch**.
> > - **cp vs ep competition remains relevant, but the operative unit may become higher-level.**
> > In dynamic search policies, an “error” may manifest not only within a branch but also in branch selection or backtracking decisions. In such cases, the natural intervention target may shift from individual attention heads to higher-level internal circuits that correspond to search operators. We see our head-level TCR as analyzing the fundamental building blocks on which such operators act.
> >
> > Overall, cp vs ep competition mechanism is still a meaningful concept for correcting erroneous reasoning trajectories within each branch, and may shift to higher-level internal search operator circuits in dynamic branch selection. we view our present findings as groundwork for mechanistically characterizing these more complex policies.
>
> We sincerely thank the reviewer again for the thorough review and thoughtful feedback on our work.
> Please let us know should here are any remaining concerns and we are definitely willing to discuss with you!

---

> ### Author Response · Authors · 2025-11-27
> **We would like to hear from Reviewer ZZNy**
>
> Hi Reviewer ZZNy,
>
> We sincerely appreciate your dedicated effort in reviewing our manuscript. We would like to follow up to see if our response addresses your concerns or if you have any further questions. We would really appreciate the opportunity to discuss this further if our response has not already addressed your concerns.
>
> Thank you again!
>
> Submission#11334 Authors

---

> > ### Comment · Reviewer_ZZNy · 2025-11-27
> >
> > The authors’ response has addressed my main concerns, and I am pleased to see that the method demonstrates effectiveness on Big-GSM. However, since no results on dedicated reasoning models are provided at this stage, I will keep my current score at 6.
> >
> > I believe a promising direction for future work is to examine reasoning models, e.g. whether self-correction behavior is functionally equivalent to inference-time suppression of Erroneous Processing Heads, following the perspective of [1]. A study along these lines would substantially strengthen the contribution. If such analysis becomes available, I would consider raising my score to 8 or even 10.
> >
> > [1] Physics of Language Models: Part 2.2, How to Learn From Mistakes on Grade-School Math Problems. Ye et al., 2024.

---

> ### Author Response · Authors · 2025-11-28
> **Thanks for the Reviewer ZZNy's reply!**
>
> Dear Reviewer ZZNy,
>
> We are very happy to know that your main concerns have been properly addressed by our response and you appreciate our experiment which demonstrates the effectiveness of TCR on Big-GSM.
>
> We truly appreciate your perspective that connecting reasoning models' self-correction behavior with inference-time suppression of erroneous processing heads could further broaden the impact of our paper. We are already conducting a follow-up analysis on examining reasoning models and will post the results as soon as they are ready and incorporate the findings into the revised version of the paper to further clarify the underlying mechanisms.
>
> ---
> > **[Update, Dec 02 UTC]**
> >
> > To examine the reasoning model, we conduct experiments with the [DeepSeek-R1-distill-Qwen-7B](https://huggingface.co/deepseek-ai/DeepSeek-R1-Distill-Qwen-7B) [1].
> We separately sample $10$ erroneous predictions of the Parity-NL (50-hop) task, the MOAS (50-hop) task, and the ObjC (30-hop) task. Then following the procedure in Section 4, we locate the $\text{top-}1$ ep heads for these tasks (i.e., $\mathbf{a_7^0}$  for the Parity-NL task, $\mathbf{a_6^0}$ for the MOAS task, and $\mathbf{a_8^4}$ for the ObjC task), respectively. We implement TCR by dynamically deactivating the task-specific $\text{top-}1$ ep head in the reasoning process when the entropy of the generated token exceeds a threshold ($0.4$). Below we show the average reasoning accuracy and response length before and after applying TCR.
> >
> > | Parity-NL| Accuracy | Response Length |
> > |----------|----------|-----------------|
> > | Original | 63.3%    | 1633.6          |
> > | TCR      | 66.9%    | 1365.3          |
> >
> > |   MOAS   | Accuracy | Response Length |
> > |----------|----------|-----------------|
> > | Original | 38.9%    | 1765.9          |
> > | TCR      | 44.1%    | 1454.3          |
> >
> > |  ObjC    | Accuracy | Response Length |
> > |----------|----------|-----------------|
> > | Original | 28.5%    | 1819.1          |
> > | TCR      | 31.7%    | 1688.5          |
> >
> > We can observe that after applying TCR, model’s average reasoning accuracy will increase (by $4.0\%$ averagely) and meanwhile the average response will shorten (by $13.6\%$ averagely), demonstrating **TCR's potentials with reasoning models**.
> >
> >
> > Thank you again for the very insightful comment! We **have already added the discussion regarding to the reasoning models to the Appendix G.9 of the revised paper**.
> >
> > [1] https://huggingface.co/deepseek-ai/DeepSeek-R1-Distill-Qwen-7B
> ---
>
> Thanks again for your time, very insightful comments and helpful suggestions, which have improved the clarity and quality of our paper.
>
> Submission#11334 Authors

---

### Author Response · Authors · 2025-12-03
**Brief summary of the discussion-phase interactions (1/2)**

Dear Area Chairs,

We sincerely thank you for your time and effort in overseeing the review process, especially under this year's added complexity due to the OpenReview identity‐leakage incident. We truly understand that navigating such uncertainty while ensuring fairness and quality places significant demands on your role.

In light of this, and following the program chairs' suggestion, we have provided a concise and transparent summary of the discussion-phase interactions to serve as a reference that eases your decision-making.

Our paper
- investigates the internal **mechanism** behind CoT reasoning hop generalization failures (i.e., correct reasoning trajectories often coexist in the models but get overshadowed by competing `erroneous processing attention heads`.) in LLMs, and
- provides an inference-time, lightweight, effective, and highly generalizable **approach**, TCR, that dynamically suppresses automatically identified `erroneous processing attention heads`, thereby enhancing chain-of-thought reasoning as the number of hops increases.

The positive acknowledgments from reviewers include
- (**Paper Quality**) The paper is **well-structured and easy-to-follow with clear logic and clean narrative** (Reviewer`ZZNy` and `VfVH`). The figures and tables are easy to understand and the appendix is helpful (Reviewer`C6Fu`);
- (**Mechanism**) The paper investigates the reasoning hop generalization problem, and **provides valuable insights** (Reviewer `iLuH` and `VfVH`) to mechanistically understand where and why the problem happens **from a genuinely new angle** (Reviewer `ZZNy`);
- (**Approach**) Based on the mechanistic findings, the paper proposes TCR, **an effective and lightweight solution** (Reviewer `ZZNy`, `iLuH`, `VfVH`, and `C6Fu`), which rectifies the errors in the inference-time and is validated on **a wide range of tasks across a reasonable number of models** (Reviewer `C6Fu`);
- (**Broader Impact**) The findings from this paper may have **broader impact on relevant research** of LLM reasoning (Reviewer `iLuH`) and phenomena of LMs struggling to be performant to varying context lengths (Reviewer `C6Fu`).

In the discussion period, we **responded to all weaknesses and questions** raised by the four reviewers and **updated the pdf file accordingly** with all changes highlighted in blue.

We are encouraged that prior to the large-scale identity leakage, three out of four reviewers explicitly acknowledged that their concerns had been addressed, resulting in a unanimous score of 6, 6, 6, 6.
- **Reviewer `ZZNy`** noted that **he/she would be willing to increase his/her score to 8/10, conditioned on our further analyses,** which we have already provided in the follow-up response.
- **Reviewer `C6Fu` raised his/her score from 4 to 6 on Nov 24 UTC [(OpenReview link)](https://openreview.net/revisions?id=Pjc2cK8YFB)** (three days before the large-scale identity leakage happened).
- **Reviewer `iLuH`** maintained his/her rating.


In the below thread, we provide a more detailed breakdown of the reviewer discussions to further clarify the points addressed and the consensus reached.

**1. Reviewer `ZZNy` (initial score = 6):**
> - **Our 1st response to Reviewer `ZZNy`**
>   - Add a new experiment that demonstrates the effectiveness of TCR on the natural language reasoning task, Big-GSM, to the **Appendix G.8** in the revised paper (**W2** and **Q1**).
>   - Clarify the scope of our work (i.e., provide a mechanistic understanding of reasoning-hop generalization in the most classical CoT and lay the groundwork to understand more sophisticated reasoning models) and its applicability to non-linear reasoning policies (**W3** and **Q2**).
>   - Establish the inherent nature (rather than human-defined) of error types with the existing evidence in our paper (**W1**) and discuss how to identify error types in real-world nature language reasoning (**Q1**).
>
> - **Feedback from Reviewer `ZZNy`:**
>   - “**The authors' response has addressed my main concerns**, and I am **pleased to see that the method demonstrates effectiveness on Big-GSM**.”
>   - “However, since no results on dedicated reasoning models are provided at this stage, I will keep my current score at 6. If such analysis becomes available, **I would consider raising my score to 8 or even 10.**” (**We have responded to the reviewer with such analysis in the 2nd follow-up response.**)
>
> - **Our 2nd response to Reviewer `ZZNy`**
>   - Add a new experiment that demonstrates the effectiveness of our proposed TCR with the reasoning model, DeepSeek-R1-distill-Qwen-7B, to the **Appendix G.9** in the revised paper (**W3** and **Q2**).

---

> ### Author Response · Authors · 2025-12-03
> **Brief summary of the discussion-phase interactions (2/2)**
>
> **2.Reviewer `iLuH` (initial score = 6):**
> > - **Our response to Reviewer `iLuH`**
> >   - Add a new experiment that demonstrates the effectiveness of our proposed TCR on the larger-scale model, Qwen2.5-14B-Instruct, to the **Appendix G.7** in the revised paper (**W2** and **Q2**).
> >   - Discuss why directly “teaching” the base model when to stop exploring error trajectories is challenging; add a new experiment that compares our original TCR, which equips the base LLM with a trained classifier to stop, with directly fine-tuning the base LLM to stop, to the **Appendix G.6** in the revised paper (**Q1**)
> >   - Discuss potential solutions to further narrow down the gap between the in-distribution and out-of-distribution performance. (**W1**)
> >
> > - **Feedback from Reviewer `iLuH`:**
> >   - “Thanks for your response. I kept the score unchanged.”
>
> **3. Reviewer `VfVH` (initial score = 6):**
> > - **Our response to Reviewer `VfVH`**
> >   - Discuss the computation overhead introduced by the TCR workflow, justify its acceptability, and add a new experiment that demonstrates TCR's transferability to the new task, Big-GSM, to the **Appendix G.8** in the revised paper (**W1**).
> >   - Added an analysis of the model's predictive entropy before and after applying TCR to **Appendix F.1 Table 5** in the revised paper. (**W2**)
> >   - Add a new experiment demonstrating that normalization is crucial for faithfully and stably locating erroneous heads to the **Appendix F.3, Figures 12(e,f)** in the revised paper. (**W3**)
> >   - Following the reviewer's suggestion, we add the discussion on the limitations of Logit Lens and the justification of its use in the paper with the existing evidence in the paper to the **Appendix B** in the revised paper. (**W4**)
> >   - Restructure Section 4 into smaller, clearer subsections, following the reviewer's suggestion. (**W5**)
> >
> > - **Feedback from Reviewer `VfVH`:**
> >   - N/A
>
> **4. Reviewer `C6Fu` (initial score = 4. Reviewer `C6Fu` increased the score to 6 on Nov 24 UTC.):**
> > - **Our response to Reviewer `C6Fu`**
> >   - Clarify the meaning of the term “reasoning hop” used in this paper and its difference from the “hop” used in the multi-hop QA literature; discuss the other line of related works on latent multi-hop reasoning failure interpretation in the **Introduction** of the revised paper. (**W1**, **W2**, **W3**, and **Q1**)
> >   - Explain the meaning of “answer” and the design rationale behind the comparative study of using correct versus erroneous predictions to localize answer-writing heads; add a new illustration figure (**Figure 3**) in **Section 4** of the revised paper; add a new comparison experiment using $10$ randomly selected tokens to locate answer-writing heads to **Appendix F.3 Figure 13** in the revised paper. (**W4** and **W5**)
> >   - Add a new experiment that systematically compares TCR with LOFIT (NeurIPS 2024) to **Appendix G.5** in the revised paper. (**W6**)
> >   - Explain how the program we propose to automatically identify and label errors works. (**Q2**)
> >   - Add a new table (**Table 3**) to the **Appendix E.1** in the revised paper to explicitly list all tasks together with the names of all possible error types. (**Q3**)
> >   - Add new experiments comparing the reasoning hop generalization performance between different implementations: directly training the base model itself versus our originally proposed TCR to the **Appendix G.6** in the revised paper, highlighting the necessity of the head classifier in our design. Discuss and compare the training costs. (**Q4** and **Q5**)
> >
> > - **Feedback from Reviewer `C6Fu` [(OpenReview link)](https://openreview.net/revisions?id=sN1BRJ3x5g):**
> >   - “Thank you for clarifying. It seems I was confused w.r.t. the vocabulary based on similar language in tangential work. **I appreciate the additional experiments. My concerns have been addressed. I have raised my score.**”
>
> Again, we sincerely appreciate your time in handling our submission and look forward to the possibility of our work contributing meaningfully to the field.
>
> Best Regards,
>
> Submission#11334 Authors

---

### Meta-Review · Area_Chair_hFLS · 2026-01-07

**Summary:**

This paper analyzes reasoning hop generalization failures in CoT-style LLMs and argues that such failures arise from internal competition between correct and erroneous reasoning signals, driven by a small number of erroneous processing attention heads. The authors support this claim through detailed token-level error analysis, attention head attribution and knockout experiments, and propose a test-time intervention (TCR) that suppresses these heads to improve performance. Reviewers consistently found the framing novel and the analysis careful, particularly from a mechanistic interpretability perspective (ZZNy, VfVH, iLuH). The discussion phase was substantial, with many added experiments and clarifications. Overall, while some scope-related uncertainties remain, reviewer sentiment is uniformly at or slightly above the acceptance threshold, which leads me to lean toward acceptance.

**Reviewer Concerns:**

Several major concerns were addressed in rebuttal. In particular, C6Fu’s detailed critiques on terminology, error definitions, baselines, and the comparison between training the base model versus test-time intervention were resolved with extensive clarification and new experiments, leading that reviewer to raise their score. Other concerns were mitigated but not fully eliminated. The authors added evaluations on Big-GSM, larger models, and a reasoning-style model (DeepSeek-R1-distill), partially addressing earlier concerns about limited evaluation scope (ZZNy, iLuH, VfVH). However, evidence on reasoning-optimized or non-linear policies remains limited, and the relationship between the proposed head-competition mechanism and self-correction behavior in RL-style models is still suggestive rather than definitive (ZZNy). Additionally, while the authors argue that error types map to shared internal mechanisms, the approach relies on relatively clean hop structure and identifiable critical token positions, leaving some uncertainty about robustness in more open-ended NL reasoning. These remaining issues appear to be about generality and scope rather than technical soundness.

**Reviewer Scores:**

Reviewer scores converged to a uniform assessment near the acceptance threshold. C6Fu explicitly raised their score from 4 to 6 after rebuttal. The other reviewers (ZZNy, iLuH, VfVH) maintained scores of 6 throughout the discussion and did not introduce new unresolved technical objections. Taken together, the reviews reflect consistent recognition of the paper’s technical quality and novelty, with remaining reservations focused on how broadly the conclusions generalize. Given this uniform, borderline-positive signal, I lean toward acceptance.

---

### Decision · Program_Chairs · 2026-01-26

Accept (Poster)